# Training Free Guided Flow Matching with Optimal Control

**Luran Wang[1], Chaoran Cheng[2], Yizhen Liao[3], Yanru Qu[2], Ge liu[2]**
[1]University of Cambridge
[2]University of Illinois Urbana-Champaign
[3]Tsinghua University
`lw703@cam.ac.uk, chaoran7@illinois.edu`
`liaoyz0711@gmail.com, yanruqu2@illinois.edu`
`geliu@illinois.edu`

## Abstract

Controlled generation with pre-trained Diffusion and Flow Matching models has vast applications. One strategy for guiding ODE-based generative models is through optimizing a target reward $\mathcal{R}(x_1)$ while staying close to the prior distribution. Along this line, some recent work showed the effectiveness of guiding flow model by differentiating through its ODE sampling process. Despite the superior performance, the theoretical understanding of this line of methods is still preliminary, leaving space for algorithm improvement. Moreover, existing methods predominately focus on Euclidean data manifold, and there is a compelling need for guided flow methods on complex geometries such as SO(3), which prevails in high-stake scientific applications like protein design. We present OC-Flow, a general and theoretically grounded training-free framework for guided flow matching using optimal control. Building upon advances in optimal control theory, we develop effective and practical algorithms for solving optimal control in guided ODE-based generation and provide a systematic theoretical analysis of the convergence guarantee in both Euclidean and SO(3). We show that existing backprop-through-ODE methods can be interpreted as special cases of Euclidean OC-Flow. OC-Flow achieved superior performance in extensive experiments on text-guided image manipulation, conditional molecule generation, and all-atom peptide design.

## 1 Introduction and related work

SDE and ODE-based generative models such as diffusion and continuous normalizing flow (CNF) have exhibited excellent performance on various domains such as images (Ho et al., 2020; Esser et al., 2024), audio (Zhang et al., 2023; Défossez et al., 2022), and discrete data (Lou et al., 2023; Cheng et al., 2024). Particularly, the simplicity of Riemannian Flow Matching on SO(3) manifold (Chen & Lipman, 2023) empowers *de novo* generation of small molecules (Song et al., 2024; Xu et al., 2023) and proteins (Yim et al., 2023; Bose et al., 2023; Li et al., 2024), leading to enormous advancement in biomedicine. Controlled generation from pre-trained diffusion and flow matching priors has gained growing interest in numerous fields, as it encompasses a wide range of practical tasks including constrained generation (Giannone et al., 2023), solving inverse problems (Liu et al., 2023; Ben-Hamu et al., 2024), and instruction alignment (Black et al., 2023; Esser et al., 2024).

There are several lines of work for guiding diffusion and flow models. Classifier-free guidance (CFG) (Ho & Salimans, 2022) trains conditional generative models that take conditions as input. Reward fine-tuning approaches update the generative model parameters to align with certain target objective functions (Black et al., 2023). Both methods require specialized training routines, which are costly and not extendable to new tasks. Training-free guidance on diffusion (Kawar et al., 2022; Chung et al., 2024; Song et al., 2023) alters the scores in the SDE generation process with the gradients from the target function to achieve posterior sampling. These guidances often rely on strong assumptions of the denoising process and require estimating target function gradients w.r.t noised samples which are often intractable. Accurate posterior sampling is only guaranteed for a limited family of objective functions such as linear. Therefore, efforts that deploy such guidance to flow models by bridging the ODE path and SDE path share similar constraints (Pokle et al., 2023; Yim et al., 2024).

Notably, two recent works showed the effectiveness of guiding pre-trained flow models by differentiating through the ODE sampling process, outperforming popular guidance-based approaches. Particularly, Ben-Hamu et al. (2024) differentiates a loss $R(x)$ through the forward-ODE w.r.t the initial noise $x_0$, which induces implicit regularization by projecting the gradient onto the "data manifold" under Gaussian path assumption. This strong confinement to the prior might hinder optimization when the target reward function diverges from the prior distribution. Liu et al. (2023) formulates an optimal control problem where a control term $u_t$ at each timestep is solved to guide the ODE trajectory. However, the gradient decomposition trick used in Liu et al. (2023) ignores the running cost of control terms and thus could lead to suboptimal behavior. Despite the good performance, there is a lack of systematic theoretical analysis on the convergence behavior and explicit regularization of the differentiate-through-ODE approaches to better guide algorithm design in this space. Furthermore, existing works predominantly focus on the Euclidean manifold due to its simplicity, and there is a compelling need for a theoretically grounded guided flow matching framework on more complex geometries such as SO(3) which is heavily used in scientific applications.

To fill the gap between the practical applications of guided generation and theoretical grounds, we propose OC-Flow, a general, practical, and theoretically grounded framework for training-free guided flow matching under optimal control formulation. Our key contributions are as follows:

1. We formulate "controlled generation with pre-trained ODE-based priors" as an **optimal control** problem with a *control term* $u_t$ and a *running cost* that regulates the trajectory proximity to the prior model while optimizing for target loss. Building upon advances in optimal control theory, we develop effective optimization algorithms for **both Euclidean and SO(3) space** through iterative updates of a *co-state flow* and control term, with **theoretical guarantees** under continuous-time formulation.

2. In Euclidean space, we show that **running cost bounds the KL divergence** between prior and OC-Flow-induced joint distribution. We develop a simple algorithm for OC-Flow through iterative gradient update and provide convergence analysis. We further demonstrate that Dflow and Flow-grad can be interpreted as special cases of Euclidean OC-Flow, **providing a unified view** of the problem.

3. We present one of the first guided flow-matching algorithm on the SO(3) manifold with theoretical grounds. Our approach extends the Extended Method of Successive Approximations (E-MSA) to SO(3) with a rigorous convergence analysis. Additionally, we propose approximation techniques to enable computationally **efficient OC-Flow on** SO(3).

4. We provide an efficient and practical implementation of OC-Flow, by introducing the vector-Jacobian product and asynchronous update scheme. We show the effectiveness of our method with extensive empirical experiments, including **text-guided image manipulation**, controllable generation of **small molecules** on QM9, and energy optimization of flow-based **all-atom peptide design**.

## 2 PRELIMINARIES AND PERSPECTIVES ABOUT FLOW MATCHING

**Euclidean Flow Matching.** Flow matching (Lipman et al., 2022; Liu et al., 2022) provides an efficient framework for training a generative model by approximating the time-dependent vector field associated with the flow represented as $\psi_t : [0,1] \times \mathbb{R}^d \to \mathbb{R}^d$. This vector field $u_t : [0,1] \times \mathbb{R}^d \to \mathbb{R}^d$ defines a probability path of the evolution of an initial noise distribution, denoted by $p_0$, towards a target distribution, $p_1$, with the pushforward probability as $p_t := (\psi_t)_* p_0$. The dynamics of the vector field that governs this flow can be described by the flow ordinary differential equation (ODE) of the form $\dot{x}_t = u_t(x_t)$, where we follow the convention to use Newton's notation with respect to time $t$ and the state at time $t$ is given by $x_t := \psi_t(x_0)$. Lipman et al. (2022) demonstrates that a tractable flow matching objective can be obtained by conditioning on the target data $x_1$. The primary goal of conditional flow matching is to train a model, $f_t^p : [0,1] \times \mathbb{R}^d \to \mathbb{R}^d$, such that it minimizes the difference between its output and the ground truth conditional vector field as $\mathcal{L}_{\text{CFM}} = \mathbb{E}_{t,p(x_0,x_1)} \|f_t^p(x_t) - u_t(x_t \mid x_1, x_0)\|^2$. The trained model $f^p$ can be employed as the marginal vector field during the inference phase. In this context, once a noise sample $x_0$ is drawn, the system's evolution can be described by the following differential equation:

$$\dot{x}_t = f_t^p(x_t), \quad x_0 \sim p_0(x). \tag{1}$$

**Rotation Group SO(3).** The formulation of flow matching can naturally extend to Riemannian manifolds (Chen & Lipman, 2023). On the Riemannian manifold, a flow is defined as a time-dependent diffeomorphisms $\varphi_t : G \to G$, which describe the continuous evolution of points on the

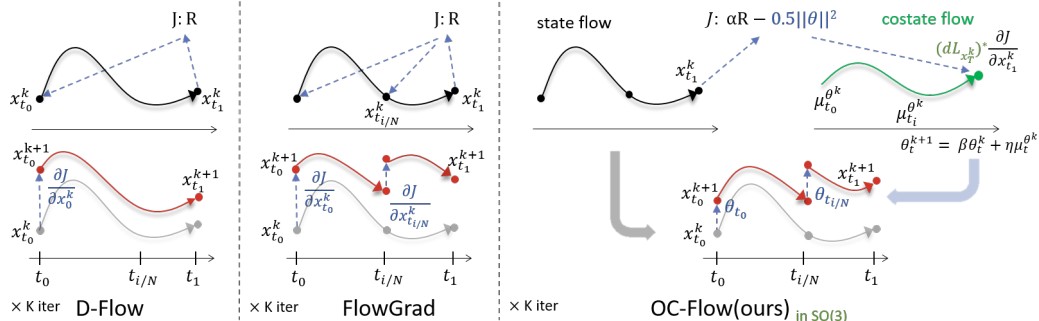

Figure 1: Comparison of backpropagation-through-ODE algorithms. For D-Flow (left) and FlowGrad (middle), the black curves represent the state trajectory at the $k$-th iteration, while the red curves show the updated trajectories at the $(k+1)$-th iteration, using gradient updates (blue dashed arrows). D-Flow updates the state at $t_0$ only, while FlowGrad propagates updates across all time steps. OC-Flow (right) incorporates the running cost and reward-weighting factor $\alpha$ into the terminal reward. The co-state flow $\mu_t$ (green curves) combines gradient information and system dynamics to iteratively update the control terms $\{\theta_t\}$, which in turn updates the states.

manifold over time, generated by a vector field $V$, with $V(p) \in T_p G$ for each $p \in G$ where $T_p$ is the tangent space at point $p \in G$. The flow evolves according to: $\frac{\mathrm{d}}{\mathrm{d}t} x_t = V(x_t) = L_{x_t} V(e)$. The CFM objective in Equation 1 can also be adapted for Riemannian flow matching, in which the ground truth vector field can be calculated as the time derivative using the exponential and logarithm maps. Details about rotation group $SO(3)$ can be found in Appendix A. In this work, we focus specifically on the $SO(3)$, the Riemannian manifold of all 3D rotations equipped with the canonical Frobenius inner product. In previous flow-based protein design models, each amino acid is associated with a rotation that defines its orientation (Yim et al., 2023; Bose et al., 2023; Li et al., 2024). Guiding such pre-trained generative models toward the desirable protein properties can potentially have a profound impact on the pharmaceutical industry.

## 3 OPTIMAL CONTROL FRAMEWORK FOR GUIDED FLOW MATCHING

The key challenge in guided generation is balancing optimization and faithfulness to the prior distribution. To address such a need, we propose the following framework. Given a pre-trained flow model, $f_t^p(x)$, parameterized by a neural network, our goal is to determine the optimal control terms $\theta_t$ that maximize the reward $\mathcal{R}(x)$ while maintaining the proximity of the resulting vector field to the original vector field induced by $f_t^p(x)$. The reward can be customized for diverse tasks such as inverse problem $\mathcal{R} = \|H(x) - y\|^2$, conditional generation $\mathcal{R} = (f(x) - c)^2$, and constrained generation $\mathcal{R} = \|x - y\|^2$. To ensure proximity, we incorporate a penalty on the state trajectory or control terms $\int_0^T L(\theta_t)$, also known as the *running cost*. Optionally, one may also introduce a metric $d(\cdot, \cdot)$ to penalize the deviation between the new terminal state $x_1^\theta$ and the prior terminal state $x_1^p$. The modified terminal reward function is then defined as: $\Phi(x_1^\theta) = \mathcal{R}(x_1^\theta) - d(x_1^\theta, x_1^p)$. and scaling the terminal reward by a constant $\alpha$, we can formulate the problem as a standard optimal control task:

$$J(\theta) := \alpha \Phi(x_T^\theta) + \int_0^T L(\theta_t)\, \mathrm{d}t \quad \text{subject to} \quad \dot{x}_t^\theta = h_t(x_t^\theta, \theta_t). \quad (2)$$

A fundamental result in optimal control theory is Pontryagin's Maximum Principle (PMP) (Pontryagin (2018)), which provides the necessary conditions for optimal solutions in control problems. Specifically, at the core of PMP is the introduction of the Hamiltonian function, $H$. This Hamiltonian is defined in terms of the state of the system, the control, and a new entity called the *co-state* $\mu$, which resides in the cotangent space of the state manifold:

$$H(t, x, \mu, \theta) = \langle \mu, h_t(x_t^\theta, \theta_t) \rangle - L(\theta). \quad (3)$$

The co-state $\mu_t$, also known as the adjoint variable, encodes the influence of the terminal cost function. Their evolution captures how the sensitivity of the system impacts the cost function, ensuring that the state variables evolve in accordance with the system dynamics. Consequently, in optimal control, the Hamiltonian must be maximized by jointly evolving the states and costates according to a system of coupled differential equations. The details of PMP conditions can be found in Appendix B.1.

## 3.1 OC-Flow on Euclidean Manifold

We first develop the algorithm and theoretical analysis for OC-flow in Euclidean space. One of the simplest choices for the control term is an additive control (Kobilarov & Marsden (2011)), which directly perturbs the prior trajectory. In fact, with the linear expansion, the additive control could be seen as a general case and is widely used in optimal control. Specifically, with $\theta_t$ representing the control input, the new state dynamics and the corresponding running cost can be defined as:

$$\dot{x}_t = h_t(x_t^\theta, \theta_t) = f_t^p(x_t) + \theta_t \quad L(\theta_t) = -\frac{1}{2}\|\theta_t\|^2. \tag{4}$$

The running cost effectively acts as a constraint on the trajectory to encourage proximity to the original prior distribution. To better understand the effect of running costs on the guided distribution, we provide the following proposition to formally prove that it can control deviation from the prior distribution measured by KL-divergence.

**Proposition 1.** *For Affine Gaussian Probability Path, the expectation of the running cost upper bounds the KL divergence between the prior joint distribution $p_1(x^p, x_1) = p_1(x^p|x_1)p_{data}(x_1)$ and the joint distribution after guidance $p_1(x^\theta, x_1) = p_1(x^\theta|x_1)p_{data}(x_1)$, with $x_1 \sim p_{data}$, $x^p$ induced by prior conditional vector field $u_t^p(x|x_1)$ and $x^\theta$ sampled by applying control $\theta_t(x_1)$ on $u_t^p$:*

$$\mathbb{E}_{x_1 \sim p_{\text{data}}(x_1)}\left[\frac{1}{2}\int_0^1 \|\theta_t(x_1)\|^2\,dt\right] \geq C \cdot \text{KL}\left(p_1(x^\theta, x_1)\|p_1(x^p, x_1)\right). \tag{5}$$

*Furthermore, for square-shaped data $x$ with non-zero probability path, the expectation of the running cost, combined with the $L_1$-distance between the prior sample $x_1^p$ and the corresponding guided sample $x_1^\theta$, upper bounds the KL divergence between the marginal distributions of the prior model $p_1^p$ and the guided model $p_1^\theta$:*

$$\mathbb{E}_{x_1^p \sim p_1^p(x)}\left[A\|x_1^p - x_1^\theta\| + B\int_0^1 \|\theta_t(x_1^p)\|^2\,dt\right] \geq \text{KL}(p_1^p \parallel p_1^\theta). \tag{6}$$

One effective approach for directly applying PMP to optimal control tasks is the Extended Method of Successive Approximations (E-MSA) (Li et al., 2018). E-MSA builds upon the basic MSA algorithm (Chernousko & Lyubushin (1982)), which iteratively updates the terms in the PMP conditions (Appendix B.2). The primary enhancement of E-MSA over the basic MSA is its ability to extend convergence guarantees beyond a limited class of linear quadratic regulators (Aleksandrov (1968)).

---

**Algorithm 1** OC-Flow on Euclidean Space

1: **Given:** Pre-trained model: $f^p$, initial state: $x_0$
2: **Initialize:** Control terms $\theta^0$, learning rate $\eta$, weight decay $\beta$
3: **for** $k = 0$ to MaxIterations **do**
4:   Solve for the state trajectory:
  $$X_{t+\Delta t}^{\theta^k} = X_t^{\theta^k} + \left(f^p(t, X_t^{\theta^k}) + \theta^k\right)\Delta t$$
5:   Update control:
  $$\theta_t^{k+1} = \beta\theta_t^k + \eta\nabla_{x_t}\Phi(X_1^k)$$
6: **end for**

---

A key assumption is the global Lipschitz condition for the functions involved. However, note that this assumption can be relaxed to a local Lipschitz condition if we can demonstrate that $x_t^\theta$ is bounded, which can be safely assumed provided that appropriate regularization techniques are applied. Furthermore several prior work has shown the Lipschitz continuity for the deep learning models. (Gouk et al. (2021), Khromov & Singh (2024))

When the E-MSA algorithm is applied to the guided controlled generation task on Euclidean space, the trajectory of the co-states $\mu_t$ can be calculated in closed form. Specifically, they can be expressed as $\nabla_{x_t}\Phi(X_1^k)$ which enables us to derive the following update rule with convergence guarantees:

**Theorem 2:** *Assume that the reward function, the prior model, and their derivatives satisfy Lipschitz continuity, bounded by a Lipschitz constant $L$. Utilizing the E-MSA, for each iteration $k$, for each constant $\gamma > 2C$ with $C$ is a function of $L$, such that under the addictive control and the running cost defined in Equation 4, the optimal update is following:*

$$\theta_t^{k+1} = \frac{\gamma}{1+\gamma}\theta_t^k + \frac{\alpha}{1+\gamma}\nabla_{x_t}\Phi(X_1^k). \tag{7}$$

*This update rule for the control term $\theta_t$ guarantees an increase in the objective function defined in Equation 2:*

$$J(\theta^{k+1}) - J(\theta^k) \geq \left(1 - \frac{2C}{\gamma}\right)\epsilon_\gamma^k, \quad \epsilon_\gamma^k \geq 0. \tag{8}$$

In practice, solving continuous ODEs requires discretization. The discretized version of the proposed algorithm is outlined in Algorithm 1. In this formulation, the weight decay term is parameterized as $\beta = \frac{\gamma}{1+\gamma}$, and the learning rate is defined as $\eta = \frac{\alpha}{1+\gamma}$. As demonstrated in Appendix C.3, the discretization error introduced by the Euler method is of the order $O(\Delta t)$. This error diminishes as the number of ODE steps increases, ensuring the algorithm converges to the global optimum.

## 3.2 Practical implementation and acceleration

### 3.2.1 Adjoint method and vector-jacobian product

A significant portion of the computational time in the OC-Flow algorithm is spent on evaluating the gradient $\nabla_{x_t} \Phi(x_1)$. The computational cost of directly back-propagating through ODE with vanilla Autograd requires saving all intermediate computation values, which results in a demanding memory complexity of $O(ND^2)$ (Pan et al., 2023; Chen et al., 2018) where $N$ is the ODE steps. We instead employ the adjoint method where the gradient $\nabla_{x_t} \Phi(x_1)$ is computed using a vector-Jacobian product by the double-backwards trick, which reduces the memory cost to $O(D^2)$.

$$\nabla_{x_{k/N}} \Phi = \left( \nabla_{x_{(k+1)/N}} \Phi \right) \cdot \Phi_{x_{(k+1)/N}} \left( x_{k/N} \right). \tag{9}$$

### 3.2.2 Asynchronous setting for Flexible Update Scheduling

In practice, discretization techniques are employed to simulate the ODEs governing both the state trajectory $x_t$ and the co-states $\mu_t$ and operate in a *synchronous* setting, where the number of time steps for the state trajectory $x_t$ coincides with the number of control terms $\theta_t$.

Here we show that OC-Flow can be extended to an *asynchronous* framework, providing greater flexibility in scheduling. We subdivide the time interval $\Delta t$ into $N$ equally spaced subintervals. Let $\{x_t\}$ denote the state trajectory over the time interval $[t, t + \Delta t]$, and $\{x_t^\theta\}$ represent the trajectory when the control term $\theta_t$ is applied in the first subinterval. The trajectory can be approximated as:

$$x_{t+\Delta t} = x_t + \frac{\Delta t}{N} \sum_{l=1}^{N-1} f^p \left( x_{t+\frac{l\Delta t}{N}}^\theta \right) + \frac{\Delta t}{N} \theta_t \approx x_t + \Delta t \left( \frac{1}{N} \sum_{l=1}^{N-1} f^p \left( x_{t+\frac{l\Delta t}{N}} \right) + \frac{1}{N} \theta_t \right). \tag{10}$$

Consequently, the asynchronous setting allows the algorithm to be applied without modification while enabling finer updates to both the control terms and state trajectories by adjusting the frequency $N$ of control term updates relative to the state trajectory simulation (see Appendix C.4 for the proof and justification of the approximations in Equation 10). In our peptide design experiment, the asynchronous setting is applied for efficient computing.

Table 1: Comparison of runtime and memory complexity of different methods used in backprop-through guided-ODE in Euclidean and SO(3) manifold. For complexity, $N$ is the number of ODE steps, $n$ is the number of effective control terms with synchronized and in the range $[1, N]$ and $D^2$ is the complexity of computing 1-step gradient (VJP or Autograd), D depends on data and model size.

| | Number of Control Terms | Running Cost | Memory Complexity | Runtime Complexity | Convergence to Optimal | Generalization to SO(3) |
|---|---|---|---|---|---|---|
| OC-Flow | $n$ | $\|\theta_t^2\|$ | $O(D^2)$ | $O(nD^2)$ | ✓ | ✓ |
| FlowGrad | $n$ | 0 | $O(D^2)$ | $O(nD^2)$ | ✗ | ✗ |
| D-Flow | 1 | Implicit | $O(ND^2)$ | $O(ND^2)$ | ✗ | ✗ |

## 3.3 Connection to other backprop-through guided-ODE approaches

Several previous works discussed backprop-through-ODE guidance in flow-matching models. Notable examples include D-Flow (Ben-Hamu et al., 2024) and FlowGrad (Liu et al., 2023). An illustration of their algorithms and ours is shown in Figure 1. In this section, we demonstrate that our framework is more general, and both of these methods can be viewed as special cases of our algorithm.

FlowGrad formulates the optimization task in a manner similar to our optimal control target in Equation 2. Specifically, it directly applies gradient descent to the control variables:

$$\theta_t^{k+1} = \theta_t^k + \alpha \nabla_{\theta_t} \Phi(X_1^k) = \theta_t^k + \frac{\alpha}{N} (\nabla_{x_t} X_{t+\Delta t})^{-1} \nabla_{x_t} \Phi(X_1^k), \tag{11}$$

which can be interpreted as a limiting case of our algorithm in Equation 7, where $\gamma \to \infty$ and given with $dt \to 0$, $\nabla_{x_t} X_{t+\Delta t} \to I$. However, as shown in Equation 8, the convergence rate is a complex

function of $\gamma$, so in practice, $\gamma$ is treated as a tunable parameter. FlowGrad's setting $\gamma \to \infty$ may undermine the convergence speed. D-Flow optimizes the reward by applying gradient descent to the initial noise $x_0$:

$$X_0^{k+1} = X_0^k + \text{LBFGS}(\nabla_{x_0} \Phi(X_1^k)). \tag{12}$$

In fact, with the update rule $x_{t+dt} = x_0 + f(x_0)\, dt + \theta_t\, dt$, the update of $\theta_0$ can be seen as an increment to $x_0$. The LBFGS algorithm provides a dynamic learning rate, aligning with our framework, where $\gamma$ is allowed to vary across iterations. Hence, D-Flow can be viewed as a special case of our asynchronous algorithm when the number of control terms is 1. A more detailed comparison of the algorithms can be found in Table 6 and additional discussion on computation efficiency is in Appendix D.

# 4 OPTIMAL CONTROL FRAMEWORK FOR GUIDED FLOW MATCHING ON SO(3)

Most optimization algorithms are primarily designed for Euclidean spaces and face significant challenges when applied to non-Euclidean settings, such as the SO(3) manifold, which plays a crucial role in drug discovery and peptide design (Huguet et al. (2024)). This section extends the E-MSA algorithm to the SO(3) manifold and presents a rigorous proof of its convergence.

## 4.1 OC-FLOW FOR SO3

To begin, we define the vector field governing the system dynamics. The state trajectory, influenced by control terms $\theta_t \in \mathfrak{so}(3)$, evolves according to the following differential equation:

$$\dot{x}_t^\theta = x_t^\theta \left( f_t(x_t^\theta) + \theta_t \right). \tag{13}$$

In this work, the left-invariant vector field is utilized, under which the Hamiltonian can be shown to reduce to a linear functional in $\mathfrak{so}(3)^*$ (Jurdjevic (1996), Colombo & Dimarogonas (2020)). Given the co-state $\mu_t \in \mathfrak{so}(3)^*$, the Hamiltonian function is redefined as:

$$H : [0,T] \times \text{SO}(3) \times \mathfrak{so}(3)^* \times \mathfrak{so}(3) \to \mathbb{R}, \quad (t,x,\mu,\theta) \mapsto \mu_t \left( f_t(x_t^\theta) + \theta_t \right) - L(\theta). \tag{14}$$

A direct approach to apply PMP conditions on the SO(3) manifold involves iteratively updating the cotangent vector $\mu_t$ and the state trajectory $x_t$ as shown in Step 4 and Step 5 in Algorithm 2 and then apply an update rule to determine the control term $\theta_t$ for the subsequent iteration with weight decay $\beta$ and learning rate $\eta$ the update for $\theta_t$ can be written as:

$$\theta_t^{k+1} = \beta \theta_t^k + \eta \tilde{\mu}_t^{\theta^k}, \tag{15}$$

where $\tilde{\mu}_t^{\theta^k}$ is defined by $\langle \tilde{\mu}_t^{\theta^k}, v \rangle = \mu_t^{\theta^k}(v)$ with $\tilde{\mu}_t \in \mathfrak{so}(3)$ for all $v \in \mathfrak{so}(3)$. The existence of $\tilde{\mu}_t^{\theta^k}$ can be derived from the Riesz Representation Theorem (Goodrich (1970)). This formulation leads to the introduction of the OC-FLow algorithm on SO(3), as detailed in Algorithm 2.

---

**Algorithm 2** OC-Flow on SO(3)

---

1: **Given:** Pre-trained model: $f_e^p$, initial state: $x_0$
2: **Initialize:** Control term $\theta^0 \in \mathfrak{so}(3)$, learning rate $\eta$, weight decay $\beta$
3: **for** $k = 0$ to MaxIterations **do**
4:     Solve for the state trajectory: $\dot{X}_t^{\theta^k} = X_t^{\theta^k} \left( f^p(t, X_t^{\theta^k}) + \theta^k \right), X_0^{\theta^k} = x_0$
5:     Solve for the adjoint variables: $\dot{\mu}_t^{\theta^k} = -\text{ad}_{\frac{\partial H}{\partial \mu}}^* \mu^{\theta^k} - (dL_{x_T^\theta})^* \frac{\partial H}{\partial x}, \mu_T^{\theta^k} = (dL_{x_T^\theta})^* \nabla \Phi(x_T^\theta)$
6:     Update control: $\theta_t^{k+1} = \beta \theta_t^k + \eta \tilde{\mu}_t^{\theta^k}$
7: **end for**

---

## 4.2 CONVERGENCE OF OC-FLOW ON SO3

To derive the proof of the convergence of our Algorithm 2, we first establish that under the PMP conditions on SO(3), the objective function $J(\theta)$, as defined in Equation 2, can be bounded. This is formalized in the following proposition:

**Proposition 3:** *Assume that the reward function, the prior model, and their derivatives satisfy Lipschitz continuity, bounded by a Lipschitz constant $L$. Then, there exists a constant $C > 0$ such that for any $\theta, \phi \in \mathfrak{so}(3)$, the following inequality holds:*

$$J(\theta) + \int_0^1 \Delta_{\phi,\theta} H(t)\, \mathrm{d}t - C\|\phi_t - \theta_t\|^2\, \mathrm{d}t \le J(\phi), \tag{16}$$

*where $X^\theta$ and $P^\theta$ satisfy the PMP conditions on SO(3) manifold, and $\Delta H_{\phi,\theta}$ denotes the change in the Hamiltonian, defined as:*

$$\Delta H_{\phi,\theta}(t) := H(t, x_t^\theta, \mu_t^\theta, \phi_t) - H(t, x_t^\theta, \mu_t^\theta, \theta_t). \tag{17}$$

Proposition 2 provides a lower bound for the difference in the objective function values under two distinct control terms that satisfy the PMP conditions described by the Hamiltonian equations in Appendix B.1.

However, applying this result directly as an optimization algorithm presents several challenges. First, the difference in the Hamiltonian $\Delta H_{\phi,\theta}(t)$ is not inherently bounded. Second, the term $\|\phi - \theta\|^2$ is non-negative, which complicates the minimization process. To address these issues, inspired by the method of E-MSA, we introduce a positive constant $\gamma$ and define an *Extended Hamiltonian*:

$$\tilde{H}(t, x, \mu, \theta, \phi) := H(t, x, \mu, \theta) - \frac{\gamma}{2}\|\theta - \phi\|^2 = \langle \mu, f_t(x) + \theta \rangle - \frac{1}{2}\|\theta\|^2 - \frac{\gamma}{2}\|\theta - \phi\|^2. \tag{18}$$

The introduction of the extended Hamiltonian enables the combination of the original Hamiltonian with the penalty term $\|\phi - \theta\|^2$ into a unified expression that can be optimized jointly. A natural approach to achieve this is by updating $\theta$ to maximize the Extended-Hamiltonian. The resulting update rule is given by:

$$\theta_t^{k+1} = \arg\max_\theta \tilde{H}(t, x^{\theta^k}, \mu^{\theta^k}, \theta, \theta^k) = \frac{\gamma}{1+\gamma}\theta^k + \frac{1}{1+\gamma}\mu_t^\theta = \beta\theta^k + \eta\mu_t^{\theta^k}. \tag{19}$$

By performing this maximization step at each iteration, we ensure that the change in the Extended-Hamiltonian, $\Delta\tilde{H}$, is non-negative, indicating that the algorithm progresses towards an optimal solution. Furthermore, we can show that when the update process converges, i.e., when $\Delta\tilde{H} = 0$ or equivalently $\Delta H = 0$, the algorithm has reached the optimal control point. These insights can be formalized in the following proposition:

**Proposition 4:** *Let $X^\theta$ and $P^\theta$ satisfy the PMP conditions . If the update rule follows Algorithm 2, we define $\epsilon_k := \int_0^1 \Delta_{\theta^{k+1},\theta^k} H(t)\, \mathrm{d}t$, and $\epsilon_k$ is bounded as:*

$$\epsilon_k := \int_0^1 \Delta_{\theta^{k+1},\theta^k} H(t)\, \mathrm{d}t, \qquad \lim_{k\to\infty} \epsilon_k = 0. \tag{20}$$

*Furthermore, when $\epsilon_k = 0$, we have $\theta = \theta^* := \arg\max_\theta J(\theta)$*

With these results, we can now extend the result in E-MSA to the $SO(3)$ manifold and establish a bound for the optimization algorithm based on the derived theoretical properties:

**Theorem 5:** *Assume that the reward function, the prior model, and their derivatives satisfy Lipschitz continuity, bounded by a Lipschitz constant $L$. Let $\theta^0 \in \mathfrak{so}(3)$ be any initial measurable control with $J(\theta^0) < +\infty$. Suppose also that $\inf_{\theta \in \mathfrak{so}(3)} J(\theta) > -\infty$. If the update of $\theta$ satisfies equation 19, for sufficiently large $\gamma$, the following inequality holds for some constant $D > 0$:*

$$D\epsilon_k \le J(\theta^{k+1}) - J(\theta^k). \tag{21}$$

Therefore, by invoking Proposition 3, we can conclude that after each update, the target function is non-decreasing and when the update process terminates, the optimal solution has been attained. This establishes the convergence of the OC-Flow algorithm on the $SO(3)$ manifold.

### 4.3 PRACTICAL IMPLEMENTATION

In practice, directly optimizing Algorithm 2 using existing ODE methods is challenging due to the nature of the adjoint variable $\dot{\mu}_t$, which is a linear functional in the dual space $\mathfrak{so}(3)^*$. Instead, we can optimize $\tilde{\mu}_t$ as defined in Section 4.1. We can decompose $\dot{\tilde{\mu}}_t$ into its projections onto a set of orthogonal bases within the $\mathfrak{so}(3)$ group.

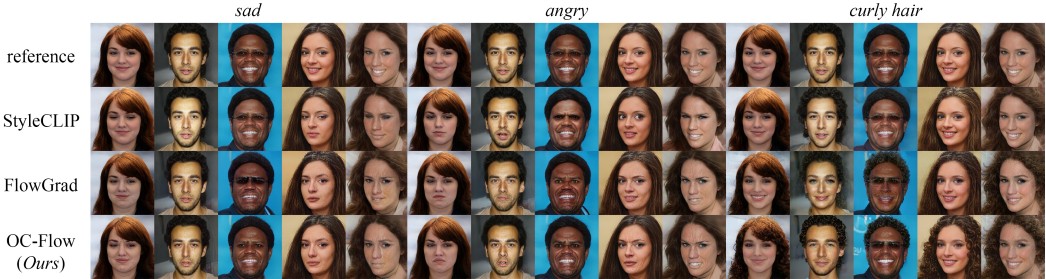

Figure 2: Visualization of text-guided generated faces with different expressions.

A frequently used choice for the basis in $\mathfrak{so}(3)$ is the canonical basis $\{E_1, E_2, E_3\}$ (McCann et al. (2023)) satisfying the condition $\langle v, E_i \rangle = 2$ for all $v \in \mathfrak{so}(3)$. Thus we can decompose the time derivative of the adjoint variable $\dot{\tilde{\mu}}_t$ as: $\dot{\tilde{\mu}}_i = \langle \dot{\tilde{\mu}}, E_i \rangle$ and $\dot{\tilde{\mu}} = \frac{1}{2} \sum_{i=1}^{3} \dot{\tilde{\mu}}_i E_i$. Thus, with the closed forms for the partial derivatives related to Hamiltonian, the practical update for $\mu_t$ in Algorithm 2 can be written as follows. The vector-Jacobian method can be applied to compute the term $\frac{\partial f_t^p}{\partial x}(x_t^k E_j)$ in Algorithm 2, which significantly reduces complexity from $O(D^4)$ to $O(D^2)$. Meanwhile, the method of asynchronous can also be applied. See Appendix C.4.

$$\dot{\tilde{\mu}}_{t,i}^k = -\langle \tilde{\mu}_t^k, [f_t^p + \theta_t, E_i] \rangle - \left\langle \tilde{\mu}_t^k, \frac{\partial f_t^p}{\partial x} x_t^k E_i \right\rangle, \quad \left\langle \tilde{\mu}_t, \frac{\partial f_t^p}{\partial x} x_t^k E_j \right\rangle = \text{Tr}\left( \tilde{\mu}_t^T \frac{\partial f_t^p}{\partial x}(x_t^k E_j) \right)$$

$$\tilde{\mu}_{t-\Delta t}^k = \tilde{\mu}_t^k - \frac{\Delta t}{2} \sum_{i=1}^{3} \dot{\tilde{\mu}}_{t,i}^k E_i, \quad \tilde{\mu}_{T,i}^k = \langle \nabla \Phi(x_T^k), x_T^k E_i \rangle. \tag{22}$$

## 5 EXPERIMENTS

### 5.1 TEXT-GUIDED IMAGE MANIPULATION

We first validate our OC-Flow on the traditional text-to-image generation task. Previous work has demonstrated the importance of alignment with the given text prompt using either automatic metrics or human preference as the reward (Black et al., 2023; Esser et al., 2024). In our text-guided image manipulation task, we want to guide the generative model pre-trained on the celebrity face dataset CelebA-HQ (Karras, 2017) to text guidance {sad, angry, curly hair} showing different facial expressions or traits. Following the same setup in Liu et al. (2023), given an input image $x_g$, the reward for alignment with the text prompt can be effectively evaluated by the CLIP model (Radford et al., 2021) pre-trained in a contrastive way to score the similarity between arbitrary image-text pairs. Following (Liu et al., 2023), we adopt the pre-trained Rectified Flow (RF) (Liu et al., 2022) as the generative prior. Inspired by Proposition 1, for this image task where the square-like assumption is satisfied, an extra terminal constraint $x_1^p - x_1^\theta$ is added as part of the terminal reward function.

We choose two state-of-the-art text-guided image manipulation baselines, StyleCLIP (Patashnik et al., 2021) and FlowGrad (Liu et al., 2023). We run Style-CLIP and FlowGrad with their official implementation and default parameter configurations. For *ours*, we set time step of 100, step size $\eta = 2.5$, weight decay of 0.995, the weight of the extra constraint of 0.4, and the number of optimization steps of 15. For qualitative comparison, we show generated examples of different text-guided expressions in Figure 2. Due to the large gap between reference and guided distributions, StyleCLIP fails to manipulate with sad.

Table 2: Comparison of methods on LPIPS, ID, and CLIP metrics. Lower LPIPS and ID indicate better performance, while higher ID and CLIP values are preferred.

| Method | LPIPS ↓ | ID ↑ | CLIP ↑ |
|---|---|---|---|
| CG + RF | 0.346 | 0.643 | 0.292 |
| CG + LDM | 0.383 | 0.513 | 0.298 |
| DiffusionCLIP | 0.398 | 0.659 | 0.285 |
| StyleCLIP + e4e | 0.359 | 0.704 | 0.267 |
| FlowGrad + RF | 0.302 | **0.737** | 0.299 |
| **OC-Flow (Ours)** | **0.207** | 0.732 | **0.302** |

Lacking in regularization, FlowGrad may change the content too much with curly hair. Our OC-Flow generally produces the best results with a good alignment with the text prompt while preserving the generative prior so as to produce reasonable faces that are not distorted much.

### 5.2 MOLECULE GENERATION FOR QM9

We further instantiate our OC-Flow for controllable molecule generation on the QM9 dataset (Ruddigkeit et al., 2012; Ramakrishnan et al., 2014), a commonly used molecular dataset containing small molecules with up to 9 heavy atoms from C, O, N, F. Following Hoogeboom et al. (2022); Ben-Hamu

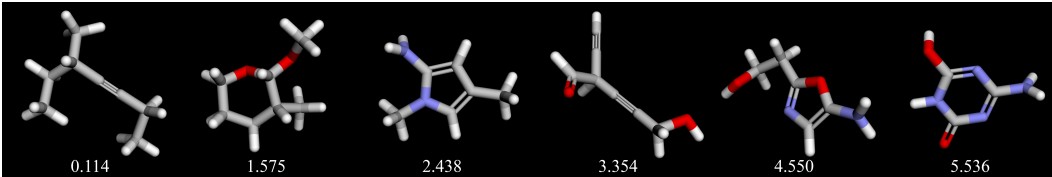

Figure 3: Visualization of OC-Flow generated molecules with various dipole moments condition.

et al. (2024), we target for conditional generation of molecules with specified quantum chemical property values including polarizability $\alpha$, orbital energies $\varepsilon_{\text{HOMO}}, \varepsilon_{\text{LUMO}}$ and their gap $\Delta\varepsilon$, dipole moment $\mu$, and heat capacity $c_v$. Such a conditional generation setting of molecules with desired properties has profound practical applications in drug design and virtual screening. To define the loss function, separate classifiers for each property are first trained to predict the property value for the generated molecule (Hoogeboom et al., 2022), and the loss can be then calculated as the mean absolute error (MAE) between the predicted and the reference property values. The pre-trained unconstrained generative model is taken from Song et al. (2024) (EquiFM), a flow-based generative model that uses an equivariant vector field parameterization for generating the atom coordinates and types via the learned flow dynamics. To demonstrate the zero-shot guidance performance on such a conditional generation task, we compare our approach with other gradient-based methods of D-Flow (Ben-Hamu et al., 2024) and FlowGrad (Liu et al., 2023) on the same pre-trained EquiFM. To be comparable to D-Flow, we follow its setting to use the L-BFGS optimizer with 5 optimization steps with linear search. We generate 1000 molecules for each property and report the MAE in Table 3. The unconditional EquiFM is also included as an upper bound for the guided models. It can be seen that our approach consistently outperforms both of them with lower MAEs, which better balances the reward optimization and the faithfulness to the prior. We provide guided generation samples in Figure 3 with respect to different target dipole moments. A clear trend from hydrocarbons with more symmetric structures to molecules with more high-electronegativity atoms of oxygen and nitrogen can be observed, indicating an increase in the dipole moment.

As we have theoretically demonstrated the impact of the regularization strength from the optimal control perspective, we further experiment with a different $\gamma$ and examine the quality of the conditionally generated molecules by evaluating additional metrics following Song et al. (2024). Specifically, we calculate the atom stability percentage (ASP), molecule stability percentage (MSP), and valid & unique percentage

Table 3: MAE for guided generations on QM9 (lower is better).

| Property | $\alpha$ | $\Delta\varepsilon$ | $\varepsilon_{\text{HOMO}}$ | $\varepsilon_{\text{LUMO}}$ | $\mu$ | $c_v$ |
| Unit | Bohr³ | meV | meV | meV | D | $\frac{\text{cal}}{\text{K·mol}}$ |
|---|---|---|---|---|---|---|
| OC-Flow(Ours) | **1.383** | 367 | **183** | **342** | **0.314** | **0.819** |
| D-Flow | 1.566 | **355** | 205 | 346 | 0.330 | 0.893 |
| FlowGrad | 2.484 | 517 | 273 | 429 | 0.542 | 1.270 |
| EquiFM | 8.969 | 1439 | 622 | 1438 | 1.593 | 6.873 |
| Classifier | 0.095 | 64 | 40 | 35 | 0.046 | 0.041 |

(VUP). Ideally, these metrics should not be greatly lower than the pre-trained model, and a higher strength of regularization should lead to higher scores. Indeed, as demonstrated in Table 4, in which we provide these scores for two different settings of $\gamma = 0.01$ and 10, all scores are higher with a higher strength of regularization at the cost of also a higher MAE. In this way, our OC-Flow effectively prevents exploitation from direct gradient descent that may hack the loss function and provides more flexible and fine-grained control over the guided generation.

Table 4: MAE and other evaluation metrics for our approach with $\gamma = 0.01$ / $\gamma = 10$.

| Property | $\alpha$ | $\Delta\varepsilon$ | $\varepsilon_{\text{HOMO}}$ | $\varepsilon_{\text{LUMO}}$ | $\mu$ | $c_v$ |
|---|---|---|---|---|---|---|
| MAE ↓ | 1.383 / 1.557 | 367 / 365 | 183 / 188 | 342 / 339 | 0.314 / 0.320 | 0.819 / 0.852 |
| ASP ↑ | 94.8 / 96.0 | 95.2 / 96.1 | 95.2 / 96.1 | 95.3 / 96.1 | 95.8 / 96.1 | 95.2 / 96.1 |
| MSP ↑ | 64.4 / 69.9 | 67.9 / 70.5 | 65.8 / 69.8 | 68.5 / 70.8 | 68.0 / 70.1 | 67.1 / 68.9 |
| VUP ↑ | 86.2 / 88.6 | 88.2 / 89.8 | 86.2 / 87.7 | 87.6 / 88.7 | 88.2 / 89.0 | 89.4 / 88.5 |

## 5.3 PEPTIDE DESIGN

We evaluate our OC-Flow approach for peptide backbone design using a test set derived from (Li et al., 2024), which includes 162 complexes clustered based on 40% sequence identity using mmseqs2 (Steinegger & Söding, 2017). Our experiments focus on PepFlow w/Bb, a model designed to exclusively sample peptide backbones while optimizing translations in Euclidean space and

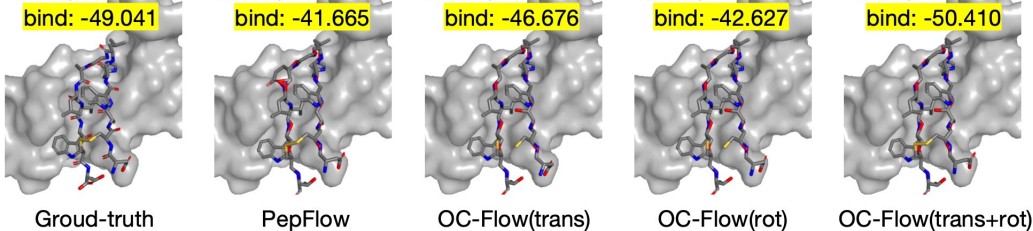

Figure 4: Visualization of OC-Flow generated peptide and unconditional generated peptide (5djd_C).

rotations in SO(3) space. The model employs the MadraX force field (Orlando et al., 2024) for energy optimization, and performance is evaluated using several key metrics. These metrics include MadraX energy, which assesses the total energy of the generated peptide structures, along with Rosetta-based measures of stability and affinity. Currently, our stability and affinity metrics are represented by their respective means: stability quantifies the energy states of peptide-protein complexes, while affinity measures the binding energies. In addition, we employ an existing metric, denoted as IMP, which measures the percentage of generated peptides that exhibit lower energy than the original ground truth. Additionally, we use the root-mean-square deviation (RMSD) to evaluate structural accuracy by aligning the generated peptides to their native structures and calculating the $C_\alpha$ RMSD. To further analyze structural characteristics, we compute the secondary-structure similarity ratio (SSR), which reflects the proportion of shared secondary structures, and the binding site ratio (BSR), which quantifies the overlap between the binding sites of the generated and native peptides on the target protein. Structural diversity is assessed using the average of one minus the pairwise TM-Score (Zhang & Skolnick, 2005) among the generated peptides, representing their dissimilarities.

We compare our method to the pre-trained unconditional PepFlow model (Li et al., 2024), serving as a baseline. We also include ablations where our model guides only translations (Euclidean) or rotations (SO(3)). As shown in Table 5, our OC-Flow method, applied to both Euclidean and SO(3) spaces, consistently outperforms the baseline, even though we only optimize for the Madrax target function. This indicates that our algorithm not only achieves higher target function scores but also captures more natural structural configurations. It generates peptide backbones that are more stable, energetically favorable, and diverse, while improving key metrics such as stability, affinity, IMP, diversity, SSR, and BSR. In comparison, optimizing in Euclidean space alone yields only marginal improvements in IMP, while optimizing rotations alone achieves comparable performance. More experimental details and ablation can be found in Appendix E.3.

Table 5: Evaluation of OC-Flow peptide design.

| | MadraX ↓ | RMSD ↓ | SSR % ↑ | BSR % ↑ | Stability ↓ | Affinity ↓ | Diversity ↑ | imp(%) ↑ |
|---|---|---|---|---|---|---|---|---|
| Ground-truth | -0.588 | - | - | - | -84.893 | -36.063 | - | - |
| PepFlow | -0.195 | 1.645 | 0.794 | 0.874 | -45.660 | -26.538 | 0.310 | 14.3 |
| OC-Flow(trans) | -0.229 | 1.774 | **0.797** | **0.876** | -48.380 | -27.328 | 0.323 | 14.4 |
| OC-Flow(rot) | -0.221 | **1.643** | 0.794 | 0.872 | -48.636 | -27.211 | 0.310 | 14.5 |
| OC-Flow(trans+rot) | **-0.263** | 2.127 | **0.797** | 0.869 | **-48.853** | **-27.468** | **0.338** | **15.0** |

## 6 CONCLUSIONS AND DISCUSSION

In this paper, we propose OC-Flow, a general and theoretically grounded framework for training-free guided flow matching under optimal control formulation. Our framework provides a unified perspective on existing backprop-through-ODE approaches and lays the foundation for systematic analysis of the optimization dynamics of this setting. Extensive empirical experiments demonstrate the effectiveness of OC-Flow. Future extensions of OC-Flow include generalizing beyond additive control terms and bridging connection with fine-tuning regimes where control terms can be solved as learning updates to the model parameters. Another extension could be scaling up the SO(3) OC-Flow to guide generative tasks for larger molecular systems such as protein motif scaffolding. One potential limitation of backprop-through-ode approaches, despite its superior result, is the higher computation cost compared to posterior sampling approaches. Such tradeoff has been demonstrated in Dflow/FlowGrad as well. Our practical implementations of OC-Flow improve the time and memory complexity (see analysis in Appendix D), where sampling on the image takes 216s compared to 15 minutes in D-Flow. We hope that our findings can guide algorithm design and motivate further model improvement in guided flow matching.

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

## A  BACKGROUND OF RIEMANNIAN MANIFOLD AND SO(3) GROUP

A Lie group $G$ is a smooth manifold equipped with group operations, such as multiplication and inversion, which are smooth maps. Specifically, $G$ is considered smooth when it possesses a $C^\infty$ differential structure. When $G$ is endowed with a left-invariant Riemannian metric, it becomes a Riemannian manifold, where the inner product of any two tangent vectors $v, w \in T_h G$ at a point $h \in G$ is preserved under left multiplication. This property is expressed as:

$$\langle L_h(v), L_h(w) \rangle = \langle v, w \rangle, \tag{23}$$

where $L_h : G \to G$ is the left multiplication map, and $\langle \cdot, \cdot \rangle : TG \times TG \to \mathbb{R}$ represents the inner product. Moreover, the tangent space at any point $x \in G$ is given by $T_x G = L_x T_e G$, where $T_e G$ is the tangent space at the identity element $e$, which is identified with the Lie algebra $\mathfrak{g}$. Consequently, the full tangent bundle $TG$ can be written as $G \times \mathfrak{g}$.

At each point $x \in G$, a tangent space $T_x G$ is attached, representing the space of tangent vectors at that point. The collection of these tangent spaces forms the tangent bundle $TG$, which itself is a smooth manifold. Additionally, for any point $h \in G$, the cotangent space $T_h^* G$ is defined as the dual space of $T_h G$, consisting of linear functionals (co-states) that act on the tangent vectors.

**Rotation Group** SO(3): The special orthogonal group SO(3), describing 3D rotations, is a compact 3-dimensional Lie group. Its Lie algebra $\mathfrak{so}(3)$ consists of skew-symmetric matrices. The group SO(3) is defined as:

$$\mathrm{SO}(3) = \{r \in \mathbb{R}^{3 \times 3} : r^\top r = r r^\top = I, \det(r) = 1\}. \tag{24}$$

It is a matrix Lie group, and its Lie algebra is given by:

$$\mathfrak{so}(3) = \{r \in \mathbb{R}^{3 \times 3} : r^\top = -r\}. \tag{25}$$

**Parametrizations of** SO(3): The skew-symmetric matrices $r \in \mathfrak{so}(3)$ can be uniquely represented by a vector $\omega \in \mathbb{R}^3$, such that for any $v \in \mathbb{R}^3$, $rv = \omega \times v$, where $\times$ denotes the cross product. This vector is known as the *rotation vector*, where its magnitude $\|\omega\|$ represents the angle of rotation, and its direction $e_\omega = \omega / \|\omega\|$ defines the axis of rotation. The mapping from $\mathbb{R}^3$ to the skew-symmetric matrix is referred to as the *hat operation*, $\hat{(\cdot)}$.

Another common parametrization of SO(3) is through *Euler angles*, described using three angles $(\phi, \theta, \psi)$. In the *x-convention*, the rotation is expressed as a sequence of three rotations: a rotation about the $z$-axis by $\phi$, followed by a rotation about the updated $x$-axis by $\theta$, and finally, a rotation about the updated $z$-axis by $\psi$.

One common inner product on $\mathfrak{so}(3)$ is the induced Frobenius inner product, given by:

$$\langle A, B \rangle = \mathrm{Tr}(A^\top B), \quad \forall A, B \in \mathfrak{so}(3), \tag{26}$$

which equips SO(3) with a Riemannian structure. The manifold SO(3) has constant Gaussian curvature and is diffeomorphic to a solid ball with antipodal points identified. The exponential map $\exp : \mathfrak{so}(3) \to \mathrm{SO}(3)$, originating from the identity element, is defined as matrix exponentiation:

$$\exp(A) = \sum_{k=0}^{\infty} \frac{A^k}{k!}, \quad \forall A \in \mathfrak{so}(3), \tag{27}$$

and can be represented more compactly via Rodrigues' rotation formula:

$$\exp(A) = I + \frac{\sin\theta}{\theta} A + \frac{1 - \cos\theta}{\theta^2} A^2, \quad \forall A \in \mathfrak{so}(3), \tag{28}$$

where $\theta = \|A\|_{\mathfrak{so}(3)} = \frac{1}{2}\|A\|_F$ is the rotation angle. Similarly, the logarithm map $\log : \mathrm{SO}(3) \to \mathfrak{so}(3)$, also originating from the identity, is the matrix logarithm:

$$\log(R) = \sum_{k=1}^{\infty} (-1)^{k+1} \frac{(R - I)^k}{k}, \tag{29}$$

or more compactly as:

$$\log(R) = \frac{\theta}{\sin\theta} A, \quad \forall R \in \mathrm{SO}(3), \tag{30}$$

where $A = \frac{(R - R^\top)}{2} \in \mathfrak{so}(3)$ and $\theta = \|A\|_{\mathfrak{so}(3)}$ is the rotation angle. In spherical geometry, the geodesic distance between two rotations is given by $d(R_1, R_2) = \|\log(R_1^\top R_2)\|_F$, and interpolation between rotations can be performed using $\exp(tA)$.

# B BACKGROUND OF PMP AND E-MSA

In this section, we shall introduce the details of the Pontryagin's Maximum Principle on both Euclidean Space and SO(3) manifold as well as the algorithms of MSA and E-MSA.

## B.1 PMP

As described in section 3, PMP offers a set of necessary conditions for an optimal control strategy. PMP states that for an optimal trajectory, there exists a co-state trajectory $\mu_t$ such that the Hamiltonian (Equation 3) is maximized (or minimized, depending on the problem) with respect to the control at every time step. Additionally, the state and co-state evolve according to a system of coupled differential equations, where the co-state variables evolve in the cotangent space of the state variables.

The governing differential equations are shown below:

**Pontryagin's Maximum Principle** Let $\theta^* \in \mathcal{U}$ be an essentially bounded optimal control, i.e. a solution to (2) with $\operatorname{ess\,sup}_{t \in [0,T]} \|\theta_t^*\|_\infty < \infty$ (ess sup denotes the essential supremum). Denote by $X^*$ the corresponding optimally controlled state process. Then, there exists an absolutely continuous co-state process $P^* : [0, T] \to \mathbb{R}^d$ such that the Hamilton's equations

$$\dot{X}_t^* = \nabla_p H(t, X_t^*, P_t^*, \theta_t^*), \quad X_0^* = x, \tag{31}$$

$$\dot{P}_t^* = -\nabla_x H(t, X_t^*, P_t^*, \theta_t^*), \quad P_T^* = \nabla\Phi(X_T^*), \tag{32}$$

are satisfied. Moreover, for each $t \in [0, T]$, we have the Hamiltonian maximization condition

$$H(t, X_t^*, P_t^*, \theta_t^*) \geq H(t, X_t^*, P_t^*, \theta) \quad \text{for all } \theta \in \Theta. \tag{33}$$

The PMP conditions can be naturally generalised to the Lie Group. Pontryagin's Maximum Principle (PMP) for Lie groups (Saccon et al. (2010)) provides the conditions that govern the flow of the state $x_t$ and its associated cotangent flow $\lambda$, describing the Hamiltonian equations that the optimal state-adjoint trajectory must satisfy. Specifically, the Hamiltonian equations are given by:

$$X_t^{-1} \dot{X}_t^* = \frac{\partial}{\partial p} H(t, X_t^*, P_t^*, \theta_t^*), \quad X_0^* = x,$$

$$\dot{\mu}_t^\theta = -\operatorname{ad}^*_{\frac{\partial H}{\partial \mu}} \mu_t^\theta - (\mathrm{d}L_{x_t^\theta})^* \frac{\partial}{\partial x} H^*, \quad \mu_T^\theta = (\mathrm{d}L_{x_T^\theta})^* \nabla_x \Phi(x_T^\theta). \tag{34}$$

The dual map $(\mathrm{d}L_g)^\star : T^\star_{gh}\mathrm{SO}(3) \to T^\star_h \mathrm{SO}(3)$ pulls back a cotangent vector at $gh$ to a cotangent vector at $h$. The coadjoint representation $\operatorname{ad}^*_X$ acts on $\mathfrak{so}(3)^*$ and is defined as:

$$\langle \operatorname{ad}^*_X \mu, Y \rangle = -\langle \mu, \operatorname{ad}_X Y \rangle, \tag{35}$$

where $\operatorname{ad}_X Y = [X, Y] = XY - YX$ for $\mu \in \mathfrak{so}(3)^*$ and $X, Y \in \mathfrak{so}(3)$. Additionally, for each $t \in [0, T]$, the Hamiltonian maximization condition is satisfied:

$$H(t, X_t^*, \mu_t^*, \theta_t^*) \geq H(t, X_t^*, \mu_t^*, \theta) \quad \text{for all } \theta \in \Theta. \tag{36}$$

## B.2 EXTENDED-METHOD OF SUCCESSIVE APPROXIMATIONS (E-MSA)

### B.2.1 METHOD OF SUCCESSIVE APPROXIMATIONS (MSA)

One numerical method for solving the Pontryagin Maximum Principle (PMP) is the Method of Successive Approximations (MSA) Chernousko & Lyubushin (1982), an iterative approach that alternates between propagation and optimization steps based on the PMP conditions. We first present the simplest form of MSA.

Consider the general state dynamics:

$$\dot{X}_t^* = f(t, X_t^*, \theta^*). \tag{37}$$

Given an initial guess $\theta^0 \in \mathcal{U}$ for the optimal control, for each iteration $k = 0, 1, 2, \ldots$, we first solve the state dynamics:

$$\dot{X}_t^{\theta^k} = f(t, X_t^{\theta^k}, \theta_t^k), \quad X_0^{\theta^k} = x, \tag{38}$$

to obtain $X^{\theta^k}$, followed by solving the co-state equation:

$$\dot{\mu}_t^{\theta^k} = -\nabla_x H(t, X_t^{\theta^k}, \mu_t^{\theta^k}, \theta_t^k), \quad \mu_T^{\theta^k} = -\nabla\Phi(X_T^{\theta^k}), \tag{39}$$

to determine $\mu^{\theta^k}$. Finally, the control is updated using the maximization condition:

$$\theta_t^{k+1} = \arg\max_{\theta \in \Theta} H(t, X_t^{\theta^k}, \mu_t^{\theta^k}, \theta), \tag{40}$$

for $t \in [0, T]$. This process is summarized in Algorithm 3.

---

**Algorithm 3** Basic MSA

---

1: Initialize: $\theta^0 \in \mathcal{U}$;
2: **for** $k = 0$ to #Iterations **do**
3:      Solve $\dot{X}_t^{\theta^k} = f(t, X_t^{\theta^k}, \theta_t^k), \quad X_0^{\theta^k} = x$;
4:      Solve $\dot{\mu}_t^{\theta^k} = -\nabla_x H(t, X_t^{\theta^k}, \mu_t^{\theta^k}, \theta_t^k), \quad \mu_T^{\theta^k} = -\nabla\Phi(X_T^{\theta^k})$;
5:      Set $\theta_t^{k+1} = \arg\max_{\theta \in \Theta} H(t, X_t^{\theta^k}, \mu_t^{\theta^k}, \theta)$ for each $t \in [0, T]$;
6: **end for**

---

### B.2.2 E-MSA

E-MSA introduces the augmented Hamiltonian

$$\tilde{H}(t, x, \mu, \theta, v, q) := H(t, x, \mu, \theta) - \frac{1}{2}\rho\|v - f(t, x, \theta)\|^2 - \frac{1}{2}\rho\|q + \nabla_x H(t, x, \mu, \theta)\|^2. \tag{41}$$

Then, they define the following set of alternative necessary conditions for optimality:

**Proposition 3 (Extended PMP)** Suppose that $\theta^*$ is an essentially bounded solution to the optimal control problem (2). Then, there exists an absolutely continuous co-state process $\mu^*$ such that the tuple $(X^*, \mu^*, \theta^*)$ satisfies the necessary conditions

$$\dot{X}_t^* = \nabla_\mu \tilde{H}(t, X_t^*, \mu_t^*, \theta_t^*, \dot{X}_t^*, \dot{\mu}_t^*), \quad X_0^* = x$$

$$\dot{P}_t^* = -\nabla_x \tilde{H}(t, X_t^*, \mu_t^*, \theta_t^*, \dot{X}_t^*, \dot{\mu}_t^*), \quad \mu_T^* = -\nabla_x \Phi(X_T^*)$$

$$\tilde{H}(t, X_t^*, \mu_t^*, \theta_t^*, \dot{X}_t^*, \dot{\mu}_t^*) \geq \tilde{H}(t, X_t^*, \mu_t^*, 0, \dot{X}_t^*, \dot{\mu}_t^*), \quad \theta \in \Theta, t \in [0, T] \tag{42}$$

The key contribution is that the control terms $\theta$ can be updated by iteration using the Extended-PMP as shown below. Meanwhile, it is proven that under this update rule, for each iteration, the target function $J(\theta)$ is non-decreasing.

---

**Algorithm 4** Extended MSA

---

1: Initialize: $\theta^0 \in \mathcal{U}$. Hyper-parameter: $\rho$;
2: **for** $k = 0$ to #Iterations **do**
3:      Solve $\dot{X}_t^{\theta^k} = f(t, X_t^{\theta^k}, \theta_t^k), \quad X_0^{\theta^k} = x$;
4:      Solve $\dot{P}_t^{\theta^k} = -\nabla_x H(t, X_t^{\theta^k}, P_t^{\theta^k}, \theta_t^k), \quad P_T^{\theta^k} = -\nabla\Phi(X_T^{\theta^k})$;
5:      Set $\theta_t^{k+1} = \arg\max_{\theta \in \Theta} \tilde{H}(t, X_t^{\theta^k}, P_t^{\theta^k}, \theta, \dot{X}_t^{\theta^k}, \dot{P}_t^{\theta^k})$ for each $t \in [0, T]$;
6: **end for**

---

## C PROOFS AND THEOREMS

### C.1 PROOF FOR PROPOSITION 1

#### C.1.1 PART 1

**Proposition 1.** *For Affine Gaussian Probability Path, the expectation of the running cost upper bounds the KL divergence between the prior joint distribution $p_1(x^p, x_1) = p_1(x^p|x_1)p_{data}(x_1)$ and*

*the joint distribution after guidance $p_1(x^\theta, x_1) = p_1(x^\theta|x_1)p_{data}(x_1)$, with $x_1 \sim p_{data}$, $x^p$ induced by prior conditional vector field $u_t(x|x_1)$ and $x^\theta$ sampled by applying control $\theta_t(x_1)$ on $u_t$.*

$$\mathbb{E}_{x_1 \sim p_{\text{data}}(x_1)} \left[ \frac{1}{2} \int_0^1 \|\theta_t(x_1)\|^2 \, \mathrm{d}t \right] \geq C \cdot \text{KL}(p_1(x^\theta, x_1)\|p_1(x^p, x_1)). \tag{43}$$

**Proof.** For Affine Gaussian Probability Paths, the conditional flow distribution of the prior vector field can be written as: $P_t^p(x|x_1) = N(\mu_t(x_1), \sigma_t(x_1)^2 I)$, the conditional vector field is written as:

$$\phi_t(x) = \mu_t(x_1) + \sigma_t(x_1)x \tag{44}$$

with its dynamics:

$$\dot{\phi}_t(x) = \dot{\mu}_t(x_1) + \dot{\sigma}_t(x_1)x, \tag{45}$$

or

$$u_t(x|x_1) = \dot{\mu}_t(x_1) + \frac{\dot{\sigma}_t}{\sigma_t}(x - \mu_t(x_1)).$$

Although control terms $\theta(x_1)$ are specifically derived for each ODE trajectory and are conditioned on the target point $x_1$, we can define a marginalized control term $\theta_t(x) = \int \theta(x_1)p(x_1|x) \, \mathrm{d}x_1$ to assemble aggregated control from all possible controls applied to each target point. Adding that to the marginal vector field can be thought of as effectively adding the target-conditioned controls into the conditional vector field, and then marginalizing:

$$u(x) + \theta_t(x) = \int (u_t(x|x_1) + \theta_t(x_1)) \, p(x_1|x) \, \mathrm{d}x_1. \tag{46}$$

Given the definition of the Gaussian path, we can derive that the additive control terms only alter the mean of the distribution by denoting it as $\theta_t(x_1)$ by noticing it is not proportional to $x$:

$$\dot{\phi}_t^\theta(x) = u_t(\phi(x)|x_1) + \theta_t(x_1) = \theta_t(x_1) + \dot{\mu}_t(x_1) + \dot{\sigma}_t(x_1)x. \tag{47}$$

The resulting pushing-forward prob is equivalently:

$$P_t^\theta(x) = \int P_t^\theta(x|x_1)q(x_1) \, \mathrm{d}x_1,$$

$$P_t^\theta(x|x_1) = N(\mu_t^\theta(x_1), \sigma_t(x_1)^2 I). \tag{48}$$

Usually, besides maximising the reward, we hope the new distribution is not too much away from the original distribution. One common constraint on the terminal state distribution is the Kullback-Leibler (KL) divergence.

Inspired by Variational Flow Matching (Eijkelboom et al. (2024)), we can the KL divergence between the joint distributions of the data point $x_1$ and the prior terminal point $x^p$, denoted as $p_1(x^p, x_1) = p_1(x^p|x_1)p_{\text{data}}(x_1)$, and that of the data point $x_1$ and the controlled terminal point $x^\theta$, denoted as $p_1(x^\theta, x_1)$: $\text{KL}(P_1^\theta(x, x_1)\|P_1^u(x, x_1))$.

The KL term can be simplified as:

$$\text{KL}(P_1^p(x, x_1)\|P_1^\theta(x, x_1)) = \iint P_1^p(x|x_1)q(x_1) \log \frac{P_1^p(x|x_1)p_{\text{data}}(x_1)}{P_1^\theta(x|x_1)p_{\text{data}}(x_1)} \, \mathrm{d}x \, \mathrm{d}x_1 \tag{49}$$

$$= \mathbb{E}_{x_1 \sim p_{\text{data}}(x_1)}[\text{KL}(P_1^p(x|x_1)\|P_1^\theta(x|x_1))].$$

Given $P_1^p(x|x_1)$ and $P_1^\theta(x|x_1)$ are two Gaussians with the same variance but different mean, and when we consider the constraint per sample, the constraint can be written as:

$$\frac{1}{2} \frac{\|\mu_1^\theta(x_1) - \mu_1^p(x_1)\|^2}{\sigma_1(x_1)^2}, \tag{50}$$

given:

$$\|\mu_1^\theta(x_1) - \mu_1^p(x_1)\|^2 = \left\| \int_0^1 \theta_t \, \mathrm{d}t \right\|^2 \leq \int_0^1 \|\theta_t\|^2 \, \mathrm{d}t. \tag{51}$$

Therefore, we have the following inequality:

$$\mathbb{E}_{x_1 \sim p_{\text{data}}(x_1)} \left[ \frac{1}{2} \int_0^1 \|\theta_t(x_1)\|^2 \, \mathrm{d}t \right] \geq \frac{1}{4\sigma_1(x_1)^2} \cdot \text{KL}(p_1(x^\theta, x_1)\|p_1(x^p, x_1)). \tag{52}$$

### C.1.2 PART 2

**Proposition 1. Part 2** *For square-shaped data $x$ with non-zero probability path, the expectation of the running cost, combined with the L1-distance between the prior sample $x_1^p$ and the corresponding guided sample $x_1^\theta$, upper bounds the KL divergence between the marginal distributions of the prior model $p_1^p$ and the guided model $p_1^\theta$:*

$$E_{x_1^p \sim p_1^p(x)} \left[ A \|x_1^p - x_1^\theta\| + B \int_0^1 \|\theta_t(x_1^p)\|^2 \, \mathrm{d}t \right] \geq \mathrm{KL}(p_1^p \parallel p_1^\theta). \tag{53}$$

**Proof.** Firstly, we can build a bound for the distance of terminal points of pre-trained model $\phi_t^p(x)$ and our controlled model $\phi_t^\theta(x)$ given the same starting point. Given $\frac{\mathrm{d}\phi_t^p(x)}{\mathrm{d}t} = f^p(x)$ with $f^p(\cdot)$ is the prior model and $\frac{\mathrm{d}\phi_t^\theta(x)}{\mathrm{d}t} = f^p(x) + \theta_t$, meanwhile as the $\theta_t$ are defined per sample, we can see them as a function of $x$. Given the ODEs governing $\phi_t^\theta(x)$ and $\phi_t^p(x)$ and use Lipschitz condition:

$$\|\phi_t^\theta(x) - \phi_t^p(x)\| \leq \int_0^1 \|f^p(\phi_t^\theta(x)) - f^p(\phi_t^p(x)) + \theta_t\| \, \mathrm{d}t \leq L \int_0^1 \|\phi_t^\theta(x) - \phi_t^p(x)\| \, \mathrm{d}t + \int_0^1 \|\theta_t\| \, \mathrm{d}t. \tag{54}$$

By Gronwall's inequality:

$$\|\phi_1^\theta(x) - \phi_1^p(x)\| \leq e^L \int_0^1 \|\theta_t\| \, \mathrm{d}t. \tag{55}$$

From the bound we build above, we can set

$$\phi_1^\theta(x) - \phi_1^p(x) = g(\theta_t, x) \quad \text{with} \quad g(\theta_t, x) \leq e^L \int_0^1 \|\theta_t\| \, \mathrm{d}t, \tag{56}$$

and then by definition:

$$\phi_{\theta,1}^{-1}(x) = \phi_{p,1}^{-1}(x - g). \tag{57}$$

Now we consider the push-forward functions and assume $p$ is non-zero for all $x$:

$$p_1^p(x) = p_0(\phi_{p,1}^{-1}(x)) \det \left[ \frac{\partial \phi_{p,1}^{-1}}{\partial x}(x) \right],$$

$$p_1^\theta(x) = p_0(\phi_{\theta,1}^{-1}(x)) \det \left[ \frac{\partial \phi_{\theta,1}^{-1}}{\partial x}(x) \right]. \tag{58}$$

Given the starting distribution is standard Gaussian, we can get the KL divergence as:

$$\mathrm{KL}(p_1^p \parallel p_1^\theta) = \int p_1(x) \left[ -\frac{1}{2} \|\phi_{p,1}^{-1}(x)\|^2 + \frac{1}{2} \|\phi_{\theta,1}^{-1}(x)\|^2 + \log \frac{\det \frac{\partial \phi_{p,1}^{-1}}{\partial x}(x)}{\det \frac{\partial \phi_{\theta,1}^{-1}}{\partial x}(x)} \right] \mathrm{d}x. \tag{59}$$

With the Mean Value Theorem, we can find a point $\tilde{x}$ between $x$ and $x - g$ so that

$$\phi_{p,1}^{-1}(x) - \phi_{p,1}^{-1}(x - g) = (\phi_{p,1}^{-1}(\tilde{x}))' g. \tag{60}$$

Assume the gradient is bounded so that $(\phi_{p,1}^{-1}(\tilde{x}))' \leq k$:

$$-\frac{1}{2} \|\phi_{p,1}^{-1}(x)\|^2 + \frac{1}{2} \|\phi_{\theta,1}^{-1}(x)\|^2 \leq k\|g\| + \frac{k^2}{2} \|g\|^2. \tag{61}$$

For the log term, with MVT again, there exists a point $\xi$ on the line segment between $x$ and $x - g$ such that:

$$h(x) - h(x - g) = \nabla h(\xi) \cdot g. \tag{62}$$

Applying this to $h(x) = \log \det f(x)$:

$$\log \det f(x) - \log \det f(x - g) = \nabla (\log \det f)(\xi) \cdot g. \tag{63}$$

Thus,

$$|\Delta(x)| = |\nabla (\log \det f)(\xi) \cdot g| \leq \|\nabla (\log \det f)(\xi)\| \|g\|. \tag{64}$$

To bound $\|\nabla(\log \det f)(\xi)\|$, the gradient of $\log \det f(x)$ with respect to $x$ is:

$$\nabla(\log \det f(x)) = (\det f(x))^{-1} \nabla(\det f(x)). \tag{65}$$

Given the derivative of the determinant of a matrix-valued function assuming $f(x)$ is square:

$$\nabla(\det f(x)) = \det f(x) \cdot \operatorname{Tr}\left(f'(x)^{-1} f''(x)\right). \tag{66}$$

Therefore:

$$\nabla(\log \det f(x)) = \operatorname{Tr}\left(f'(x)^{-1} f''(x)\right). \tag{67}$$

Assuming that:

- $\|f'(x)^{-1}\| \leq M$ for some constant $M$;
- $\|f''(x)\| \leq N$ for some constant $N$.

Then:

$$\|\nabla(\log \det f(x))\| \leq \|f'(x)^{-1}\| \cdot \|f''(x)\| \cdot n \leq MNn = K. \tag{68}$$

Therefore:

$$\log \frac{\det \frac{\partial \phi_{p,1}^{-1}}{\partial x}(x)}{\det \frac{\partial \phi_{\theta,1}^{-1}}{\partial x}(x)} \leq K\|g\|. \tag{69}$$

Overall, we can bound the KL divergence using the integration of $\theta_t$ and the integration of the square of $\theta_t$:

$$\mathrm{KL}(p_1^p \parallel p_1^\theta) \leq \mathbb{E}_{x_1 \sim p_1^p(x)}\left[(k+K)\|g\| + \frac{k^2}{2}\|g\|^2\right]$$

$$\leq \mathbb{E}_{x_1 \sim p_1^p(x)}\left[(k+K)e^L\|x_1 - x_1^\theta(x_1)\| + \frac{k^2}{2}e^L\int_0^1 \|\theta_t\|^2\,\mathrm{d}t\right]. \tag{70}$$

## C.2 PROOF FOR THEOREM 2

Since the Extended Method of Successive Approximations (E-MSA) is applied in this context, its convergence properties are directly inherited. In this section, we focus on how our update rule is derived from the E-MSA update rule, specifically under the framework of additive control terms and running cost.

Define Hamiltonian $H$ and Extended Hamiltonian $\tilde{H}$:

$$H(t, x, \mu, \theta) = \mu_t \cdot f(x, t) + \mu_t \cdot \theta_t - \frac{1}{2}\|\theta_t\|^2,$$

$$\tilde{H}(t, x, \mu, \theta, \dot{x}, \dot{\mu}) = \mu_t \cdot f(x, t) + \mu_t \cdot \theta_t^{k+1} - \frac{1}{2}\|\theta_t^{k+1}\|^2 - \frac{\gamma}{2}\|\theta_t^{k+1} - \theta_t^k\|^2. \tag{71}$$

Apply Extended MSA:

$$\dot{x}_t^k = \theta_t^k + f(x_t^k, t),$$

$$\dot{\mu}_t^k = -\nabla_x H(t, x^k, \mu^k, \theta^k) = -\nabla_x f(x_t^k, t)\mu_t^k,$$

$$\mu_1^k = \alpha \nabla_{x_1} \Phi(x_1^k),$$

$$\theta_t^{k+1} = \operatorname{argmax}_{\theta_t} \tilde{H}(t, x, \mu, \theta, \dot{x}, \dot{\mu}). \tag{72}$$

$\mu_t^k$ can be calculated in closed form as below, with $\mathcal{T}\exp$ is the time-order exponential:

$$\mu_t^k = \mathcal{T}\exp\left(\int_t^1 \nabla_x f(x_s^k, s)\,\mathrm{d}s\right) \cdot \alpha \nabla_{x_1} \Phi(X_1^k) \tag{73}$$

Thus, the update rule of $\theta_t$ is:

$$\theta_t^{k+1} = \frac{\gamma}{1+\gamma}\theta_t^k + \frac{\alpha}{1+\gamma}\mathcal{T}\exp\left(\int_t^1 \nabla_x f(x_s^k, s)\,\mathrm{d}s\right) \cdot \nabla_{x_1} \Phi(X_1^k). \tag{74}$$

Further, the time-order exponential term can be simplified as follows. Evans (1983) provides a method to calculate $p(t) = \nabla_{x_t} x(1)$ efficiently by defining the adjoint and using the following ODEs:

$$\dot{p}(t) = -\nabla_x f(x_t^k, t) p(t), \quad p(1) = \nabla_{x(1)} x(1) = I. \tag{75}$$

Sakurai & Napolitano (2020) provides an alternative viewpoint of the adjoint ODE, they show there would be a closed form solution:

$$\dot{p}(t) = A(t) p(t), \quad p(t) = \mathcal{T} \exp\left[-\int_t^1 A(s)\, \mathrm{d}s\right] p(1). \tag{76}$$

Thus, combining the closed form solution and the adjoint representation, we find that:

$$\nabla_{x_t} x(1) = \mathcal{T} \exp\left[\int_t^1 \nabla_x f(x_s^k, s)\, \mathrm{d}s\right]. \tag{77}$$

Given the linearity of the ODEs, multiply a constant to the terminal by change $p(t) = \nabla_{x_t} x(1)$ into $p(t) = \nabla_{x_t} x(1) \nabla_{x_1} \Phi(x_1)$, the conclusions would not change, we have:

$$\nabla_{x_t} x(1) \nabla_{x_1} \Phi(x_1) = \mathcal{T} \exp\left[\int_t^1 \nabla_x f(x_s^k, s)\, \mathrm{d}s\right] \nabla_{x_1} \Phi(x_1). \tag{78}$$

Therefore, our update rule is in fact:

$$\theta_t^{k+1} = \frac{\gamma}{1 + \gamma} \theta_t^k + \alpha \nabla_{x_t} \Phi(X_1^k). \tag{79}$$

The solutions can be naturally generalised to the space of $\mathbb{R}^N$ if we use $\mathrm{Tr}(A^\top B)$ to replace $A \cdot B$.

## C.3 DISCRETIZATION ERROR

Given the discretization method we are using is Euler step, we can show the error in terminal states due to Euler method with turning continous setting $\dot{x}_t^\theta = h_t(x_t^\theta, \theta_t)$ into discrete setting $x_{t+1} = x_t + h_t(x_t^\theta, \theta_t) \Delta t$ is bounded. Consider the first update of the states, define the local truncation error as:

$$\tau_k := x_{t+1}^* - x_t^* - h_t(x_t^\theta, \theta_t) \cdot \Delta t. \tag{80}$$

By definition:

$$\tau_k = \int_{t_k}^{t_{k+1}} h_s(x_s^\theta, \theta_s)\, \mathrm{d}s - h_t(x_t^\theta, \theta_t) \cdot \Delta t. \tag{81}$$

Given $\Delta t$ is small, with Taylor expansion:

$$h_s(x_s^\theta, \theta_s) = h_t(x_t^\theta, \theta_t) + \frac{\partial h}{\partial x}(x_s^\theta - x_t^\theta) + \frac{\partial h}{\partial t}(s - t) + \text{higher order terms.} \tag{82}$$

Similarly:

$$x_s^\theta - x_t^\theta = h_t(x_t^\theta, \theta_t)(s - t) + \text{higher order terms.} \tag{83}$$

We can then obtain

$$h_s(x_s^\theta, \theta_s) = h_t(x_t^\theta, \theta_t) + \left(\frac{\partial h_t}{\partial x} h_t + \frac{\partial h}{\partial t}\right)(s - t) + \text{higher order terms.} \tag{84}$$

Therefore:

$$\int_{t_k}^{t_{k+1}} h_s(x_s^\theta, \theta_s)\, \mathrm{d}s = \Delta t \cdot h_t(x_t^\theta, \theta_t) + \frac{\Delta t^2}{2}\left(\frac{\partial h_t}{\partial x} h_t + \frac{\partial h}{\partial t}\right) + \text{higher order terms,}$$

$$\tau_k = \frac{\Delta t^2}{2}\left(\frac{\partial h_t}{\partial x} h_t + \frac{\partial h}{\partial t}\right) + \text{higher order terms.} \tag{85}$$

Now we accumulate Local Errors to Global Error. Define error in $k^{\text{th}}$ step:

$$e_k = x^*(t_k) - x_k,$$

$$e_{k+1} = x^*(t_{k+1}) - x_{k+1} = (x^*(t_k) + \Delta t \cdot f(x^*(t_k), u^*(t_k), t_k) + \tau_k) - (x_k + \Delta t \cdot f(x_k, u_k^*, t_k)). \tag{86}$$

Simplifying:
$$e_{k+1} = e_k + \Delta t \left(f(x^*(t_k), u^*(t_k), t_k) - f(x_k, u_k^*, t_k)\right) + \tau_k. \tag{87}$$

Assuming $f$ is Lipschitz continuous in $x$ and $u$:
$$\|f(x^*(t_k), u^*(t_k), t_k) - f(x_k, u_k^*, t_k)\| \le L_f \|x^*(t_k) - x_k\| = L_f \|e_k\|. \tag{88}$$

Error Recurrence Inequality:
$$\|e_{k+1}\| \le \|e_k\| + \Delta t \cdot L_f \|e_k\| + \|\tau_k\| = (1 + \Delta t \cdot L_f)\|e_k\| + \|\tau_k\|. \tag{89}$$

Solving the Error Inequality given $\|\tau_k\| \le C(\Delta t)^2$, where $C$ is a constant depending on $f$ and its derivatives. We will solve the inequality:
$$\epsilon_{k+1} \le \alpha \epsilon_k + C(\Delta t)^2, \tag{90}$$

where $\epsilon_k = \|e_k\|$ and $\alpha = 1 + \Delta t \cdot L_f$. Thus:

$$\epsilon_{k+1} \le \alpha^{k+1} \epsilon_0 + C(\Delta t)^2 \sum_{j=0}^{k} \alpha^{k-j}. \tag{91}$$

Since $\epsilon_0 = 0$ (assuming $x_0$ is exact), the first term drops out. The sum becomes:

$$S_k = \sum_{j=0}^{k} \alpha^{k-j} = \frac{\alpha^{k+1} - 1}{\alpha - 1}. \tag{92}$$

For small $\Delta t$:
$$\alpha \le e^{\Delta t \cdot L_f},$$
$$\alpha^{k+1} \le e^{(k+1)\Delta t \cdot L_f} = e^{L_f t_{k+1}}. \tag{93}$$

Compute the Sum $S_k$:
$$S_k \le \frac{e^{L_f t_{k+1}} - 1}{e^{\Delta t \cdot L_f} - 1} \le \frac{e^{L_f t_{k+1}} - 1}{\Delta t \cdot L_f}. \tag{94}$$

Final Bound on $\epsilon_{k+1}$:

$$\epsilon_{k+1} \le C(\Delta t)^2 \cdot \frac{e^{L_f t_{k+1}} - 1}{\Delta t \cdot L_f} = \frac{C\Delta t(e^{L_f t_{k+1}} - 1)}{L_f}. \tag{95}$$

At the Final Time $t_f$:
$$\epsilon_N = \|x^*(t_f) - x_N\| \le \frac{C\Delta t(e^{L_f t_f} - 1)}{L_f}. \tag{96}$$

For multiple rounds of updates, the error would be accumulated, for the N+1$^{th}$ update, an additional local error is added to $\tau_k$:
$$\varepsilon_{k,N+1} = x_{t,N}^* - x_{t,N} + (h_t(x_{t,N}^*, \theta_N^{t,*}) - h_t(x_{t,N}, \theta_N^t)) \cdot \Delta t. \tag{97}$$

As the magnitude of this additional term is still $O(\Delta t^2)$, the order of magnitude of $\tau_k$ does not change, thus our conclusion does not change. The case in the SO(3) manifold should be similar, noticing the discretization also uses the Euler step.

## C.4 ASYNCHRONOUS SETTING AND VJP IN SO(3)

In practice, as described by equation 22, discretization techniques are employed to simulate the ordinary differential equations (ODEs) governing both the state trajectory $x_t$ and the corresponding cotangent vector $\mu_t$. Most existing methods, as well as the algorithm presented earlier, operate under a *synchronous* setting, where the number of time steps for the state trajectory $x_t$ matches the number of control terms $\theta_t$.

However, OC-Flow can be extended to an *asynchronous* framework by approximation to allow greater flexibility in update scheduling.

Rather than employing the standard update rule $x_{t+\Delta t} = x_t + f(t, x_t, \theta_t)\Delta t$, we subdivide the time interval $\Delta t$ into $N$ equally spaced subintervals, applying the control term $\theta_t$ only during the first subinterval. The update for $x_{t+\Delta t}$ is given by:

$$x_{t+\Delta t} = x_t + \frac{\Delta t}{N} f(x_t, \theta_t) + \frac{\Delta t}{N} f^p \left( x^\theta_{t+\frac{\Delta t}{N}} \right) + \cdots + \frac{\Delta t}{N} f^p \left( x^\theta_{\frac{(N-1)\Delta t}{N}} \right). \tag{98}$$

Moreover, for intermediate steps, we define:

$$x_{t+\frac{i\Delta t}{N}} = x_t + \frac{\Delta t}{N} f(x_t, \theta_t) + \sum_{l=1}^{i-1} \frac{\Delta t}{N} f^p \left( x^\theta_{t+\frac{l\Delta t}{N}} \right), \tag{99}$$

where $x_{t+\frac{i\Delta t}{N}}$ denotes the state at the $i$-th subinterval.

Recall that $f(x_t, \theta_t) = f^p(x_t) + \theta_t$, when we have $\frac{\Delta t}{N}$ is small enough, the update rule in Equation 10 can be approximated as a case in Equation 4 by considering $\nabla_\theta x_{t+\Delta t}$:

$$
\begin{aligned}
\nabla_\theta x_{t+\Delta t} =& \frac{\Delta t}{N} + \frac{\Delta t}{N} \nabla_\theta f^p \left( x^\theta_{t+\frac{\Delta t}{N}} \right) + \cdots + \frac{\Delta t}{N} \nabla_\theta f^p \left( x^\theta_{t+\frac{(N-1)\Delta t}{N}} \right) \\
=& \frac{\Delta t}{N} + \left( \frac{\Delta t}{N} \right)^2 \nabla_x f^p \left( x^\theta_{t+\frac{\Delta t}{N}} \right) + \cdots + \left( \frac{\Delta t}{N} \right)^N \nabla_x f^p \left( x^\theta_{t+\frac{(N-1)\Delta t}{N}} \right) \\
=& \frac{\Delta t}{N} + O \left( \left( \frac{\Delta t}{N} \right)^2 \right).
\end{aligned} \tag{100}
$$

Therefore, if we denote $x_t$ as the trajectory of the state variable $x$ over the time interval $[t, t + \Delta t]$, and $x^\theta_t$ as the trajectory when the control term $\theta_t$ is applied in the first subinterval, it can be reasonably approximated as:

$$x_{t+\Delta t} \approx x_t + \frac{\Delta t}{N} \sum_{l=1}^{i-1} f^p \left( x_{t+\frac{l\Delta t}{N}} \right) + \frac{\Delta t}{N} \theta_t. \tag{101}$$

As a result, for the asynchronous setting, the step 4 in Algorithm 1 should be modified as:

$$X^{\theta^k}_{t+\Delta t} = \left( \frac{1}{N} \sum_{l=1}^{i-1} f^p(x_{t+\frac{l\Delta t}{N}}) + \frac{1}{N} \theta^k \right) \Delta t. \tag{102}$$

For the case on the SO$(3)$ manifold, the asynchronous setting can be deployed using the Taylor expansion of the matrix exponential $\exp(A)$, and noting that when $\frac{\Delta t}{N}$ and $\Delta t$ are sufficiently small, the terms become commutative. We can derive the approximation as follows:

$$
\begin{aligned}
x_{t+\Delta t} =& x_t \exp \left( \frac{\Delta t}{N} f(x_t, \theta_t) \right) \exp \left( \frac{\Delta t}{N} f^p \left( x^\theta_{t+\frac{\Delta t}{N}} \right) \right) \cdots \exp \left( \frac{\Delta t}{N} f^p \left( x^\theta_{\frac{(N-1)\Delta t}{N}} \right) \right) \\
\approx& x_t \exp \left( \frac{\Delta t}{N} \sum_{l=1}^{i-1} f^p \left( x_{t+\frac{l\Delta t}{N}} \right) + \frac{\Delta t}{N} \theta_t \right) \\
\approx& x_t \exp \left( \Delta t \left( f^p_e(x_t) + \frac{1}{N} \theta_t \right) \right).
\end{aligned} \tag{103}
$$

The vector-Jacobian method can also be applied to compute the term $\frac{\partial f^p_t}{\partial x}(x^k_t E_j)$ in Algorithm 2:

$$\left\langle \tilde{\mu}_t, \frac{\partial f^p_t}{\partial x} x^k_t E_j \right\rangle = \mathrm{Tr} \left( \tilde{\mu}^T_t \frac{\partial f^p_t}{\partial x} (x^k_t E_j) \right). \tag{104}$$

## C.5 PROOF FOR PROPOSITION 3

The key inequality we used for the proof of Proposition 3 is following:

$$\langle h, v \rangle \leq \|h\| \|v\|, \qquad \|x\| = 1. \tag{105}$$

for $h, v \in \mathfrak{so}(3)$ and $x \in \mathrm{SO}(3)$ Before proving the proposition 3, we first show that the cotangent vector $\{\mu_t\}$ is also bounded.

**Lemma 2:** Assume all functions satisfy Lipschitz condition. Then, there exists a constant $K' > 0$ such that for any $\theta$,

$$\|\mu_t^\theta\| \le K',$$

for all $t \in [0, T]$.

**Proof.** Using necessary condition and setting $\tau := T - t$, $\tilde{\mu}_\tau^\theta := \mu_{T-\tau}^\theta$ we get

$$\dot{\tilde{\mu}}_\tau^\theta = \mathrm{ad}^*_{\frac{\partial H}{\partial \mu}} \tilde{\mu}_\tau^\theta + (\mathrm{d}L_x)^*(\nabla_x \langle \tilde{\mu}_\tau^\theta, f \rangle), \quad \tilde{\mu}_0^\theta = (\mathrm{d}L_{x_T^\theta})^* \nabla \Phi(x_T^\theta). \tag{106}$$

With Lipschitz condition, we have $\|\tilde{\mu}_0^\theta\| \le \|\nabla \Phi(x_T^\theta)\| \|x_T^\theta\| \le K$, $\|\nabla_x f(t, x_t^\theta, \theta_t)\| \le K$, and $\|ad^*_{\frac{\partial H}{\partial \mu}} \mu\| \le \|\frac{\partial H}{\partial \mu}\| \|\mu\| \le K \|\mu\|$. Hence,

$$\|\dot{\tilde{\mu}}_\tau^\theta\| \le K \|\tilde{\mu}_\tau^\theta\|, \tag{107}$$

and

$$\|\tilde{\mu}_\tau^\theta\| - \|\tilde{\mu}_0^\theta\| \le \|\tilde{\mu}_\tau^\theta - \tilde{\mu}_0^\theta\| \le \int_0^t \|\dot{\tilde{\mu}}_s^\theta\| \, \mathrm{d}s \le \int_0^t (K \|\tilde{\mu}_\tau^\theta\|) \, \mathrm{d}s, \tag{108}$$

and by Gronwall's inequality,

$$\|\tilde{\mu}_\tau^\theta\| \le \|\tilde{\mu}_0^\theta\| e^{KT} =: K'. \tag{109}$$

This proves the claim since it holds for any $\tau$.

Now given all related terms are bounded, we can prove Proposition 2.

**Proposition 3:** *Assume that the reward function, the prior model, and their derivatives satisfy Lipschitz continuity, bounded by a Lipschitz constant $L$. Then, there exists a constant $C > 0$ such that for any $\theta, \phi \in \mathfrak{so}(3)$, the following inequality holds:*

$$J(\theta) + \int_0^1 \Delta_{\phi,\theta} H(t) \, \mathrm{d}t - \|\phi_t - \theta_t\|^2 dt \le J(\phi), \tag{110}$$

*where $X^\theta$ and $P^\theta$ satisfy the PMP conditions in Equation 34, and $\Delta H_{\phi,\theta}$ denotes the change in the Hamiltonian, defined as:*

$$\Delta H_{\phi,\theta}(t) := H(t, x_t^\theta, \mu_t^\theta, \phi_t) - H(t, x_t^\theta, \mu_t^\theta, \theta_t). \tag{111}$$

**Proof.** Firstly, by the definition of Hamiltonian and PMP conditions, we always have:

$$I(x^\theta, \mu_t^\theta, \theta) := \int_0^T \langle \mu_t^\theta, f_t^\theta \rangle - H(t, x_t^\theta, \mu_t^\theta, \theta) - L(\theta) \, \mathrm{d}t \equiv 0. \tag{112}$$

Define $\delta\mu_t = \mu_t^\phi - \mu_t^\theta$ and $\delta f_t = f_t^\phi - f_t^\theta$, the difference in $I$ can be decomposed as:

$$
\begin{aligned}
0 &\equiv I(x^\phi, \mu_t^\phi, \phi) - I(x^\theta, \mu_t^\theta, \theta) \\
&= \int_0^T \langle \mu_t^\theta, \delta f_t \rangle + \langle \delta\mu_t, f_t^\theta \rangle + \langle \delta\mu_t, \delta f_t \rangle \, \mathrm{d}t \\
&\quad - \int_0^T H(t, x_t^\phi, \mu_t^\phi, \phi) - H(t, x_t^\theta, \mu_t^\theta, \theta) \, \mathrm{d}t \\
&\quad - \int_0^T (L(\phi_t) - L(\theta_t)) \, \mathrm{d}t.
\end{aligned}
\tag{113}
$$

Now define $U(t) = \int_0^t f_s \, \mathrm{d}s = \log(x_t x_0^{-1})$ and by integrating by parts:

$$\int_0^T \langle \mu_t^\theta, \delta f_t \rangle \, \mathrm{d}t = \langle \mu_t^\theta, \delta U_t \rangle |_0^T - \int_0^T \langle \dot{\mu}_t^\theta, \delta U_t \rangle \, \mathrm{d}t,$$

$$\int_0^T \langle \delta\mu_t, \delta f_t \rangle \, \mathrm{d}t = \langle \delta\mu_t, \delta U_t \rangle|_0^T - \int_0^T \langle \delta\dot\mu_t, \delta U_t \rangle \, \mathrm{d}t. \tag{114}$$

Combine the two terms above:

$$\int_0^T \langle \mu_t^\theta, \delta f_t \rangle + \langle \delta\mu_t, f_t^\theta \rangle \, \mathrm{d}t = \langle \mu_t^\theta, \delta U_t \rangle|_0^T + \int_0^T \langle \delta\mu_t, \frac{\partial H^\theta}{\partial \mu} \rangle + \left\langle \mu_t^\theta, \mathrm{ad}_{\frac{\partial H^\theta}{\partial \mu}} \delta f_t \right\rangle + \left\langle \frac{\partial H^\theta}{\partial x}, \delta f_t x_t^\theta \right\rangle \mathrm{d}t. \tag{115}$$

Similarly, we get:

$$\begin{aligned}
\int_0^T \langle \delta\mu_t, \delta f_t \rangle \, \mathrm{d}t &= \frac{1}{2} \int_0^T \langle \delta\mu_t, \delta f_t \rangle \, \mathrm{d}t + \frac{1}{2} \int_0^T \langle \delta\mu_t, \delta f_t \rangle \, \mathrm{d}t \\
&= \frac{1}{2} \langle \delta\mu_t, \delta U_t \rangle|_0^T - \frac{1}{2} \int_0^T \langle \delta\dot\mu_t, \delta U_t \rangle \, \mathrm{d}t + \frac{1}{2} \int_0^T \langle \delta\mu_t, \delta f_t \rangle \, \mathrm{d}t \\
&= \frac{1}{2} \langle \delta\mu_t, \delta U_t \rangle|_0^T + \frac{1}{2} \int_0^T \left\langle \mu_t^\phi, \mathrm{ad}_{\frac{\partial H^\phi}{\partial \mu}} \delta U_t \right\rangle - \left\langle \mu_t^\theta, \mathrm{ad}_{\frac{\partial H^\theta}{\partial \mu}} \delta U_t \right\rangle \mathrm{d}t \quad (116) \\
&\quad + \frac{1}{2} \int_0^T \left\langle (\mathrm{d}L_{x_t^\phi})^* \frac{\partial H^\phi}{\partial x} - (\mathrm{d}L_{x_t^\theta})^* \frac{\partial H^\theta}{\partial x}, \delta U_t \right\rangle \mathrm{d}t \\
&\quad + \frac{1}{2} \int_0^T \left\langle \delta\mu_t, \frac{\partial H^\phi}{\partial \mu} - \frac{\partial H^\theta}{\partial \mu} \right\rangle \mathrm{d}t.
\end{aligned}$$

With mean value theorem and $x, \mu$ are bounded by constant $L$, we can always find $x_t^\gamma$ between $x_t^\phi$ and $x_t^\theta$, $\mu_t^\gamma$ between $\mu_t^\phi$ and $\mu_t^\theta$, $\gamma$ between $\phi$ and $\theta$, so that :

$$\begin{aligned}
&\int_0^T \left\langle (\mathrm{d}L_{x_t^\phi})^* \frac{\partial}{\partial x} H(t, x_t^\phi, \mu_t^\phi, \phi) - (\mathrm{d}L_{x_t^\theta})^* \frac{\partial}{\partial x} H(t, x_t^\theta, \mu_t^\theta, \theta), \delta U_t \right\rangle \mathrm{d}t \\
&= \int_0^T \left\langle (\mathrm{d}L_{x_t^\theta})^* \frac{\partial}{\partial x} H(t, x_t^\theta, \mu_t^\theta, \phi) - (\mathrm{d}L_{x_t^\theta})^* \frac{\partial}{\partial x} H(t, x_t^\theta, \mu_t^\theta, \theta), \delta U_t \right\rangle \mathrm{d}t \\
&\quad + \int_0^T \left\langle \nabla_x((\mathrm{d}L_{x_t^\gamma})^*) x_t^\gamma \delta U_t \nabla_x H(t, x_t^\phi, \mu_t^\phi, \phi), \delta U_t \right\rangle \mathrm{d}t \\
&\quad + \int_0^T \left\langle (\mathrm{d}L_{x_t^\theta})^* \nabla_x^2 H(t, x_t^\gamma, \mu_t^\phi, \phi) x_t^\gamma \delta U_t, \delta U_t \right\rangle \mathrm{d}t \quad (117) \\
&\quad + \int_0^T \left\langle (\mathrm{d}L_{x_t^\theta})^* \nabla_\mu \nabla_x H(t, x_t^\theta, \mu_t^\gamma, \phi) \delta\mu_t, \delta U_t \right\rangle \mathrm{d}t \\
&\leq \int_0^T \left\langle (\mathrm{d}L_{x_t^\theta})^* \frac{\partial}{\partial x} H(t, x_t^\theta, \mu_t^\theta, \phi) - (\mathrm{d}L_{x_t^\theta})^* \frac{\partial}{\partial x} H(t, x_t^\theta, \mu_t^\theta, \theta), \delta U_t \right\rangle \mathrm{d}t \\
&\quad + C \int_0^T \|\delta U_t\|^2 + \|\delta\mu_t\| \|\delta U_t\| \, \mathrm{d}t.
\end{aligned}$$

Using the same method we get:

$$\begin{aligned}
\int_0^T \langle \delta\mu_t, \delta f_t \rangle \, \mathrm{d}t &\leq \frac{1}{2} \langle \delta\mu_t, \delta U_t \rangle|_0^T + \frac{1}{2} \int_0^T \left\langle \mu_t^\phi, \mathrm{ad}_{\frac{\partial H^\phi}{\partial \mu}} \delta U_t \right\rangle - \left\langle \mu_t^\theta, \mathrm{ad}_{\frac{\partial H^\theta}{\partial \mu}} \delta U_t \right\rangle \mathrm{d}t \\
&\quad + \frac{1}{2} \int_0^T \left\langle \frac{\partial}{\partial x} H(t, x_t^\theta, \mu_t^\theta, \phi) - \frac{\partial}{\partial x} H(t, x_t^\theta, \mu_t^\theta, \theta), \delta U_t x_t^\theta \right\rangle \mathrm{d}t \\
&\quad + \frac{1}{2} \int_0^T \left\langle \frac{\partial}{\partial \mu} H(t, x_t^\theta, \mu_t^\theta, \phi) - \frac{\partial}{\partial \mu} H(t, x_t^\theta, \mu_t^\theta, \theta), \delta\mu_t \right\rangle \mathrm{d}t \quad (118) \\
&\quad + C \int_0^T \|\delta U_t\|^2 + \|\delta\mu_t\| \|\delta U_t\| \, \mathrm{d}t.
\end{aligned}$$

With boundary conditions:

$$\left\langle \mu_t^\theta + \frac{1}{2}\delta\mu_t, \delta U_t \right\rangle |_0^T = \left\langle \mu_T^\theta + \frac{1}{2}\delta\mu_T, \delta U_T \right\rangle$$

$$= \left\langle (dL_{x_t^\theta})^* \nabla\Phi(x_T^\theta), \delta U_t \right\rangle + \frac{1}{2}\left\langle (dL_{x_t^\phi})^* \nabla\Phi(x_T^\phi) - (dL_{x_t^\theta})^* \nabla\Phi(x_T^\theta), \delta U_t \right\rangle \quad (119)$$

$$\leq \Phi(x_T^\phi) - \Phi(x_T^\theta) + K\|\delta U_T\|^2.$$

Using same method to $H(t, x_t^\phi, \mu_t^\phi, \phi) - H(t, x_t^\theta, \mu_t^\theta, \theta)$, we obtain:

$$\left[ \Phi(x_T^\theta) + \int_0^T L(\theta_t)\,dt \right] - \left[ \Phi(x_T^\phi) + \int_0^T L(\phi_t)\,dt \right]$$

$$\leq K\|\delta U_T\|^2 - \int_0^T \Delta H_{\phi,\theta}(t)\,dt + \frac{1}{2}\int_0^T \left\langle \mu_t^\phi, \mathrm{ad}_{\frac{\partial H^\phi}{\partial\mu}}\delta U_t \right\rangle - \left\langle \mu_t^\theta, \mathrm{ad}_{\frac{\partial H^\theta}{\partial\mu}}\delta U_t \right\rangle dt$$

$$+ \frac{1}{2}\int_0^T \left\langle \frac{\partial}{\partial x}H(t, x_t^\theta, \mu_t^\theta, \phi) - \frac{\partial}{\partial x}H(t, x_t^\theta, \mu_t^\theta, \theta), x_t^\theta\delta U_t \right\rangle dt \quad (120)$$

$$+ \frac{1}{2}\int_0^T \left\langle \frac{\partial}{\partial\mu}H(t, x_t^\theta, \mu_t^\theta, \phi) - \frac{\partial}{\partial\mu}H(t, x_t^\theta, \mu_t^\theta, \theta), \delta\mu_t \right\rangle dt$$

$$+ C\int_0^T \|\delta U_t\|^2 + \|\delta\mu_t\|\|\delta U_t\|\,dt.$$

By definition:

$$\delta U_t = \int_0^T f(t, x_T^\phi, \phi) - f(t, x_T^\theta, \theta)\,dt, \quad (121)$$

and so

$$\|\delta U_t\| \leq \int_0^t \|f(s, x_s^\phi, \phi) - f(s, x_s^\theta, \theta)\|\,ds$$

$$\leq \int_0^t \|f(s, x_s^\phi, \phi) - f(s, x_s^\theta, \phi)\|\,ds + \int_0^t \|f(s, x_s^\theta, \phi) - f(s, x_s^\theta, \theta)\|\,ds \quad (122)$$

$$\leq \int_0^T \|f(s, x_s^\theta, \phi) - f(s, x_s^\theta, \theta)\|\,ds + K\int_0^t \|\delta U_s\|\,ds.$$

By Gronwall's inequality:

$$\|\delta U_t\| \leq e^{KT}\int_0^T \|f(s, x_s^\theta, \phi) - f(s, x_s^\theta, \theta)\|\,ds. \quad (123)$$

To estimate $\delta\mu$, we use the same substitution as in Lemma 6 with $\tau = T - t$, we get:

$$\delta\tilde{\mu}_\tau = \delta\tilde{\mu}_0 + \int_0^\tau (dL_{\tilde{x}_s^\phi})^* \nabla_x H(s, \tilde{x}_s^\phi, \tilde{\mu}_s^\phi, \phi) - (dL_{\tilde{x}_s^\theta})^* \nabla_x H(s, \tilde{x}_s^\theta, \tilde{\mu}_s^\theta, \theta)\,ds$$

$$+ \int_0^\tau \left( \mathrm{ad}_{\frac{\partial H}{\partial\mu}}^* \tilde{\mu}_\tau^\phi - \mathrm{ad}_{\frac{\partial H}{\partial\mu}}^* \tilde{\mu}_\tau^\theta \right)ds. \quad (124)$$

Using Lemma 1 and Liptichitz conditions:

$$
\begin{aligned}
\|\delta\tilde{\mu}_\tau\| &\leq \|\delta\tilde{\mu}_0\| + \int_0^\tau \|(\mathrm{d}L_{\tilde{x}_s^\phi})^*\nabla_x H(s,\tilde{x}_s^\phi,\tilde{\mu}_s^\phi,\phi) - (\mathrm{d}L_{\tilde{x}_s^\theta})^*\nabla_x H(s,\tilde{x}_s^\theta,\tilde{\mu}_s^\theta,\theta)\|\,\mathrm{d}s \\
&\quad + \int_0^\tau \left\|\mathrm{ad}^*_{\frac{\partial H}{\partial\mu}}\tilde{\mu}_\tau^\phi - \mathrm{ad}^*_{\frac{\partial H}{\partial\mu}}\tilde{\mu}_\tau^\theta\right\|\,\mathrm{d}s \\
&\leq K\|\delta U_T\| + KK'\int_0^T \|\delta U_t\|\,\mathrm{d}t + K\int_0^\tau \|\delta\tilde{\mu}_s\|\,\mathrm{d}s \\
&\quad + \int_0^T \|(dL_{\tilde{x}_s^\theta})^*\nabla_x H(s,\tilde{x}_s^\theta,\tilde{\mu}_s^\theta,\phi) - (dL_{\tilde{x}_s^\theta})^*\nabla_x H(s,\tilde{x}_s^\theta,\tilde{\mu}_s^\theta,\theta)\|\,\mathrm{d}s \\
&\leq e^{KT}K\left(\|\delta U_T\| + K'\int_0^T \|\delta U_t\|\,\mathrm{d}t\right) \\
&\quad + e^{KT}K\int_0^T \|\nabla_x H(s,\tilde{x}_s^\theta,\tilde{\mu}_s^\theta,\phi) - \nabla_x H(s,\tilde{x}_s^\theta,\tilde{\mu}_s^\theta,\theta)\|\,\mathrm{d}s.
\end{aligned}
\tag{125}
$$

Using the bound of $U_t$, we obtain:

$$
\begin{aligned}
\|\delta\tilde{\mu}_\tau\| &\leq K''\left(\int_0^T \|f(s,x_s^\theta,\phi) - f(s,x_s^\theta,\theta)\|\,\mathrm{d}s\right) \\
&\quad + e^{KT}K\int_0^T \|\nabla_x H(s,\tilde{x}_s^\theta,\tilde{\mu}_s^\theta,\phi) - \nabla_x H(s,\tilde{x}_s^\theta,\tilde{\mu}_s^\theta,\theta)\|\,\mathrm{d}s.
\end{aligned}
\tag{126}
$$

Also we obtain:

$$
\begin{aligned}
\frac{1}{2}\int_0^T &\left\langle \mu_t^\phi, ad_{\frac{\partial H^\phi}{\partial\mu}}\delta U_t\right\rangle - \left\langle \mu_t^\theta, ad_{\frac{\partial H^\theta}{\partial\mu}}\delta U_t\right\rangle\,\mathrm{d}t \\
&= \frac{1}{2}\int_0^T \left\langle ad^*_{\frac{\partial H^\theta}{\partial\mu}}\mu_t^\phi - ad^*_{\frac{\partial H^\theta}{\partial\mu}}\mu_t^\theta, \delta U_t\right\rangle\,\mathrm{d}t \\
&\leq C\int_0^T \|\delta\mu_t\|\|\delta U_t\|\,\mathrm{d}t.
\end{aligned}
\tag{127}
$$

Finally we get:

$$
\begin{aligned}
J(\theta) - J(\phi) \leq & -\int_0^T \Delta H_{\phi,\theta}(t)\, \mathrm{d}t \\
& + \frac{1}{2} K'' \|\delta U_T\|^2 \\
& + K'' \int_0^T \left( \|\delta U_t\|^2 + \|\delta \mu_t\|^2 \right) \mathrm{d}t \\
& + \frac{1}{2} \int_0^T \|\delta \mu_t\| \|f(t, x_t^\theta, \phi_t) - f(t, x_t^\theta, \theta_t)\|\, \mathrm{d}t \\
& + \frac{1}{2} \int_0^T \|\delta U_t\| \|\nabla_x H(t, x_t^\theta, \mu_t^\theta, \phi_t) - \nabla_x H(t, x_t^\theta, \mu_t^\theta, \theta_t)\|\, \mathrm{d}t \\
\leq & -\int_0^T \Delta H_{\phi,\theta}(t)\, \mathrm{d}t \\
& + C \left( \int_0^T \|f(t, x_t^\theta, \phi_t) - f(t, x_t^\theta, \theta_t)\|\, \mathrm{d}t \right)^2 \\
& + C \left( \int_0^T \|\nabla_x H(t, x_t^\theta, \mu_t^\theta, \phi_t) - \nabla_x H(t, x_t^\theta, \mu_t^\theta, \theta_t)\|^2\, \mathrm{d}t \right)^2 \\
\leq & -\int_0^T \Delta H_{\phi,\theta}(t)\, \mathrm{d}t \\
& + C \int_0^T \|f(t, x_t^\theta, \phi_t) - f(t, x_t^\theta, \theta_t)\|^2\, \mathrm{d}t \\
& + C \int_0^T \|\nabla_x H(t, x_t^\theta, \mu_t^\theta, \phi_t) - \nabla_x H(t, x_t^\theta, \mu_t^\theta, \theta_t)\|^2\, \mathrm{d}t.
\end{aligned}
\tag{128}
$$

Therefore, given the form of the addictive control terms and the running cost, we can derive the final term of our claim:

$$
J(\theta) + \int_0^1 \Delta_{\phi,\theta} H(t)\, \mathrm{d}t - C\|\phi_t - \theta_t\|^2\, \mathrm{d}t \leq J(\phi).
\tag{129}
$$

### C.6  PROOF FOR PROPOSITION 4

Due to the similarity between the bound derived in Proposition 3 and the bound obtained from the E-MSA method, the proofs of Proposition 4 and Theorem 5 follow the same reasoning as outlined in Section 3.3 of Li et al. (2018). For the sake of completeness, we provide the full derivations here.

**Proposition 4:** *Let $X^\theta$ and $P^\theta$ satisfy the PMP conditions in Equation 34. If the update rule follows Equation 19, we define $\epsilon_k := \int_0^1 \Delta_{\theta^{k+1}, \theta^k} H(t)\, \mathrm{d}t$, and $\epsilon_k$ is bounded as:*

$$
\epsilon_k := \int_0^1 \Delta_{\theta^{k+1}, \theta^k} H(t)\, \mathrm{d}t, \qquad \lim_{k \to \infty} \epsilon_k = 0.
\tag{130}
$$

*Furthermore, when $\epsilon_k = 0$, we have $\theta = \theta^* := \arg\max_\theta J(\theta)$* To establish convergence, define

$$
\epsilon_k := \int_0^T \Delta H_{\theta^{k+1}, \theta^k}(t)\, \mathrm{d}t \geq 0.
\tag{131}
$$

**Proof.** By definition, if $\epsilon_k = 0$, then from the update rule which maximizes the Hamiltonian, we must have

$$
0 = -\epsilon_k \leq -\frac{\gamma}{2} \int_0^1 \|\theta^{k+1} - \theta^k\|^2\, \mathrm{d}t \leq 0,
\tag{132}
$$

and so

$$
\max_\theta \tilde{H}(x_t^{\theta^k}, \mu_t^{\theta^k}, \theta, \dot{x}_t^{\theta^k}, \dot{\mu}_t^{\theta^k}) = \tilde{H}(x_t^{\theta^k}, \mu_t^{\theta^k}, \theta_t^k, \dot{x}_t^{\theta^k}, \dot{\mu}_t^{\theta^k}).
\tag{133}
$$

Therefore, we always have the quantity $\epsilon_k \geq 0$ and it measures the distance from the optimal solution, and if it equals 0, then we reach the optimum.

## C.7 PROOF FOR THEOREM 5

**Theorem 5:** *Assume that the reward function, the prior model, and their derivatives satisfy Lipschitz continuity, bounded by a Lipschitz constant L. Let $\theta^0 \in \mathfrak{so}(3)$ be any initial measurable control with $J(\theta^0) < +\infty$. Suppose also that $\inf_{\theta \in \mathfrak{so}(3)} J(\theta) > -\infty$. If the update of $\theta$ satisfies equation 19, for sufficiently large $\gamma$, the following inequality holds:*

$$D\epsilon_k \leq J(\theta^{k+1}) - J(\theta^k) \tag{134}$$

*for some constant $D > 0$*

**Proof.** Using Proposition 3 we have

$$J(\theta^k) - J(\theta^{k+1}) \leq -\epsilon_k + C \int_0^T \|\theta^{k+1} - \theta^k\|^2 \, \mathrm{d}t. \tag{135}$$

From the Algorithm 2 maximizing step, we know that

$$H(t, X_t^{\theta^k}, P_t^{\theta^k}, \theta_t^k) \leq H(t, X_t^{\theta^k}, P_t^{\theta^k}, \theta_t^{k+1}) - \frac{\gamma}{2}\|\theta^{k+1} - \theta^k\|^2. \tag{136}$$

Hence, we have

$$J(\theta^k) - J(\theta^{k+1}) \leq -\left(1 - \frac{2C}{\gamma}\right)\epsilon_k. \tag{137}$$

Pick $\gamma > 2C$, then we shall have $J(\theta^k) - J(\theta^{k+1}) \leq -D\epsilon_k$ with $D = (1 - \frac{2C}{\gamma}) > 0$.

Moreover, we can rearrange and sum the above expression to get

$$\sum_{k=0}^M \epsilon_k \leq D^{-1}\left(J(\theta^{M+1}) - J(\theta^0)\right) \leq D^{-1}\left(\inf_{\theta \in \mathcal{U}} J(\theta) - J(\theta^0)\right), \tag{138}$$

and hence $\sum_{k=0}^\infty \epsilon_k < +\infty$, which implies $\epsilon_k \to 0$ and the algorithm converges to the optimum.

## D COMPUTATIONAL EFFICIENCY

Table 6: Comparison of runtime and memory complexity of different methods used in backprop-through guided-ODE in Euclidean and SO(3) manifold. For complexity, $N$ is the number of ODE steps, $n$ is the number of effective control terms with synchronized and in the range $[1, N]$ and $D^2$ is the complexity of computing 1-step gradient (VJP or Autograd), D depends on data and model size. $c$ is the deficiency introduced by L-BFGS optimizer.

| | Number Of Control Terms | Memory Complexity | Runtime Complexity | Convergence to Optimal | Generalization to SO(3) |
|---|---|---|---|---|---|
| OC-Flow | $n$ | $O(D^2)$ | $O(nD^2)$ | ✓ | ✓ |
| FlowGrad | $n$ | $O(D^2)$ | $O(nD^2)$ | ✗ | ✗ |
| D-Flow | $N$ | $O(ND^2)$ | $O(cND^2)$ | ✗ | ✗ |
| Red-Diff | N/A | $O(D)$ | $O(LND)$ | ✗ | ✗ |

In terms of memory complexity, both our implementation and FlowGrad utilize the vector-Jacobian approach, which allows solving differentiation through ODE-integral with the adjoint method, significantly reducing memory from $O(ND^2)$ (D-Flow) to $O(D^2)$. In comparison, D-Flow does not employ the adjoint method, significantly increasing the memory complexity to $O(ND^2)$, as detailed in Table 6.

In terms of runtime, both FlowGrad and OC-Flow are predominantly influenced by the number of control terms. Various strategies are implemented to reduce either the number of control terms (or

Table 7: Illustration how different methods decrease the runtime and memory complexity and enhance model capability

|  | Effective Timestep | Memory Complexity | Runtime Complexity | Generalization to SO(3) |
|---|---|---|---|---|
| OC-Flow | $n$ | $O(D^2)$ | $O(nD^2)$ | ✗ |
| w/o-asynchronous | $N$ | $O(D^2)$ | $O(ND^2)$ | ✗ |
| w/o-VJP (adjoint) | $N$ | $O(ND^2)$ | $O(ND^2)$ | ✗ |
| OC-Flow-SO(3) | $n$ | $O(D^2)$ | $O(nD^2)$ | ✓ |
| w/o-asynchronous | $N$ | $O(D^2)$ | $O(ND^2)$ | ✓ |
| w/o-VJP (adjoint) | $N$ | $O(ND^4)$ | $O(ND^4)$ | ✓ |

the effective time steps requiring back-propagation). Specifically, FlowGrad applies a straightening technique, and in oc-flow, this concept is extended to an asynchronous setting. Consequently, the runtime complexity is expressed as $O(nD^2)$ for both FlowGrad and OC-Flow. In contrast, D-Flow does not imply an asynchronous setting, necessitating N steps of Autograd, resulting in $O(ND^2)$ complexity. In addition, D-Flow is heavily relying on L-BFGS optimizer (Table 4), which also adds a significant increase in runtime due to e.g., uncontrollable additional iteration of linear search, which we estimate with a factor of constant $c$.

Table 7 provides a more direct comparison of the impact of the asynchronous setting and the vector-Jacobian product (VJP) method.

Compared to optimization-based algorithms, in some methods such as *Red-Diff* and DPS, the gradient-guidance can be approximated directly. These methods demonstrate significantly lower memory complexity and faster runtime. However, their capability is notably constrained. As reported in the Dflow paper, *Red-Diff* encounters difficulties in handling noise, even in simple linear inverse problems involving images.

When oc-flow is adapted to the SO(3) manifold, an additional calculation is introduced for each control term. Specifically, this involves computing the trace of the product of two $D \times D$ matrices, which adds a computational cost of $O(D^2)$. Despite this additional burden, the overall complexity remains in the order of $O(D^2)$. The VJP method is also applied for oc-flow on the SO(3) manifold, as shown in Equation 15, for the term $\langle \tilde{\mu}_t^k, \frac{\partial f_t^p}{\partial x} x_t^k E_i \rangle$, where $\frac{\partial f_t^p}{\partial x} x_t^k$ is computed using VJP. This additional complexity is justified by oc-flow's improved convergence to optimal solutions on the SO(3) manifold. Notably, the OC-Flow-SO3 involves computing full Jacobian which could induce a complexity of $O(D^4)$ if directing computing it. Our Jacobian-Vector Product derivation significantly reduces the complexity of OC-Flow-SO3 from $O(D^4)$ to $O(D^2)$, which enabled our efficient implementation.

Table 8: Memory Usage and Runtime on Text-guided Image (256*256) Manipulation. *Note: ODE steps = 100, optimization steps = 15.*

|  | FlowGrad | | DFlow | OC-Flow | |
|---|---|---|---|---|---|
| Fast Simulation | on | off | - | on | off |
| Peak GPU Mem (GB) | 5.2 | 5.2 | OOM | 5.4 | 5.4 |
| Runtime per Sample (s) | 114.7 | 206.2 | - | 115.7 | 216.7 |

In Table 8 we show memory usage and runtime on high-dimensional images, evaluated on a single A100 with 40G memory. We compare FlowGrad, DFlow, and OC-Flow on images. (Liu et al., 2023) proposed fast simulation through skipping some Euler steps if the relative velocity change is smaller than a threshold (1e-3 in (Liu et al., 2023)). We compare the 3 models with or without fast simulation. As for DFlow, we encountered Out-Of-Memory error even on 40G GPU. (Ben-Hamu et al., 2024) used gradient checkpoint to bypass the OOM error, however, due to the lack of code and implementation details of DFlow, we are unable to fully reproduce their implementation. For reference, DFlow takes 15 minutes per image (128*128) on a single 32GB V100 GPU, according to their paper. Generally speaking, FlowGrad and OC-Flow have similar memory usage and runtime except for DFlow.

Table 9: Memory Usage and Runtime Comparison of OC-Flow on Euclidean and SO(3).

|  | SO(3) | Euclidean |
|---|---|---|
| Peak GPU Mem (GB) | 1.6 | 1.2 |
| Runtime per Sample (s) | 296.6 | 188.2 |

For peptides 9, since FlowGrad and DFlow are not applicable in $SO(3)$, we only report results for OC-Flow using our proposed asynchronous algorithm for efficient simulation, evaluated on a single A100 with 40G memory. Comparing the results, rotation is computationally 0.5x more expensive than translation due to the additional cost introduced by Eq. 22. However, as shown in Table 7, the costs of rotation and translation remain within the same order of magnitude. It is important to note that the extra cost of rotation is justified, as the additional computations required by Eq. 22 are inherent to operations on the $SO(3)$ manifold. Moreover, as demonstrated in Table 5, generation in $SO(3)$ is crucial for the task.

Table 10: Runtime Comparison on Molecule. *Note: ODE steps = 50, SGD steps = 20, L-BFGS steps = 5 with inner steps = 5.*

|  | EquiFM | FlowGrad | DFlow | | OC-Flow | |
|---|---|---|---|---|---|---|
| Optimizer | N/A | SGD | SGD | L-BFGS | SGD | L-BFGS |
| Runtime per Sample (s) | 2.4 | 37.6 | 31.3 | 102.8 | 38.1 | 103.7 |

In Table 10, we compare FlowGrad, DFlow, and OC-Flow on molecule, evaluated on a single A100 with 40G memory. For reference, we also list EquiFM model as used in our method. To align with DFlow, we also adopt L-BFGS optimizer and find it crucial in performance yet slows down the efficiency.

# E  EXPERIMENTAL DETAILS

## E.1  TEXT-GUIDED IMAGE MANIPULATION

In our text-to-image generation experiment, we adopted the pipeline presented in Liu et al. (2023), utilizing the generative prior from Liu et al. (2022). We employed standard evaluation metrics: LPIPS and ID (face identity similarity) as introduced in Kim & Ye (2021) to assess the differences between the original image and the manipulated image. Additionally, the CLIP score was used to evaluate the alignment between the generated image and the provided text prompt.

To enforce consistency with the original image and inspired by Proposition 1, we introduced a constraint term to the terminal reward function to penalize significant deviations from the original image:

$$\Phi(x_1) = \lambda \text{CLIP}(x_1, T) - (1 - \eta)\|x_1 - x_1^p\|. \tag{139}$$

Here, the hyperparameter $\lambda$ was set to 0.7 across all experiments, and the Euler discretization step was set to $N = 100$ and the number of optimization iterations $M = 15$. As discussed in Theorem 2, increasing the learning rate $\eta$ results in greater emphasis on the terminal reward, leading to a higher CLIP score but lower LPIPS and ID scores. The weight decay is a function of $\gamma$, which is tuned to maximize $(1 - \frac{2C}{\gamma})\epsilon_\gamma^k$ for iteration k. In this experiment, due to the limitation of storage, we set $\gamma$ the same for all k. In our implementation, the learning rate $\eta$ was set to 2.5, and the weight decay $\beta$ was set to 0.995.

Baseline configurations were aligned with those reported in Liu et al. (2023), and the results presented in Table 2 reflect the same experimental conditions. For quantitative comparison, we used the CelebA dataset, randomly sampling 1,000 images, which were manipulated based on text guidance: `{old, sad, smiling, angry, curly hair}`.

### E.2 MOLECULE GENERATION

In our QM9 generation experiment, we mostly followed the conditional generation pipeline in Hoogeboom et al. (2022). An equivariant geometric GNN was trained for each property on half of the QM9 data as the classifier, which was then frozen during our training-free controlled generation. The EquiFM (Song et al., 2024) checkpoint, trained on the whole QM9 training data, was loaded as the generative prior. The test time properties were sampled from the whole training dataset, making it slightly different from the settings in Ben-Hamu et al. (2024). Therefore, we reimplemented the D-Flow algorithm with 5 optimizer steps and 5 inner steps each with a linear search using the L-BFGS optimizer. The results roughly matched those reported in the D-Flow paper with slightly worse MAEs as we included the whole training dataset for property sampling. Our proposed OC-Flow also used the same optimization hyperparameters and also almost the same running time as the D-Flow. For FlowGrad, we followed the suggestion in the original paper to use 20 SGD steps to update the learnable parts, which ran slightly faster than OC-Flow and D-Flow.

For all properties, MAE was used as the optimization target and $\gamma$ is the regularization coefficient such that $\gamma \int_0^1 \|\theta_t\|^2 \, \mathrm{d}t$ is the additional OC loss. For all optimization methods, we always used a fixed number of 50 Euler steps so $\theta$ can be indexed by discrete indices. As the integral is done with a step size of 1/50, any $\gamma$ is effectively $\tilde{\gamma} = 2\gamma/50 = 0.04$ when taking the derivative with respect to $\theta$ or $x$. Therefore, $\gamma = 10$ effectively corresponds to $\tilde{\gamma} = 0.4$, which is still a valid optimization scheme.

We noted the difference in the optimizer in these settings in order to be consistent with the D-Flow setup. We provide additional ablation studies on the effect of the optimizer, in which all guided generation approaches used the SGD optimizer with 20 iterations and a learning rate of 1, following the FlowGrad setup. The results are summarized in Table 11. It can be clearly demonstrated that D-Flow performance is significantly worse than OC-Flow and even FlowGrad, the latter of which is a special case of our OC-Flow. Indeed, we can safely conclude that the advantage of D-Flow came solely from its optimizer of using L-BFGS. Using the SGD optimizer caused it to perform even worse than the unconditional EquiFM baseline on the dipole moment $\mu$. On the other hand, OC-Flow achieved consistent improvements, using either the L-BFGS or the simple SGD optimizer, demonstrating our superior performance.

Table 11: Ablation on optimizer. MAE for guided generations on QM9 (lower is better).

| Property Unit | $\alpha$ Bohr³ | $\Delta\varepsilon$ meV | $\varepsilon_{\text{HOMO}}$ meV | $\varepsilon_{\text{LUMO}}$ meV | $\mu$ D | $c_v$ $\frac{\text{cal}}{\text{K·mol}}$ |
|---|---|---|---|---|---|---|
| OC-Flow(Ours) | **1.907** | **346** | **187** | **300** | **0.362** | **0.972** |
| D-Flow-SGD | 5.753 | 1241 | 571 | 1195 | 1.639 | 2.982 |
| FlowGrad | 2.484 | 517 | 273 | 429 | 0.542 | 1.270 |
| OC-Flow-LBFGS(Ours) | **1.383** | 367 | **183** | 342 | **0.314** | **0.819** |
| D-Flow-LBFGS (Ben-Hamu et al., 2024) | 1.566 | **355** | 205 | 346 | 0.330 | 0.893 |
| EquiFM | 8.969 | 1439 | 622 | 1438 | 1.593 | 6.873 |

### E.3 PEPTIDE DESIGN

In our peptide experiments, we adopted PepFlow (Li et al., 2024) as the baseline model, utilizing the pre-trained checkpoint provided in the original PepFlow paper. The test dataset split was also based on the one defined in the PepFlow framework. For hyperparameter tuning, we randomly selected 10 complexes from the dataset. After tuning, the full set of 162 complexes was used for guided sampling and evaluation to ensure a comprehensive performance assessment.

To enable flexible update scheduling, we adopted an asynchronous setting in OC-Flow. This design maintains the same ODE time steps as PepFlow while utilizing fewer control terms. Specifically, 200 time steps are used for ODE simulation, with 10 control terms, each controlling 20 time steps. As shown in Table 7, this approach reduces memory and runtime complexity without compromising the accuracy of the ODE simulation. Furthermore, for consistency and comparability across experiments,

we strictly controlled the initial noise during reruns to ensure consistency and comparability across experiments.

We used the pre-trained model as the initialization for our experiments, allowing us to build upon the pre-trained weights and achieve consistent performance improvements through hyperparameter adjustments. In OC-Flow(rot), we used $\alpha = 0.95$ and $\beta = 0.8$; in OC-Flow(trans), $\alpha = 0.9$ and $\beta = 1.2$; and in OC-Flow(both), $\alpha = 0.95$ and $\beta = 2.0$. For all methods, we followed the same hyperparameter choices outlined in the PepFlow paper to ensure fairness.

For evaluation, in addition to MadraX, the reward function used and optimized during training, we included several key metrics not used for training to comprehensively assess the physical validity and overall performance of the generated structures:

- **Stability**: Calculated using Rosetta over five independent runs, averaged to reduce high variance.
- **Affinity**: Measured by Rosetta to determine the binding energy of designed peptides, also averaged over five runs.
- **IMP (Improvement Percentage)**: The percentage of peptides with improved affinities (lower binding energies) compared to the native peptides, aligning with the definition used in PepFlow.
- **Diversity**: Calculated as the average of $1 - \text{TM-Score}$ among generated peptides, reflecting structural dissimilarities.
- **SSR (Secondary-Structure Similarity Ratio)**: The proportion of shared secondary structures between the designed peptide and the native peptide.
- **BSR (Binding Site Ratio)**: The overlapping ratio between the binding site of the generated peptide and the native binding site on the target protein.

Due to the time-intensive nature of Rosetta evaluations, we drew 10 samples per pocket for our experiments, in contrast to PepFlow's use of 64 samples. This approach enabled us to achieve reproducible and fair comparisons across methods without excessive computational costs. Moreover, by employing OC-Flow with guidance, we demonstrated that superior performance can be achieved with fewer samples while maintaining consistency in evaluation settings.

As shown in Table 5, we demonstrated the importance of optimization on the $SO(3)$ manifold for peptide design. To further evaluate the impact of using optimal control for updating rotations compared to standard gradient descent as in Euclidean space, we implemented Naive-$SO(3)$. In this implementation, the gradient of the terminal reward with respect to the control terms is computed directly and mapped to $\mathfrak{so}(3)$, and gradient descent is used to update the control terms.

$$\theta_t^{k+1} = \beta\theta_t^k + \eta[\nabla_{\theta_t^k}\Phi(x_1^{\theta^k})]_{\mathfrak{so}(3)},$$

$$\dot{x}_t^{\theta^{k+1}} = (f^p(x_t^{\theta^{k+1}}) + \theta_t^{k+1})x_t^{\theta^{k+1}}. \tag{140}$$

For the experimental setup, we conducted a comparative study using a randomly selected subset of 30 pockets. Each peptide was tested under identical conditions, with the primary difference being the parameterization and update method for rotations.

Table 12: Comparison of OC-Flow-SO3 and naive SO3 gradient descent

| | MadraX ↓ | RMSD ↓ | SSR % ↑ | BSR % ↑ | Stability ↓ | Affinity ↓ | Diversity ↑ | imp(%) ↑ |
|---|---|---|---|---|---|---|---|---|
| Ground-truth | -0.610 | - | - | - | -91.107 | -39.807 | - | - |
| PepFlow | -0.157 | **1.932** | 0.788 | **0.882** | -39.807 | -28.080 | 0.322 | 11.6 |
| Naive-SO(3) | 0.275 | 5.206 | 0.769 | 0.748 | 75.842 | -21.901 | **0.635** | 6.0 |
| OC-Flow-SO(3) | **-0.191** | 1.943 | **0.794** | 0.874 | **-50.947** | **-29.027** | 0.332 | **14.0** |

As shown in Table 12, the results indicate that updating rotations on $SO(3)$ using optimal control outperforms the naive method in terms of energy optimization and stability. The significant performance gap can be attributed to the accuracy loss during the projection of the gradient onto $\mathfrak{so}(3)$, which may disrupt the delicate dynamics of the $SO(3)$ space. Furthermore, due to the complexity of the $SO(3)$

manifold, gradient-based methods are more prone to becoming trapped in local optima, whereas the optimal control-based algorithm provides a pathway toward achieving the global optimum.

Our efforts to reproduce the PepFlow baseline involved extensive steps, including reaching out to the original authors to address the absence of certain scripts and obtaining partial instructions for the evaluation pipelines. In the original PepFlow paper, affinity and stability are reported as percentages. However, to facilitate more fine-grained comparisons, we opted to report absolute energy values instead. Additionally, to ensure fairness in comparison, we included IMP (Interaction Metric for Peptides) as an evaluation metric, aligning it with the affinity measure used in PepFlow. Furthermore, our experiments were conducted using the latest version of MadraX, ensuring that our results are both robust and reproducible. These updates provide a consistent and comprehensive framework for evaluating and comparing future methods in peptide design.

