# OpenReview forum: "Training Free Guided Flow-Matching with Optimal Control"
_ICLR.cc/2025/Conference — ICLR 2025 Poster_

### Official Review · Reviewer_zuF2 · 2024-11-01

**Soundness:** 3
**Presentation:** 3
**Contribution:** 2
**Rating:** 6
**Confidence:** 3

**Summary:**

The paper proposes a novel method based on optimal control to optimize generations obtained by ODE-based generative models (e.g. flow matching). The paper proposes algorithms for generative models in Euclidean space and in SO(3), generalizes previously existing approaches, and studies the convergence of the proposed method.

**Strengths:**

The paper tackles the problem of changing the generation process of ODE-based generative models in order to produce samples that maximize a certain reward, while staying close to the original ODE trajectory (through a regularization term). This problem is relevant in multiple domains where additional signals/information are available at inference time.

To the best of my knowledge, this is the first paper to formalize this guidance and control framework in SO(3), which is a group used by many methods in structural biology.

The approach generalizes existing methods that optimize the trajectory of ODE-samplers (D-Flow, GradFlow), and outperforms them in multiple benchmarks.

**Weaknesses:**

**Computational cost.** The method proposed in the paper, as well as its predecessors (D-Flow, GradFlow) all require optimizing the sampling process *for each sample* produced by the generative model. In other words, producing a single sample requires solving an optimization problem, for which computing the loss requires simulating the full ODE. This has a high cost in memory and computation time.

While GradFlow proposed a clever way of reducing the memory cost of this process, which is also adopted in this paper, this optimization process is inherently computationally expensive, significantly increasing the time cost of producing each sample. Previous work used some approaches to try to alleviate this (e.g. FlowGrad uses an adaptive solver to minimize the number of steps used during generation, by setting the step-size as a function of the estimated curvature of the flow at the current point), but simulation is still considerably slower than the baselines without this optimization process. While generation times are not discussed in this work, the D-Flow paper states that producing a single molecule takes around 3 minutes (they use 100 function evals for discretization of the ODE), and for images the time to generate a single output ranges from 4 to 15 minutes depending on the task. While the purpose of these works is not increasing generation efficiency, but generating better samples through guidance and optimization, they exacerbate the main limitation of diffusion models / flow models, which is their slow generation. In the paper I am unable to find generation time for the experiments. Given the method's nature, I think these should be reported and discussed.

I think related to this point, experiments tend to be on the smaller end. Celeba-HQ for images, molecule generation with up to 9 heavy atoms, and peptides, which are short proteins (less than 50 residues). I understand these methods are able to produce better samples given the external guidance, while other approaches are unable to leverage such information, which is quite valuable. Still, I think providing values for computational cost / generation time, and comparing against plain approaches (baselines that do not require tuning) would be good. I would expect the cost of producing one sample is between 10x-100x more than baselines that do not optimize the sampling process (since there are ~20 optimization steps, and some gradient computation too), but happy to be shown otherwise. This does not account for the fact that lower memory requirements by the baselines would allow producing more samples in parallel too.

**Questions:**

Proposition 1. What is the prior terminal point $x^p$? Is it $x^p_0$ or $x^p_1$ (that is, terminal as in time $t=1$ or $t=0$). If $t=0$ then the joint $p_1(x^p, x_1)$ is a delta distribution (since ODE is deterministic)? If $t=1$ then $x^p$ and $x_1$ are independent (since $x^p$ is generated from random noise independent of $x_1$)?

GradFlow could in principle be used for manifolds too? The control terms would live in the tangent space of the manifold at the current point? They do not propose this in GradFlow so this is not something to compare against, and would even be consider a novelty and addition to the paper. But this work got me wondering if there’s any obvious reason why this would fail?

I’m not sure I completely agree with the paper’s title “Training free…”. I understand training is the same, and that this method can be used for any pre-trained flow model. But it does require solving an optimization problem, albeit with few iterations (~20) but expensive ones. The difference is that this optimization happens at inference time.

---

> ### Author Response · Authors · 2024-11-22
> **Summary of concerns**
>
> Thank you for the great suggestion. **Following your suggestion, we now provide comprehensive discussion on time and memory complexity in appendix D**, including theoretical complexities for OCFlow, baselines and RedDiff (table 6) and actual runtime and memory usage of OC-Flow/Dflow/FlowGrad (table 8-10). You’re absolutely correct that the family of methods that optimizes ODE trajectory (Dflow/FlowGrad/ours) share the same limitation of inducing higher computation cost compared to posterior sampling (e.g. RED-Diff, DPS). To mitigate the cost, we introduce both an asynchronized update and adjoint method with vector-jacobian-product implementation on both Euclidean and SO3 to drastically reduce memory and runtime complexity on SO3 from O(D^4) to O(D^2) (table 7), while also being more efficient than Dflow (e.g., memory cost $O(ND^2)$ -> $O(D^2)$). This effectively enables us to sample 256x256 image under 3.5min whereas Dflow (without gradient ckpt) ran OOM on image task. The self-reported runtime for Dflow on 128x128 image is 15min, significantly higher than our efficient implementation of OC-Flow. Note that this is potentially because DFlow highly relies on the use of L-BFGS optimizer to achieve reasonable results (see our ablation on optimizer table 11), which adds a lot of runtime complexity due to e.g., uncontrollable line search complexity in L-GFBS. Our implementation of OCFlow-SO3 is also efficient (at the same order as Euclidean) and only 1.5x the runtime for Euclidean (e.g., 296s for SO3 and 188s for Euclidean in peptide design).
>
> We compare the cost of producing one sample in our method relative to direct sample without optimization, and the ratio is approximately 30x, as you predicted. We have updated discussion to talk about the tradeoff between computation cost and performance in comparison to “plain approach” like Red-Diff (line538). We believe that users could decide the tradeoff depending on applications and OC-Flow would be especially valuable for applications that require better quality and reward optimization with less latency requirement.
>
> **Q1: regarding proposition 1**
>
>
> We appreciate your valuable feedback regarding the ambiguity in notation. We have updated proposition1 with clearer notation definitions and **extended theoretical results to bounding marginal distributions**. The intuition here is to theoretically show that the running cost can control deviation from prior distribution measured by KL-divergence.
>
> For part 1 of proposition1, for a specific deterministic path of the prior model, its terminal point is defined as $x_1^p$ and the data used to train this path is $x_1$. Through the training process, the conditional distribution of the terminal point of this path given the training point is a delta distribution. However the joint distribution of $x^p$ and the training samples $x_1$, given by
> $p_{1}(x_p, x_1) = p_{1}(x^p|x_1)p_{data}(x_1)$
> is not a delta distribution. As the control terms are defined per sample, it is reasonable to consider them as being incorporated into the conditional path \(p(x|x_1)\) and then marginalized. This is because, as you noted, the ODE path is deterministic, and focusing on a specific path corresponds to dependence on a specific training sample \(x_1\), which defines the path.
>
> However, we acknowledge that it is probably more ideal to bound the marginal distribution, and therefore we now provide **additional result in proposition 1** which establish the **KL bound between marginal distributions of prior and guided model** as a function of running cost, under slightly more limiting assumptions (Appendix C.1.2).

---

> > ### Author Response · Authors · 2024-11-22
> >
> > **Q2: Can GradFlow be generalized to manifolds by projecting into tangent space?**
> >
> >
> > We appreciate the thoughtful consideration of additional possibilities in SO3. In principle, GradFlow can be applied to manifolds, as it is essentially a form of stochastic gradient descent (SGD), and SGD is well-defined for manifolds. However, a practical challenge arises because PyTorch, the commonly used framework, assumes a Euclidean space and does not enforce manifold constraints. Consequently, gradients must be manually mapped to the tangent space, which can result in accuracy loss.
> >
> > While this issue may be partially addressed by using rotation vectors in \( SO(3) \) instead of rotation matrices, the reward function often lacks restrictions, making it difficult to ensure that the gradient is correctly mapped to the tangent space. Furthermore, due to the inherent complexity of manifolds, SGD is unlikely to converge to a global optimum on manifolds. Although SGD in Euclidean spaces is theoretically proven to coincide with the optimal solution under specific conditions, achieving a global optimum remains challenging in most practical tasks.
> >
> > In contrast, methods based on optimal control are theoretically guaranteed to converge to the global optimum, providing a more robust and simple solution in manifold settings. **To support the claims above, we conducted an additional experiment**, which is detailed in Appendix E.3. In peptide design task, we compare the performance of our proposed OC-Flow-$\mathrm{SO}(3)$ algorithm with Naive-$\mathrm{SO}(3)$ solution we implemented, that maps the gradient to the tangent space followed by SGD on the control terms. The results demonstrate that our OC-Flow-$\mathrm{SO}(3)$ significantly outperforms the naive gradient mapping and SGD-based method.
> >
> > **Q3: regarding definitions of training-free**
> >
> > Thank you for the perspective, and we agree that if we consider “training” as broad concept of “optimization of any parameters”, the OC-Flow formulation might be interpreted as training the control terms. We refer “training” as the narrower definition in ML where train/finetune happens on the vector field model parameters, which will be amortized once trained and one could use the model for any new inputs without adaptation. A concurrent work “Adjoint matching” is an example of actually finetuning the entire flow model to amortize the reward optimization process. Our gradient-guided methods employ a different mindset that involves shifting the optimization effort to inference time. We believe both methods have pros and cons, and we’d like to highlight several benefit of inference-time approach:
> >
> > 1) **Our method is more flexible and accurately optimizes reward for every sample**, since unique control terms are solved per sample. It can be used to **guide any new reward functions** as long as its gradients are accessible, so we don’t need to worry about generalization to new samples or new reward. On the other hand, RLHF approach needs to amortize all optimization tasks for all possible samples with the model parameters, which may subject to all common generalization issues in ML such as under/overfitting, biased training data and OOD. It is also not able to adapt easily to new reward, so it’s possible that a retrain is needed everytime the reward changes.
> >
> > 2) In terms of scalability, the number of optimized control terms in our method is approximately $O(tD^2)$, where $t$ is the effective number of time steps and $D$ is the input dimension. In contrast, deep learning models typically have hundreds of millions or even billions of parameters, which is much harder and more expensive to optimize. We showed in appendix D that actual runtime of OC-Flow ranges from 30s to 299s, where we can generate 256x256 image under 3.5min (20 iterations, 100 ODE steps, table 8) on single A100GPU (in fact we can generate 8 images in parallel as memory cost is 5G). This is more efficient compared to approaches like RLHF or fine-tuning methods such as DRaFT [1], which require 8 hours on 16 TPUv4s (equivalent to 192 hours on a single A100 GPU). **Therefore, unless more than 2880\*8 images are needed for the same prompt, the total time required for fine-tuning exceeds that of our method.**
> >
> > [1] Clark, Kevin, et al. "Directly fine-tuning diffusion models on differentiable rewards." *arXiv preprint arXiv:2309.17400* (2023).

---

> > > ### Comment · Reviewer_zuF2 · 2024-11-26
> > > **Thanks for the time analysis**
> > >
> > > I appreciate the time analysis and the improvements over D-Flow. Regarding "Therefore, unless more than 2880*8 images are needed for the same prompt, the total time required for fine-tuning exceeds that of our method." Why "for the same prompt"? If I recall correctly, Direct Preference Optimization methods for diffusion models can be used in a conditional way? Meaning that the final fine-tuned model can be used for any prompt.

---

> > ### Comment · Reviewer_zuF2 · 2024-11-26
> > **Thank you for the response**
> >
> > I may be missing something here about Proposition 1. (The original one, I appreciate the extension to the marginals.) What is meant by "$x_1$ being the data used to train this path"? The path is originally defined by sampling noise $x_0$ and simulating the ODE, which results in $x^p$ as I understand? My question can be rephrased as what is $p_1(x^p|x_1)$?

---

> ### Author Response · Authors · 2024-11-26
> **Answers to followup questions**
>
> Thank you for your insightful follow-up questions, we are happy to provide our answers as follows:
>
> *Q1. Regarding DPO applied on a conditional reward function.*
>
> Thank you for the question. We’d like to note that the training time we used as reference here is from the DRaFT paper[1] which is finetuned for a fixed and relatively easy reward (Aesthetic Score, compressibility, etc.). It is probably better if we said “More than 2880*8 images are needed for a fixed reward function”.  As we mentioned in the rebuttal, amortizing the reward optimization effectively relies on the model’s capability to **generalize** to any new reward landscape conditioned on any prompt, which is by far still a challenging issue in RLHF. DPO(or in general RLHF) is known to be subject to overfitting and model collapse[2][3], and to our knowledge, there hasn’t been a really strong theoretical guarantee of how DPO can generalize across very different domains/conditions. From the RL point of view (if we view the prompts as part of the state), optimal policy convergence is often only guaranteed with sufficient coverage of all possible states (meaning it’s less data efficient), and it is usually globally greedy (instead of per-state optimal).
>
> Meanwhile, the actual time required for alignment with complex conditional reward functions could be much higher than 196 GPU hours. E.g., SD3[4] reported 2K iterations DPO finetuning on 8B models, effectively requiring 16T parameter updates. That said, we **do not want to claim that inference-time guidance is necessarily better than RLHF**. As noted in our rebuttal, we believe there are pros and cons for both, and the choice of the methodology should depend on the use cases and the scale of the problem.
>
> *Q2. Clarification on conditional probability $p_1(x|x1)$ in proposition 1*
>
> Thank you for the detailed question. We now realize where the confusion is from and have adjusted our description of Proposition 1 for better clarity.   In part 1 of Proposition 1, we analyze the effect of control terms $\theta(x_1)$ when they are applied to the conditional probability path $p_t(x^p|x_1)$ given a terminal data $x_1$ which is induced by a conditional vector field and is probabilistic, instead of a single deterministic ODE trajectory simulated with a trained marginalized vector field and fixed noise $x_0$. This allows us to analyze probabilities with a more tractable form than the complex push-forward equation and get tighter bounds.   Note that in a guided sampling setting, a different set of control terms are solved for each $x_1$ (or equivalently $x_0$ due to deterministic ODE), and therefore we note control terms as $\theta(x_1)$ to illustrate its dependency on $x_1$.  Overall, this analysis states the effect of $x_1$-dependent control terms on Gaussian paths conditioned on target training data $x_1$, to help us get some **intuitions** on the potential effect of $\theta(x_1)$ before marginalization. As we mentioned in the rebuttal, we acknowledge that bounding marginal distribution induced by controlling the marginal vector field is more ideal, and therefore we added part 2 which is directly derived from applying the push-forward equation.
>
> Another way of understanding part 1 is to imagine our controlled generation as a few-shot finetuning scenario, where the target data $x_1$ is given and fixed. Therefore, we still need to rely on conditional paths as in the training stage but can start off from a known prior generative model $p^p(x_t|x_1)$ to consider the distance to the “finetuned” model $p^\theta(x_t|x_1)$. Note that this is purely a thought experiment to provide an intuition of our OC-inspired framework, as we have demonstrated in later theorems that such expensive finetuning can be effectively replaced by our iterative OC-Flow algorithm in Alg 1.
>
> We hope the above discussion addresses your questions, and we're happy to continue the inspiring discussion if further questions arise.
>
> [1] Clark, Kevin, et al. "Directly fine-tuning diffusion models on differentiable rewards." arXiv preprint arXiv:2309.17400 (2023)
>
> [2]Zhu, Banghua, Michael Jordan, and Jiantao Jiao. "Iterative Data Smoothing: Mitigating Reward Overfitting and Overoptimization in RLHF." Forty-first International Conference on Machine Learning.
>
> [3] Xiong, Wei, et al. "Iterative preference learning from human feedback: Bridging theory and practice for rlhf under kl-constraint." Forty-first International Conference on Machine Learning. 2024.
>
> [4]Esser, Patrick, et al. "Scaling rectified flow transformers for high-resolution image synthesis." Forty-first International Conference on Machine Learning. 2024.

---

> > ### Comment · Reviewer_zuF2 · 2024-12-02
> > **Thanks for the clarifications**
> >
> > I thank the authors for the clarifications, and I agree that RLHF-type of methods would need to repeat the optimization for different rewards, while this method is directly applicable. I keep my acceptance score for the work.

---

> > > ### Author Response · Authors · 2024-12-02
> > > **Thank you for your insightful review and your support**
> > >
> > > Dear reviewer zuF2,
> > > We sincerely thank you for your valuable review and your kind support of our work. We enjoyed the insightful discussion with you during the rebuttal period. Thank you for your time.
> > > Best, authors

---

### Official Review · Reviewer_jxTp · 2024-11-04

**Soundness:** 2
**Presentation:** 1
**Contribution:** 2
**Rating:** 6
**Confidence:** 3

**Summary:**

This paper attempts to solve the problem of conditional generation using Flow-Matching models. In particular, they propose a unifying framework (OC-Flow) from which other approaches (such as D-Flow and FlowGrad) can be derived, and operates in Euclidean and SO(3) geometries. Subsequently, the authors provide extensive theoretical analysis of OC-flow to prove convergence and theoretical properties. Finally, the authors apply OC-Flow to text-guided image generation and peptide design tasks.

**Strengths:**

* The paper has a strong theoretical grounding, is well placed within the existing literature and goes to extensive efforts to prove theoretical properties of the proposed methodology. Additionally, the paper provides significant context and background, referencing existing works in Flow-Matching models. Finally, the authors provide a ton of detailed proofs in the appendix, and theoretical analysis in the main body of the text.
* The paper provides a legitimate contribution to formalizing and extending existing guided-flow matching techniques to complex geometries (such as SO(3))
* The authors compare the proposed methodology to similar existing methods to demonstrate competitive performance
* Table 1 gives a good comparison to understand the contribution of OC-flow vs D-Flow and Flow-Grad

**Weaknesses:**

__Theoretical Concerns__
* The main concern with the theoretical aspects of the paper is that the authors perform all the theoretical analysis in the continuous regime, but then implement the practical algorithms in a discrete regime. They even mention this limitation in the conclusion (“we also note that our practical algorithm…”). This presents a potentially significant hole in the paper, as the objects being proved, and the objects being validated empirically are not the same, and thus the theoretical content in the paper is not necessarily applicable to the experimental results (as noted by the author). To address this, we recommend that the authors 1) provide theoretical analysis of the discrete regime of the algorithm, 2) discuss in detail how the continuous approximates or bounds the behavior of the discrete implementation, or 3) implement a continuous version of the algorithm. These additions would help bridge the gap between theory and practice.

__Experimental Concerns__
* There is a significant lack of information regarding the experiments and reproducibility. There are no details as to how these models are trained, what the architectures are, the implementation, etc. This would need to be addressed before the paper could be accepted. To address this, we recommend that the authors give detailed descriptions of the model architectures and hyperparameters, training procedures and optimization details, data preprocessing steps, computational resources used, and code availability or plans for release. This would greatly improve the reproducibility of the work.
* Similarly, no error bars/confidence intervals are reported on any of the experiments. In fact, it is unclear whether their model is better than existing baselines (i.e. in table 5, they claim 0.795 is better than 0.793, but no CI, similarly in tables 2, 3 and 4, they report outperforming existing methods but do not report CI.). Consequently, it is not possible to assess their claims that they outperform existing models, given the lack of confidence intervals and the close proximity of the performance values. To address this, we recommend that the authors 1) run multiple trials and report mean and standard deviation for all metrics, 2) perform appropriate statistical significance tests (e.g. t-tests) when comparing to baselines and 3) include error bars or confidence intervals in all tables and figures.

* Finally, one of the major claims in the paper is that OC-flow can optimize in euclidean and SO(3) space, and that optimization in SO(3) provides benefits in tasks such as protein design. However, the authors do now present results split into Euclidean and SO(3) algorithms. It is unclear how/if the extension to SO(3) is even beneficial, and additional experimental details/results are needed to validate this claim as well. This is brought up by "our OC-Flow method, fully optimized in both Euclidean and SO(3) space" on page 10.

__Presentation Concerns__
* The paper has some issues with the presentation that make it quite difficult to asses which parts are novel contributions, and which parts are existing works that are being used. The authors/paper would benefit greatly from having a clear vision of what they are proposing and why, and subsequently moving large portions of the detailed proofs to the appendix to not muddy understanding with unnecessary detours. For example, what are co-state flow and E-MSA, and why do we care about these constructs? How do these constructs factor into the actual problem of performing guided matching flow generation? Are they purely used for proving convergence analysis? And if so, then it should be framed/explained as such. In fact, the structure of the paper would greatly benefit from having a section which clearly describes the proposed methodology in terms of implementation, and a separate section for the theoretical analysis of the proposed method, since the current structure makes it very difficult to separate the method as-such, from the additional theoretical concepts only necessary for proving convergence.
  * To address these issues, we recommend that the authors clearly delineate novel contributions from existing work, as well as practical details from theoretical proofs. Adding sections such as "contributions", "proposed methodology and implementation", and "theoretical results" would greatly improve the structure of the paper.
  * Additionally, we recommend that the authors provide clearer explanations of key concepts such as co-state flow and E-MSA, as well as highlighting their importance to the key contributions in the paper.
  * Finally, by restructuring the paper to highlight the novel aspects, and moving detailed proofs to the appendix, the overall quality and clarity of the paper would be much improved.
* Furthermore, several significant objects/theorems are introduced with very little explanation. For example, co-state variables are introduced as “shadow prices representing the sensitivity of the optimal value function to changes in the state variables”. But what are shadow prices? How does this analogy help when there is little thought/exposition given to the co-state/Hamiltonian introduced by the PMP? I would like to see a much more principled approach to writing, where each theorem introduced is clearly placed within the larger context of the work, and has a clear purpose in support of theoretical results.
* Similarly, the lack of figures greatly hinders understanding. Additionally, figure 1 does not clearly articulate what it is presenting, and what the various sub-figures and equations represent.

__Contribution Concerns__:
* First of all, due to the presentation it is not clear what the contributions of the paper are, and what is preexisting work being leveraged for proving theoretical properties of the method. However, my understanding is that there are two main contributions of the paper: 1) they formulate conditional generation using flow-matching models as a control problem in equation 2, and 2) given that formulation, the authors demonstrate that OC-Flow is a generalization of D-Flow and Flow-Grad that can be optimized in SO(3), as well as proving various convergence properties.
* In light of this understanding, it seems like the contributions of the paper are limited in scope. First of all, equation 2 seems to be fairly trivial extension of existing Flow-Matching/Continuous Normalizing Flow formulations (see Fjelde et al (2024)). Furthermore, given the lack of definitive experimental results, it is unclear whether this extension provides tangible benefit over existing methods, especially when considering the additional complexity. One of the major proposed benefits of the method is optimization in SO(3), but no ablation studies are given to demonstrate that SO(3) provides additional benefits over simple euclidean optimization.
* Additionally, the experimental reproducibility of the paper is quite poor, with no experimental parameters given, and no experimental source code provided.
*Finally, the scope of the contribution is somewhat niche. In particular, this paper focuses on classifier-guided generation using flow-matching models in SO(3). While useful for certain problems, it likely does not have wide-reaching implications outside of a few target applications.


__Citations__
* Fjelde, T., Mathieu, E., & Dutordoir, V.. (2024). An Introduction to Flow Matching.

**Questions:**

* I would like to see a comparison of the runtime of OC-flow vs the other methods. While Table 1 suggests that the memory consumption of OC-flow is lower than D-flow and on par with Flow-Grad, I would be concerned that the additional complexity of solving in SO(3) adds significant computational costs.
* I would like to have a better understanding of what parts of the paper are core to the methodology (i.e. actually implementing OC-Flow), versus what parts of the paper are necessary for proving convergence. I would then like to see separate sections/subsections for the proposal, and the subsequent analysis.
* I would like to see a clearer presentation of a conventional flow-matching model, and how the proposed method extends this standard formulation, ideally in the form of before/after equations to get a clear and unambiguous idea of the elements being added/proposed.

---

> ### Author Response · Authors · 2024-11-22
> **Summary of concerns**
>
> We sincerely thank you for your detailed and comprehensive feedback, and we’re happy to provide further clarifications and results to address all your comments:
>
> **1. Significance of our contribution**
>
>
> We’d like to first elaborate on the scope and significance of our contributions to address an important potential misunderstanding.
>
> - Our work is the first one that establishes a **general and formally defined optimal control framework** for guiding pre-trained flow-matching models and proposes solutions that optimize the **true OC objective** (eq2) that comprise both the reward term and the **running cost** $\int_\|\theta_t\|^2$. We provide novel, theoretically grounded, and empirically **effective algorithms for both Euclidean and SO3**, instead of merely an extension to SO3. We note that running cost is crucial in controlling how much the guided distribution diverges from the pre-trained CFM distribution. Prior to our work, no algorithms handled the running cost properly. Our flexible framework enables **tunable tradeoffs between reward maximization and faithfulness to the prior distribution**which previous methods cannot offer as they ignore running costs and only focus on naively back-propagating reward gradients with ad-hoc-ly chosen fixed parameters.**Solving the OC problem is nontrivial**, as it involves optimizing objectives containing integrals and complex dynamics that naive SGD cannot address. Making sure the algorithm **converges is essential** to guarantee the best tradeoff between reward optimization and closeness to prior, and a non-converging naive SGD solution could lead to complete failure (e.g., table 12). Although we show that Dflow and FlowGrad are one special case of OC-Flow in Euclidean space, the OC perspectives were not offered in their original papers, as they primarily focus on naive reward gradient backpropagation. We believe the **perspectives and technical depth of OC-Flow exceed far beyond the scope of Dflow and FlowGrad**, therefore categorizing it as an “extension” or “generalization” is improper.
>
> - **Our focus and scope cover both Euclidean and SO3**, instead of only SO3. Our results on image and QM9 are purely in Euclidean data, and our method **achieves top performance uniformly across both manifolds**, unlocking a wide range of real-world applications.
>
> - **We respectfully disagree that our contribution on SO3 is trivial.** Firstly, we now provide **comprehensive ablation study** to **demonstrate the contribution** of OC-Flow-SO3 in peptide design. We also show how naive solutions of applying SGD on SO3 by projection to tangent space will fail (see table 5 and 11). Secondly, **solving dynamics on SO3 is significantly harder than on Euclidean**, and we provide not only novel convergence bounds on SO3 but also efficient algorithms that **significantly speed up the computation** (see general response and runtime discussion below) with VJP (section 4.3 and C4). We believe our result not only enables guiding rotation generation in protein but can also be significant to **OC community** and may benefit robotics/control as well.
>
> - Finally, we clarify that **OC-Flow’s focus is not on training CFM or proposing new FM models**, but rather on guiding the sampling process of FM with OC to achieve conditional/constrained generation, which we state clearly in title/abstract. Our method is an inference time procedure that deviates significantly from normal forward ODE sampling and not a “trivial extension” of FM. We reviewed suggested Fjelde’s work, and **it talks about the general CFM formulation and training, which is orthogonal to the problem we’re studying.** We also cannot find any formula in Fjelde et al., that is close to eq2 in our optimal control framework.
>
> - Regarding the question “I would like to see a clearer presentation of a conventional FM,..., ideally in the form of before/after equations to get an idea of the elements being added/proposed.”:
> We now improved our figure 1 and its caption to better demonstrate the OC-Flow procedure. In the high-level, the state flow at iter=0 is the original ODE trajectory of prior model (the typical CFM inference), through **iterative update of costate and state trajectory** we achieve the final guided sample. Another way of telling the difference for before/after is by contrasting eq1 (prior model’s dynamic), with eq2 (OC dynamic, more defined in eq4/eq8), where vector is altered by control term $\theta$ and follows $\dot{x}_t^\theta = h_t(x_t^\theta, \theta_t)$ and a complex optimization loss on trajectory terminal reward and running cost is applied, that requires iteratively update control and trajectory to solve optimal control.

---

> ### Author Response · Authors · 2024-11-22
>
> 2. **Experimental significance and reproducibility**
>
>
> - Following your suggestion, additional experimental details are now included into the Experiment section and Appendix E, where we have reported all implementation details necessary to reproduce the results. Additionally, the code is being prepared and will be made publicly once accepted, and we've provided an anonymous link under overall comment for you to access our code for inspection. We’d like to clarify that **we do not train any new FM models** and the only place training is required is the reward function in QM9. To make sure fair and reproducible comparison, **all of the pretrained FM models we used are open-sourced publicly available models (unlike in Dflow).** We also control the initial noise and optimization parameters to be the same across methods.
>
> - **Statistical significance of the result:** our results are **averaged over a large number of random samples (~10³)** and therefore have high statistical power. E.g., in text-to-image experiment, 1000 images were sampled for each of the 5 text prompts, resulting in a total of 5000 samples. For QM9, 1000 samples were drawn per property and per model, and for the peptide design experiment, 1620 samples were generated (162 pockets, each sampled 10 times). Given the large sample sizes, we believe the mean statistic itself is significant enough as the standard error and confidence interval is small. Furthermore, **our presentation follows the same format as in FlowGrad and DFlow papers**, where CIs were not required as the sample size is very large.
>
> - In the experiment of peptide generation, the goal is to **show effective of guided FM method** on optimizing MadraX energy (which **is reward function used to guide FM**), while staying close to pepflow’s prior distribution. Therefore, metrics like SSR/BSR/RMSD are provided to show OC-Flow samples’ characteristics are **similar to PepFlow samples’** and not necessarily need to be better. Thus, we do not claim that 0.793 is better than 0.795, but rather showing that they remain similar after guidance.
>
> We’d like to also clarify that our focus is on both Euclidean and SO3, instead of only SO3. Our results on image and QM9 are in Euclidean, and our method achieve top performance uniformly across both manifolds.
>
> 3. **SO3 ablation**
>
>
> - Thank you for your suggestion on presenting the performance of our algorithm in $\mathrm{SO}(3)$. We now include additional ablation study in Table 5 and appendix E.3. We show that OC-Flow on SE(3), i.e. **combination of Euclidean (translation) and SO3 (rotation), achieved best performance**, and applying OC-Flow to single modality (SO3 or Euclidean) alone can also improve Pepflow as well. As shown in Table 5, incorporating $\mathrm{SO}(3)$ optimization doubles the increase of the Madrax score from 0.34 to 0.68 compared to only optimizing translation (Euclidean).
>
> - Additionally with table 12, we provide ablation study to show the **necessity of OC-Flow-SO3** on guiding flow matching on rotation (which is a SO3 manifold), where naively applying gradient update with projection to tangent space of SO3 fails to optimize and even lead to worse samples than unconditional generation.
>
> 4. **Runtime and scalability**
>
>
> Thank you for the suggestion. **We now provide comprehensive discussion on time and memory complexity in appendix D**, including theoretical complexities for OCFlow and baselines (table 1, table 6) and **actual runtime and memory usage** of OC-Flow/Dflow/FlowGrad (table 8-10). A key contribution we offer is the practical and efficient implementation of OC-Flow in both Euclidean and SO3, through introduction of Vector-Jacobian-Product formulation (eq15 for SO3, 3.2.1 for EU) and asynchronized update (3.2.2), which effectively reduce complexity from O(D^4) to O(D^2). We detailed the effectiveness of these scalability efforts in table 7.
>
> Our efficient SO3 implementation mitigates the complexity of the algorithm, which is shown to be in the same order as the Euclidean version (O(D^2)). Our runtime profile in peptide design (table 9) proves that sampling of rotation (so3) only **cost 1.5x of the time that of sampling translation (eu), and does not add “significant computational cost”**.
>
> Our method samples 256x256 **image under 3.5min**, whereas Dflow ran OOM on the same image task. The self-reported runtime for Dflow on 128x128 image is **15min**, significantly higher than our efficient implementation of OC-Flow.

---

> > ### Author Response · Authors · 2024-11-22
> >
> > 5. **Discretization bound**
> >
> >
> > We’d like to mention that most real-world SDE/ODE problems require discretization in practice, and while many existing theoretical works rely on continuous assumptions, they remain effective when implemented with discretization. **Nevertheless, we now provide additional theoretical analysis of the discretization gap, which is now included in Appendix C.3.** A key observation is that the discrete version of our algorithm corresponds to the Euler method for a continuous ODE, a well-studied approach for which bounds on the discretization gap are well established. Specifically, since our system is an ODE, it is possible to bound the discretization error for each step under the assumption of a sufficiently small step size, so we can also bound the terminal discretization error, representing the difference between the terminal points of the discrete and continuous trajectories.
> >
> > We also prove that the accumulated error after multi-round optimizations is of the same order as the terminal discretization error. Therefore, we can conclude that in Euclidean space, the discretization error is of the order $O(\Delta t)$ and becomes negligible with sufficiently dense steps. In practice, we use 100 time steps for text2img experiment, 50 timesteps for QM9 and 200 time steps for Peptide design, which makes the assumption valid. For the $\mathrm{SO}(3)$ case, since the discretization also uses the Euler method, the analysis and results can be naturally extended. Detailed explanations are provided in the appendix C.3.
> >
> > 6. **Notation Explanation**
> >
> >
> > Thank you for pointing out areas where the notation explanations could be improved. In response, we have provided a clearer explanation of costates (line 154) and motivations of theorems in the main text.
> >
> > Costates, also referred to as adjoint variables, play a fundamental role in optimal control theory as Lagrange multipliers for the system's dynamic constraints. In the context of Pontryagin's Maximum Principle (PMP), costates encode the cost functional where $\mu_T = \nabla_x \phi(x_T)$. Their evolution reflects how the influence of the cost function changes with the system's sensitivity, where $\mu_t$ satisfies $\frac{d\mu_t}{dt} = -\nabla_x H$, with the Hamiltonian $H$ representing the system dynamics.
> >
> > In Euclidean space, costates function similarly to gradients, and as demonstrated in Theorem 2, under a gradient guidance task, costates align with the gradient. When the system evolves on a manifold, such as a Lie group, and the states are elements of this manifold, the costates evolve within the cotangent space, which is dual to the tangent space of the manifold. While costates in this setting are no longer gradients, their flow in the cotangent space ensures that the associated state flow is consistent with the geometry of the manifold and the system dynamics.

---

> > > ### Author Response · Authors · 2024-11-22
> > >
> > > 7. **Paper Structure and Presentation**
> > >
> > >
> > > Thank you for your detailed advice regarding the presentation of the paper. However, we would like to note that the arrangement of our initial submission has already followed multiple suggestions you made. Below, we outline our proposed structure and address the issues related to the paper's presentation:
> > >
> > > - **The main and novel contributions of our paper are clearly and faithfully detailed in the Introduction section (65-84).** We provide **discussion to existing work** and their limitations in the introduction, and motivate our contribution in establishing formal OC formula with running cost that better balance proximity to prior distribution and reward optimization, as well as convergence guarantee.
> > >
> > > - We **provided background for conventional FM** and its original ODE dynamics in sec 2 (eq1), basing off which we develop OC dynamics in eq 2, and its detailed form for Euclidean (eq4) and SO3 (eq8). Since the main focus of the paper is guiding pretrained FM and not introducing new FM model, we feel the amount of context is sufficient for understanding OC-Flow.
> > >
> > > - We acknowledge that optimal control is a sophisticated topic that may require non-trivial amount of domain knowledge. Since the audience of paper may be broader, **we have made several effort to simplify the content in main text**, by keeping only the minimal required knowledge on E-MSA and PMP (which are essential for algorithm develop) and defer extensive mathematical details and backgrounds to appendix B. We also defer significant amount of proofs to appendix, and only keeping the key conclusions: proposition1 (show running cost effectively controls the divergence of sampling distribution), theorem 2 (continuous OC-Flow in Euclidean, and its convergence rate), section 4.2 (novel conclusions for E-MSA, PMP on SO3, and definition of E-Hamiltonian which is important for understanding).
> > >
> > > - In terms of **separation of subsections**, we’d like to mention that we indeed have structured our methods (Sec3,4) into subsections with specific focuses. E.g., in section 3, we start with establishing general OC framework, then split into subsection 3.1 that introduces Euclidean OC-Flow algorithm, 3.2 for practical implementations of OC-Flow-Euclidean, and 3.3 for theoretical discussion on connections to Dflow/FlowGrad. In section 4, we clearly split it into 4.1 algorithms on SO3, 4.2 convergence analysis, and 4.3 practical implementation.
> > >
> > > Overall, we believe that our mindset for presenting the paper closely resembles what reviewer jxTp has suggested, but we’re happy to address further comments related to presentation.

---

> ### Author Response · Authors · 2024-11-25
> **Looking forward to constructive discussions**
>
> Dear reviewer jxTp, once again we sincerely appreciate your detailed and constructive feedback. Following your suggestions, we have improved our manuscript and we believe our rebuttal addressed all of your comments which we summarize as the following:
>
> Following your kind suggestion, we addressed your comments on OC-Flow’s significance, discretization errors, and its significance and scalability in SO(3) tasks. **A theoretical bound for discretization error** is included in Appendix C.3, along with additional **theoretical analyses and empirical studies on the running memory and time** also provided in Appendix D and Table 6, demonstrating the better scalability of OC-Flow. The **significance of OCFlow-SO(3)** is supported by extensive ablation results in Table 5 and 12, and its scalability is detailed in Table 7 and 9. We also improved our presentation by providing more **interpretations and motivations** of our algorithm and clearer notation definitions and figures. We include extensive **experimental details** in Appendix E, and we have also provided links to our codebase to enhance **reproducibility**. We provided detailed explanations of our contributions, key notations (e.g., costate), and paper organization to address your questions.
>
> We really appreciate your valuable suggestions. We hope that our rebuttal has addressed all your concerns and questions, and we'd appreciate it if you could kindly respond if there are any further questions.

---

> > ### Author Response · Authors · 2024-12-01
> > **Gentle Reminder Regarding Review Discussion**
> >
> > Dear Reviewer jxTp,
> > Thank you again for your careful review and thorough feedback. As the discussion period is ending soon, we would greatly appreciate it if you could kindly review our rebuttals. We greatly value your feedback and believe our rebuttal has addressed all of the concerns you raised, including new theoretical results, scalability analysis, and ablation studies to further support our claims, improved paper presentation, and clarifications on our contributions and methods, following your suggestions.
> >
> > If there are any remaining points of confusion or lingering concerns, we would be happy to discuss them and provide further clarification. Thank you for your time and effort in reviewing our submission, and we look forward to your response.
> >
> > Best regards, Authors

---

> ### Comment · Reviewer_jxTp · 2024-12-03
> **Reviewed Score**
>
> I would like to thank the authors for taking the time to address some of the comments/concerns. In particular, I note the addition of significant details pertaining to the experimental setup as well as some additional analysis of the discretization error in the appendices.
>
> While some of the writing could still benefit from a bit more polish (and I note a lack of confidence intervals for many of the reported results), the authors did close some of the gaps I had mentioned. I revised my score.

---

> > ### Author Response · Authors · 2024-12-03
> > **Thank you for your recognition and your decision to raise the score**
> >
> > We are sincerely grateful that our rebuttal and clarifications can successfully address your previous concerns and questions, and we thank you for your high recognition of our theoretical contributions and additional experimental results. Your review has been constructive for us, and we will make sure our clarifications are reflected in our final revision to make our work more comprehensive, concise, and rigorous.

---

### Official Review · Reviewer_o479 · 2024-11-04

**Soundness:** 4
**Presentation:** 3
**Contribution:** 4
**Rating:** 8
**Confidence:** 2

**Summary:**

This paper provides a novel approach for guided generation using pre-trained diffusion and flow matching models. Traditional methods of guiding ODE-based generative models often require expensive retraining and work mainly on Euclidean manifolds, but OC-Flow uses an optimal control training-free framework beyond Euclidean spaces to the SO(3) manifold. Experiments on tasks like text-guided image manipulation, conditional molecule generation, and peptide design validate the method’s effectiveness​.

**Strengths:**

- As far as I know, the approach is original in framing guided flow matching as an optimal control problem. The authors develop a general framework for non-Euclidean geometries with strong theoretical backing, which is fairly rare.
- Another strength of this work is that it is a general approach, i.e. OC-Flow can be used effectively for a variety of applications such as image and molecular data.
- Unlike existing approaches, OC-Flow allows training-free guidance, making it computationally efficient and more applicable in real-life settings.
- The SO(3) results such as on improved molecular generation accuracy, validate the importance of using this geometric inductive bias for generative models.
- Through the framing of existing approaches as special cases under their optimal control formulation, this paper helps clarify the connections between gradient-based techniques like FlowGrad and D-Flow.
- The model consistently demonstrates improvement over previous work.
- I really appreciate the figures included in the paper to illustrate and compare the methods against existing approaches.

**Weaknesses:**

- My main question is regarding scalability. While the model performs well on selected benchmarks, to me it is still unclear how OC-Flow scales to high-dimensional datasets such as large molecules. A discussion of potential scalability limits and memory efficiency in such cases would strengthen the paper.
- Moreover, even though the formal contributions are great and well-formalized, to me the paper is still quite hard to read. Since the theoretical results are one of the main contributions, I think it would be valuable to add more intuitive explanations of the proofs and why they are there. For example, theorem one provides a bound based on VFM on KL between the model and a terminal point, but some intuition of why this bound is provided would make the paper more approachable.
- Adding to this point, the theoretical assumptions made (e.g. Lipschitz continuity, boundedness) are clear and needed for the argument, but some reflection (perhaps on a high level) on whether these hold in practice would help to interpret the method's advantages.
- While the focus is on continuous CNFs, a brief comparison with discrete flow techniques could contextualize OC-Flow's advantages or limitations more clearly, especially as discrete methods have shown promise in similar applications.

**Questions:**

- The paper shows promising results, but could the authors elaborate on potential ways to enhance scalability, especially when applied to e.g. large molecules or more complex target distributions in general?
- How does OC-Flow compare to recent works in Riemannian FM? What about SO(3) and SE(3)?
- Could OC-Flow be adapted for hybrid tasks where both Euclidean and Riemannian components are present? This might extend its applicability to even broader fields where you need this hybrid. Does the method allow for this directly or not?
- Since OC-Flow is designed to be computationally efficient, could the authors comment on real-time applications requiring 'immediate' guidance?

---

> ### Author Response · Authors · 2024-11-22
> **Summary of concerns**
>
> We sincerely thank your insightful feedback and your high recognition of our work’s **theoretical innovation** from the optimal control perspective and **consistent improvement** over various datasets from different ML domains. We’re happy to address your questions and concerns as follows.
>
> **Q1 Scalability and Real-time applications**
> Thank you for the perspective. We have provided a comprehensive theoretical analysis and empirical evaluation of the **runtime and memory complexity** of OC-Flow and other baselines in our revision (appendix D) and **common rebuttal**. We note that the text-image alignment task on the CelebA-HQ dataset contains images with a **resolution of 256x256x3**, which is potentially larger than the average size of protein backbone data (Lx3). We acknowledge that differentiate-through-ODE approaches (OC-flow, Dflow, FlowGrad) are by nature more computation-heavy than the posterior sampling approach, despite its outstanding guidance.
>
> Performance. Nevertheless, we have particularly **designed efficient algorithms** on both SO3 and Euclidean, e.g., asynchronized update, adjoint method, and vector-jacobian-product implementation, which significantly reduced the time and memory complexity on SO3 from O(ND^4) to O(D^2) (see table 1 and Appendix D table 7), and also advantageous compared to DFlow on Euclidean (i.e. reduce memory cost from O(ND^2) to O(D^2), and runtime efficiency from $O(cND^2)$ to $O(nD^2)$ as we don't rely heavily on L-BFGS, as shown in Table 11). Our empirical benchmark of runtime demonstrates **speedup and memory saving** compared to Dflow. We show that OC-flow samples 256x256 image in 216s, and small molecules in 38s, which might be tolerable for certain real-time applications depending on latency requirement, as opposed to Dflow which self-reported to take 15min on 128x128 image and ran OOM in our image dataset. Following your suggestion, we include discussion on scalability and time complexity in the discussion section. We acknowledge that real-time application of our method may depend on multiple factors such as size of the model, data and **hyperparameters** during sampling such as number of control terms and iterations. Still, OC-Flow provides an advantage compared to existing approaches in the same family, and would be valuable for applications where high sample quality and reward optimization is desired.
>
>
> **Q2 Intuition behind Theorems**
> We have updated the theoretical proof in our revised manuscript with **clearer notation definitions, intuitive motivations (line 167)**, and some **corrections** on some inadvertent notation errors. We’d like to clarify that the KL bond in proposition 1 is not between a model and a terminal point, but between the sample distribution of the pre-trained model and the guided model. The intuition here is to theoretically show that the running cost can control deviation from prior distribution measured by KL-divergence. We found that bounding the divergence between joint distribution p(x, x_data) gives cleaner and tighter bounds. However, we note that it is probably more ideal to bound the marginal distribution, and therefore we now provide **additional result in proposition 1** which establish the **KL bound between marginal distributions of prior and guided model** as a function of running cost, under slightly more limiting assumptions.
>
> Proposition 1 is crucial for our OC-Flow framework, as it provides **theoretical guarantees** for the optimal control formulation and **practical algorithms** for applying such a control:
>
> - By tuning the control term $\int_0^1 \|\theta_t\|^2 dt = 0$, we can indeed control how far away our guided model deviates from the pre-trained model to prevent adversarially hacking the reward.
> - By using the control term $\int_0^1 \|\theta_t\|^2 dt = 0$, we circumvent the intractable KL divergence calculation while still enjoying the theoretical benefits from optimal control.

---

> > ### Author Response · Authors · 2024-11-22
> >
> > **Q3 Theoretical Assumptions**
> >
> >
> > Regarding the theoretical assumptions, we noted that the Lipschitz continuity assumption is standard and ubiquitous in optimal control literature like the E-MSA work. **Intuitively, the Lipschitz continuity states that the vector field encoder should be smooth enough in a way that local changes in the noised data can be bounded well.** In flow matching, we assume the affine Gaussian probability path, whose intermediate noised data values are indeed smoothly interpolated along the path. Therefore, a well-trained prior model should satisfy such smoothness assumptions. A similar assumption is imposed on the reward function landscape and **its gradients to be smooth enough such that a small change in the final generation can also be bounded.** We noted that in many force fields like MadraX, the energy is smooth with respect to the coordinates. Another popular choice for reward model is neural network scores trained on feedbacks or quality scores (e.g., CLIP). Discussion on the Lipschitz behavior of NN has also been made in previous work like [1].
> >
> > We noted that these assumptions are necessary only for rigorous derivation of theoretical bounds. In practice we found the algorithms to be effective in most of the test cases we encountered. Following your advice, we now include some discussion on practicality of these assumptions.
> >
> >
> > **Q4 Comparison to Discrete Models**
> >
> >
> > We interpret your questions as a comparison with “discrete flow techniques” in the following two possible aspects:
> > 1) Comparison with models with discrete formulation like earlier MDP diffusion models, or
> > 2) Applications of OC-Flow on discrete data like natural language sequences. We will further elaborate on them as follows.
> >
> > Regarding the comparison with the diffusion model, we first noted that diffusion and flow matching models can be **converted to each other under a unified framework** (e.g., Stochastic Interpolant, [2]). Therefore, some of the diffusion-based models can be adapted for flow matching models. Conversely, OC-Flow can also be **applied to diffusion models with deterministic sampling** (e.g., DDIM). Despite the similarity, we note the optimal-control formulation is **intrinsically continuous** in the time domain, where E-MSA can be adapted for controlling the continuous flow matching model. Therefore, adapting the deterministic flow formulation is **easier and theoretically more concise** to derive the optimal-control formulation compared to SDE in diffusion models.
> >
> > Regarding the application of OC-Flow on FM for discrete data (e.g., natural language modeling), we noted that a branch of work relies on continuous parameterizations of the categorical distributions with a Euclidean assumption [3, 4]. In this way, our theoretical results on convergence and bounds still hold, and our OC-Flow can be effectively applied in these cases. Another branch of work, however, relies on discrete jumps between Markov chain states [5, 6]. As we have discussed above, our continuous formulation is incompatible with these discrete models.
> >
> > Regarding discretization of continuous **algorithm**, we now include **additional theoretical guarantees of the discretization gap** introduced by Euler method (Appendix C.3).
> >
> > **Q5 Riemannian FM and Hybrid Task**
> >
> >
> > We are fully aware of the Riemannian FM work — in fact, our peptide generation experiments are directly based upon the PepFlow model, a **special case of Riemannian FM** on the Riemannian manifold of SO(3) (all rotations) and T(3) (all translations). In our initial submission, OC-Flow on the peptide generation task is **already a hybrid task** over the special Euclidean group SE(3), which can be effectively written as the semidirect product of the SO(3) (non-Euclidean manifold) and T(3) (Euclidean manifold):
> >
> > $$
> > \mathrm{SE}(3) \cong \mathrm{T}(3) \rtimes \mathrm{SO}(3)
> > $$
> >
> > In other words, both translations and rotations are optimized simultaneously in our experiments, where performance improvement regarding the downstream energy function as the reward was also observed. In this way, our OC-Flow can be indeed **directly applied to arbitrary hybrid manifolds** as the direct or semidirect product of manifolds where the optimal control formulation can be derived.

---

> > > ### Author Response · Authors · 2024-11-22
> > >
> > > [1] Khromov, Grigory, and Sidak Pal Singh. "Some intriguing aspects about lipschitz continuity of neural networks." *arXiv preprint arXiv:2302.10886* (2023).
> > > [2] Albergo, Michael S., Nicholas M. Boffi, and Eric Vanden-Eijnden. "Stochastic interpolants: A unifying framework for flows and diffusions." *arXiv preprint arXiv:2303.08797* (2023).
> > > [3] Gat, Itai, et al. "Discrete flow matching." *arXiv preprint arXiv:2407.15595* (2024).
> > > [4] Hoogeboom, Emiel, et al. "Argmax flows and multinomial diffusion: Learning categorical distributions." *Advances in Neural Information Processing Systems* 34 (2021): 12454-12465.
> > > [5] Austin, Jacob, et al. "Structured denoising diffusion models in discrete state-spaces." *Advances in Neural Information Processing Systems* 34 (2021): 17981-17993.
> > > [6] Campbell, Andrew, et al. "Generative flows on discrete state-spaces: Enabling multimodal flows with applications to protein co-design." *arXiv preprint arXiv:2402.04997* (2024).

---

> > > > ### Comment · Reviewer_o479 · 2024-11-24
> > > >
> > > > Dear authors.
> > > >
> > > > Thanks a lot for the comments. This work seems novel, and it provides useful insight into flow models that are missing in the current literature. I raised my score to an 8.

---

> > > > > ### Author Response · Authors · 2024-11-24
> > > > > **Thank you for your recognition and for kindly raising your score**
> > > > >
> > > > > Dear reviewer o479, thank you so much for your high recognition of our contribution and your kind active support of our work. We really appreciate your time in reviewing our paper and rebuttals and your insightful review that helps make our work more comprehensive and clear.

---

### Official Review · Reviewer_faUf · 2024-11-06

**Soundness:** 2
**Presentation:** 3
**Contribution:** 2
**Rating:** 6
**Confidence:** 3

**Summary:**

This paper proposed a new framework for controlled generation using pre-trained diffusion and flow matching models, dubbed OC-Flow. The method is based on sound theory in optimal control that offers additional convergence guarantees in Proposition 1 and Theorem 2 (under two key assumptions of affince Gaussian path and Lipschitz continuity of the gradient of guided loss). Several benchmarks on guided-image manipulation, molecular generation and protein design with generative models are performed to demonstrate the effectiveness of the method.

**Strengths:**

- Well-motivated problem, overall nicely written paper with clear literature review.
- The methodological and theoretical parts of the paper are well-sounded.
- Providing a framework that has convergence analysis is always welcomed.

**Weaknesses:**

**Major: questionable and inconsistent baselines' results in empirical benchmarks**

- While on the first task (section 5.1 text-guided image manipulation) the authors have report/insert exactly other baselines' results (originally in Table 2 of the FlowGrad paper); the results on two remaining tasks in section 5.2 (molecule generation) and section 5.3 (peptide design) do not match the results reported in their respective original paper.
- More specifically, the results in Table 3 do not match those of Table 4 in D-Flow paper (Ben-Hamu et al. 2024); results in Table 5 do not match those of Table 1 in PepFlow paper (Li et al. 2024). In fact, if one instead takes into account the original results, the baseline D-Flow actually perform better in MAE metrics compared to OC-Flow in Table 3. For Table 5 the metrics reported are in different scale.
- I am therefore request the authors to clarify this discrepancies between results reported in their paper and the results reported in the respective original works of compared baselines. Otherwise, I think the practical performance of OC-Flows remains questionable.

Ben-Hamu et al. (2024). D-Flow: Differentiating through Flows for Controlled Generation. Proceedings of the 41 st International Conference on Machine Learning, Vienna, Austria. PMLR 235, 2024.

Li et al. (2024). Full-Atom Peptide Design based on Multi-modal Flow Matching. Proceedings of the 41 st International Conference on Machine Learning, Vienna, Austria. PMLR 235, 2024.

**Questions:**

See weaknesses.

---

> ### Author Response · Authors · 2024-11-22
> **Summary of concerns**
>
> Thank you for recognizing our theoretical significance, motivation and clear presentation. We fully understand your concerns, and we’d like to assure you that reproducibility has been our top priority. In fact, we spent a lot of effort in **making sure our benchmarks are valid, fair, and reproducible**, instead of unquestioningly copying numbers from papers, which we detail as below:
>
> **1. FlowGrad is reproducible**
> We note that FlowGrad authors have kindly open-sourced all necessary files (code, checkpoint, scripts) and disclosed adequate details for us to exactly reproduce their experiments. Therefore, it helps us save a lot of effort and ensure fair comparison with their reported results directly.
>
> **2. Discrepancy with D-Flow self-reported result**
> On the contrary, during our reproduction of the D-Flow baselines for molecule generation on the QM9 dataset, we noticed a **critical lack of model checkpoints, code, and reproduction instructions**, which prohibited direct comparison to their numbers. In guided generation task, both the base pretrained flow model (used as prior) and the reward model will impact the generation result, therefore we believe a **fair comparison should be done using same generative prior model and reward model**. However, DFlow paper trained their own prior model and reward model, and **as of today still have not open sourced both models**. We have tried contacting the authors but ended up hearing no response. Without the reward model, we cannot guide and evaluate our samples to match their setting. Therefore, we strongly argue that direct comparison with their table 4 is both unfair and unreasonable, due to **unwilling discrepancies in both generative priors and reward models**. In our effort to improve reproducibility, in our reproduction of the D-Flow baseline, we instead used the **publicly available checkpoint from the EquiFM model**, a flow-based molecule generative model trained on QM9. Though the base model architecture may not be as good as the newer one trained in the D-Flow paper, we believe such an approach **ensures the reproducibility and rigor** of our experiment, and benefit future benchmarking effort as everyone now has access to the model weights and can fairly compare. In our QM9 benchmark, all the methods use the **same prior model (EquiFM)** and **same reward model** we trained, and we also control the initial noises to further reduce variance.
>
> Our training of the reward model (molecule property predictor) closely resembles Dflow’s implementation, which is detailed in our appendix E.2. Our predictions (table 3) **match** the ones reported in Dflow, although there is **inevitable variances** between two NN models even if they are trained on **same data**.
>
> Our reproduction of Dflow strictly followed the hyperparameter choice in their paper and used the LBFGS optimizer with 5 inner steps and 5 outer steps (**lbfgs** is very important for Dflow to perform well according to our ablation table 11). Our experiments of the D-Flow baseline demonstrated a similar trend to the original D-Flow paper results, and the consistent but minor performance gap in all properties indicates this is a systematic behavior that should be **attributed to the difference in the pre-trained generative model**.
>
> Tested with the **same publicly available pre-trained generative model**, we believe our benchmark results are more comparable and reproducible. We will open-source our code and reward model for reproducibility once the paper gets accepted.

---

> > ### Author Response · Authors · 2024-11-22
> >
> > **3. Difference in PepFlow experiments**
> > We also encountered challenges while reproducing the PepFlow baseline for peptide generation. Several factors prevented us from directly utilizing their reported results. First, it is better to control the initial noise to reduce variance of the result when comparing unconditional and guided generation, necessitating a rerun of pepflow. Secondly, our reward function MadraX, which are essential for evaluation, **require sampled PDBs** which were unavailable in the pepflow repository and we need to regenerate. To address these issues, we reached out to the authors, who kindly provided partial scripts and verbal instructions to help reproduce their evaluation pipeline, though the paper samples and their original evaluation scripts were lost.
> >
> > We used the **publicly open-sourced PepFlow checkpoint and same test dataset provided in the repo**. We believe the discrepancies, although small in scale, could be attributed to the following reasons:
> >
> > 1) PepFlow reports **affinity% and stability%** as the percentages of peptides with improved properties compared to the native peptide. In our table 5, we report the **absolute energy values** for affinity and stability in order to get a finer-grained evaluation. Therefore the scale of these columns are not the same, as energy is usually negative and percentage is always positive.
> >
> > 2) We also observed that Rosetta evaluations, which were used for calculating stability and affinity, exhibit high variance. To mitigate this, **we performed five independent runs for each evaluation** and averaged the results to ensure robustness. Furthermore, given the time-intensive nature of Rosetta evaluations, we drew **10 samples per pocket compared to PepFlow's 64 samples**, while controlling for initial random noise across all methods. This setup allowed us to demonstrate that with guided optimization, we can achieve better results with fewer samples without compromising fairness. Our result is average across 162 pockets with 10 samples per pocket, comprising 1620 samples which is a large enough size to make sure statistical power.
> >
> > 3) We believe our reproduced pepflow result on RMSD(1.645), SSR (79.4%), BSR(87.4%) roughly matches the ones reported in table 1 of PepFlow (RMSD 2.07, SSR 83.4%, BSR 86.9%), and even **better in RMSD/BSR**.
> >
> > In our experiments, we strictly adhered to the hyperparameter settings outlined in the PepFlow paper (200 ODE steps). In our updated experiment results, we included more comprehensive ablation and compare baseline (PepFlow), OC-Flow(trans) optimizing translation only in Euclidean, OC-Flow(rot) optimizing rotation only in SO3, and OC-Flow(trans+rot) jointly optimizing translation and rotation (SE3). We also updated our results using the latest version of MadraX.
> >
> > ---
> >
> > **Summary**
> >
> >
> > These efforts underline **our commitment to rigorous and reproducible experimentation**, ensuring a **fair comparison** between methods while addressing the limitations in baseline resources. We have documented all updates in detail and will release our code upon acceptance to facilitate further research and reproducibility in this domain. We have updated our paper to include **additional experimental details, parameters, and instructions** to make sure reproducibility (Appendix E). We further provide **comprehensive ablation studies on SO3**, as well as **theoretical and actual runtime and memory complexity** of the experiments to demonstrate scalability.

---

> > > ### Comment · Reviewer_faUf · 2024-11-24
> > > **Thank you for your rebuttal**
> > >
> > > My apologies but your rebuttal left me very confused. Despite your strong claim on reproducibility of your paper and the criticism on the reproducibility of others, you decided not to provide any implementation (python code) + checkpoints of your trained mode. How am I as a reviewer can objectively evaluate the claim you made then?

---

> > > > ### Author Response · Authors · 2024-11-24
> > > > **additional resource to address your concern**
> > > >
> > > > Thank you for taking the time to respond. We would like to first emphasize that our proposed experiment setting is more fair and reproducible (e.g., compared to Dflow) particularly because it is **less dependent** on having access to implementation artifacts, specifically:
> > > > - All experiments in our paper **employed open-source publicly available models** for both the pre-trained FM model (ReFlow, EquiFM, PepFlow) and reward (CLIP, MadraX). The only place that required us to train a model is the molecule property prediction reward, which we implemented based on open-source repo following instructions in e3diffusion[Hoogeboom et al. (2022)].
> > > > - We provide a **detailed description of the algorithm** (algo1 for OC-Flow-Euclidean, algo2 for OC-Flow-SO3) and **all key equations** e.g., Vector-Jacobian product (3.2.1, equation 15) to enable precise re-implementation of OC-Flow. We also include **extensive experimental and implementation details** in Sec5 and Appendix E, including all hyperparameters and evaluation procedures to ensure reproducibility.
> > > >
> > > > We also want to clarify that our rebuttal **did not mean to criticize**, but to elaborate on the factual limitations we encountered and **address your questions** regarding why we cannot directly take the numbers from table 4 Dflow, and what is the cause for the discrepancy. Particularly:
> > > > - Copying numbers from Dflow table 4 will lead to **unfair, biased, and invalid benchmarking**, as in the guided generation task it is **crucial to utilize the same pre-trained model and reward model**, both of which are unavailable in Dflow. Based on your comment, we can not “objectively evaluate” the results in Dflow Table 4 either, and we therefore would like to avoid direct comparison with such results.
> > > > - Our reproduction of Pepflow **actually matches their statistics**, and we explained the change of scale is due to our different definitions of affinity and stability metric (pepflow uses the percentage of improvement, and ours uses **definite energy**).
> > > >
> > > > We note that code submission is **not required** for review, and we chose not to disclose it due to several concerns. **We are sorry if that creates frustration for you, and we’re now providing an anonymous URL for accessing our code and other implementation materials under the “overall comments” which is visible to ACs and reviewers only**. We request that you kindly keep it confidential and use it for only review purposes. As we promised in our rebuttals, we have already been planning to open-source our code once the paper gets accepted.
> > > >
> > > > We’d sincerely appreciate it **if you could take into consideration our written paper (e.g., methods, theoretical and empirical results, technical discussions, etc.) as the basis for your objective evaluation, and kindly evaluate our contributions and soundness from multiple perspectives**, besides reproduction of the experiments. We hope our response clears up your concerns regarding reproducibility, and we’re happy to continue the discussion if you have further questions.

---

> ### Comment · Area_Chair_XedZ · 2024-11-26
>
> Dear reviewer faUf,
>
> Thank you for engaging in discussions with the authors.
>
> I'd like to stress that sharing an anonymized version of the code is not a mandatory requirement for acceptance at ICLR. Therefore, the initial absence of shared code by the authors should not be used as a reason to recommend rejection. The only relevant guidelines that I am aware of are the [author guidelines](https://iclr.cc/Conferences/2025/AuthorGuide) on reproducibility statements. These guidelines state that a reproducibility statement is strongly encouraged, but optional, and can potentially include an anonymized link to code. Therefore, in your review, focus on judging reproducibility aspects as much as possible based on the content and information provided in the paper and the rebuttal, and where appropriate see if the shared code can help alleviate any specific concerns that you have based on the paper. If you have any remaining reproducibility concerns or concerns of other sorts after the rebuttal of the authors, please clarify clearly why this is the case so that your recommendation can be constructive to the final paper recommendation.
>
> Finally, please treat the code that the authors have kindly made available as strictly confidential.
>
> Many thanks,
>
> AC

---

> > ### Comment · Reviewer_faUf · 2024-11-27
> >
> > Dear Mr. AC,
> >
> > I request that you and the authors stop pressing me and let me perform my reviewing duty as objectively as possible.
> >
> > When the authors of a submission claimed that they could reproduce other baselines and reported a vastly different (and inferior) metrics number, while having their own method beat the baselines, my opinion is that I cannot possibly check that claim without scrutinying the implementation via the Python code. I would like to remind you that the other baselines have been peer-reviewed already, and at the moment putting more faith in the peer-review works than a submission in progress is very natural.
> >
> > What would happen if 6 months from now another submission extends this work and stated that they also cannot reproducing the current baseline?
> >
> > Once again, I beg you to please let me take my time to check the authors' claim. I will come up with my own conclusion soon.
> >
> > Yours very truly,

---

> > > ### Comment · Area_Chair_XedZ · 2024-11-27
> > >
> > > Dear authors and reviewer faUf,
> > >
> > > I think it's a good moment to try to reset how we interact with each other on this forum. At this point, both the authors and reviewer faUf have expressed concerns about feeling pressured. This isn't desirable for either party, and I'm sure no one is intending to pressure anyone, it certainly isn't my intention. So let's pay attention to this in all of our future communication. Everyone has put in a lot of work already, so going forward, let's assume we're all trying to help each other. We should now all be on the same page with respect to the ICLR's guidelines around sharing source code during the reviewing process, so let us continue to discuss the content of the submitted paper, where possible with help from the shared code.
> > >
> > > Reviewer faUf has done their best to review the paper and has expressed concerns about the discrepancy in results compared to peer reviewed papers. Checking that baselines are fairly represented is part of the reviewer's job. In turn, the authors have done their best to try to explain why reproducing some of the baselines under exactly the same conditions as the reference work is difficult due to unreleased code of that reference. The reference code not being publicly available is outside of the control of the authors of this submission. At the request of the reviewer, the authors have shared their code. Reviewer faUf has indicated that they will take time to look into the shared code and the additional details provided by the rebuttal to see if this addressed their concerns. I look forward to hearing reviewer faUf's conclusion.
> > >
> > > Kind regards,
> > >
> > > AC

---

> ### Author Response · Authors · 2024-11-27
>
> Dear AC, thank you for your kind effort in facilitating a constructive conversation and overseeing the review process for fairness and adherence to ICLR’s reviewer and author guidelines, especially for mentioning the reproducibility policy, and the focus on making judgments based on the content and information provided in the paper and the rebuttal.
>
> Dear reviewer faUf, we value constructive feedback and respect the rigor of the review process, and would like to continue the discussion on experimental differences to **further address several key points**:
>
> **Regarding your claim that we “reported vastly different metrics”**
>
> We respectfully disagree with the characterization that we reported “vastly different metrics.” For the unconditional generation with PepFlow, our implementation was **validated** by the PepFlow authors, and our statistics such as RMSD, SSR, and BSR **align with published numbers and are even better on RMSD/BSR**. Regarding Dflow, despite the significant challenges posed by the lack of access to their self-trained FM model, reward model, and Dflow implementation, we strived to produce results **that remain comparable, with deviations ranging from 3% to 12%**. We believe such deviation **does not** constitute "vastly different",  especially **when a different prior FM model and reward** is used.  We believe rerunning baselines when using new scoring functions (i.e. reward model) and experiment setting (i.e., base model) is a very common and reasonable practice, which many papers adopted without the need to excessively explain the discrepancy caused by such rerun, not mentioning that the discrepancy if caused by close-sourced nature of the baseline which we do not have control of.
>
> We also want to mention that the **phenomenon where Dflow’s performance varies with the change of the base model can be theoretically explained and understood** well. As we mentioned in the introduction, a potential limitation of Dflow is the **“strong confinement to the prior”** which “might **hinder optimization** when the target reward function diverges from the prior distribution”. Specifically, Dflow strictly projects gradients onto the model-induced manifold and only moves the noise, which **may not be sufficient or optimal**. OC-Flow instead uses multiple control terms that **allow the trajectory to deviate** from prior ODE for better reward optimization while **controlling the divergence from a prior distribution with effective running cost**. We believe the fact that **Dflow performance deviated when switching from an author-trained FM model to an open-sourced EquiFM model is exact evidence of DFlow’s strong adherence to the base model**, which aligns with our observation that there might be too few controls in Dflow to guard against imperfections in prior. OC-Flow instead **offers tunable tradeoffs** between reward maximization and faithfulness to the prior distribution, with a good theoretical **convergence guarantee**.
>
> Consider the seminal *"Attention is All You Need"* paper, a landmark in our field with over 140,000 citations. Its reproducibility has also been questioned, where BLEU scores relative differences often exceeded **20%**. As highlighted in the widely cited paper, *"A Call for Clarity in Reporting BLEU Scores"* (2,800 citations), reproducibility challenges are not uncommon for impactful works. We believe focusing disproportionately on unwilling and unavoidable reproducibility differences—especially for baselines without released code or models—risks **detracting from the broader contributions of new research**.
>
> Finally, we’d like to reiterate why the proposed comparison is unreasonable:  in a guided generation task, request to reproduce the “MAE on QM9 **author-trained reward model prediction of Dflow-guided, author-trained flow model samples**” and comparing with such result when **none of the necessary resources—the reward model, the author-trained flow model, or the Dflow implementation—are publicly available** is unfair and unreasonable.  To illustrate, this is analogous to comparing the results of “guiding SD3 with CLIP reward and method A” against “guiding SD1.5 with AlphaCLIP reward and method B”, which is unfair and scientifically unmeaningful. We also respectfully disagree with the assertion that Dflow’s **peer-reviewed status warrants blind acceptance** and use of its reported results. Furthermore, there is no evidence that Dflow has undergone the same level of scrutinization as reviewer faUf requested, and even as of today, there is no publicly available codebase from Dflow that allows academic researchers to objectively examine its claim.

---

> > ### Comment · Reviewer_faUf · 2024-11-27
> >
> > Thank you for the open discussion. It seems that the authors have their reasons and I now tend to have faith in them with the reproducibility statement. I therefore raised my score towards acceptance as my major concern has been resolved.

---

> > > ### Author Response · Authors · 2024-11-27
> > > **Thank you for your review and kind support**
> > >
> > > Dear reviewer faUf,
> > >
> > > We sincerely thank you for your time and effort in thoroughly evaluating our work and your open-mindedness to accept the practical difficulties we encountered. We also appreciate the discussion which helped us re-examine our claims and made the paper stronger. Thank you for your recognition of our strength and your kind support of our work. We will make sure to further polish our paper and release code bases to maintain the standard of reproducibility.

---

### Author Response · Authors · 2024-11-22
**Overall Response**

We sincerely thank our reviewers for their insightful feedback which enabled us to further improve our paper. We greatly appreciate reviewers recognition of our clear presentation (faUf, o479, zUf2), theoretical significance (all reviewers), soundness of method (faUf, o479), importance of problem setting(faUf,o479,zuF2), novel contribution(o479,zuF2), and empirical superiority(o479,zuF2). We would like to address several common feedbacks and highlight key improvements we made with the following response:

**Additional experimental results**

- To further establish the strength of our OC-Flow-SO3, we conduct new peptide design experiments with **comprehensive ablations to study the effectiveness of Euclidean, SO3, and joint Euclidean+SO3 OC-Flow**. As shown in table 5, applying OC-Flow on rotation(SO3) or translation(Euclidean) alone effectively improves the results, while applying SO3+Euclidean OC-Flow jointly achieved the best performance, doubling the increase of the Madrax score from 0.34(single modal) to 0.68(joint).

- We further include ablation on SO3 to show the **necessity of OC-Flow-SO3 on guiding SO3 FM**(table 12), where naively applying gradient update with projection to tangent space of SO3 fails to optimize and leads to worse results than unconditional samples.

- We study the sensitivity to optimizers in QM9, where we show that DFlow heavily relies on L-BFGS (used in their paper) which leads to high runtime complexity, whereas OC-Flow achieves similar strong performance without L-BFGS (table 11), enabling efficient implementation.

**Scalability and runtime**

- To address the requests from all reviewers regarding runtime and scalability, we now provide a comprehensive **theoretical analysis** and **empirical evaluation** of the runtime and memory complexity of OC-Flow and other baselines in our revision (appendix D). We show that our efficient algorithm on both SO3 and Euclidean (e.g., with **asynchronized update**, **adjoint method** and **vector-Jacobian product formulation**) significantly reduced the time and memory complexity on SO3 from $O(ND^4)$ to $O(D^2)$ (table 6/7). On Euclidean, our method is better than Dflow in terms of memory ($O(ND^2)$ ->$O(D^2)$) and runtime ($O(cND^2)$ ->$O(nD^2)$).

- Our empirical benchmark of runtime demonstrates **speedup and memory saving** compared to Dflow. We show that OC-flow samples 256x256 images in **216s**, and small molecules in **38s**, as opposed to Dflow which self-reported to take **15min** on 128x128 images and ran OOM in our 256x256 image data.

- We also show that **OC-Flow-SO3 can be efficiently implemented with the same order of complexity as OC-Flow-Euclidean**, and runtime of **296s (SO3)** vs. **188s (EU)** confirms the analysis. With the above, we make sure that OC-Flow is scalable and efficient.

**Reproducibility**

- We provide more extensive experimental details in Appendix E, including hyperparameters and implementation details to make sure reproducibility. We plan to open source our code soon.
- Our efforts for **fair, valid and reproducible experiments** includes:
  1. Making sure benchmarks are controlled, comparing all methods **using same publicly available open-sourced pretrained FM model and reward model**.
  2. Reporting metrics with large number of samples (~10^3) for significance.
  3. Reaching out to authors and following exact same configuration when implementing baselines.

**Additional Theoretical Results**
- We provide additional analysis on **discretization error bounds** of our continuous algorithm in C.3. As our discretization is Euler method of ODE, we prove the discretization error is of the order $O(\Delta t)$ and becomes negligible with sufficiently dense steps. In practice, we use 100 time steps for text2img experiment, 50 timesteps for QM9 and 200 time steps for Peptide design, which makes the assumption valid.
- We extend our proposition 1 to also provide additional **KL bound between marginal distributions of prior and guided model** as a function of running cost. The intuition here is to theoretically show that the running cost can control deviation from prior distribution, which is key to our algorithm motivation.

**Presentation**
- Following advises from reviewers, we improved our presentation of contribution, theory, figures, and notations, and provide more intuitions that help readers understand the significance of the work. We also include discussions on efficiency-performance tradeoffs in inference-time guided generation approaches, as well as scalability of OC-Flow.


We are sincerely grateful for the reviewers' time and their insightful feedback that helped us improve the practical impact of the paper. We believe our efforts have addressed all of the reviewers' comments, and we welcome further discussions.

---

### Comment · Area_Chair_XedZ · 2024-11-25
**Last day for reviewers to ask questions to the authors!**

Dear reviewers,

Tomorrow (Nov 26) is the last day for asking questions to the authors. With this in mind, please read the rebuttal provided by the authors and their latest comments, as well as the other reviews. If you have not already done so, please explicitly acknowledge that you have read the rebuttal and reviews, provide your updated view and score _accompanied by a motivation_, and raise any outstanding questions for the authors.

**Timeline**: As a reminder, the review timeline is as follows:
- November 26: Last day for reviewers to ask questions to authors.
- November 27: Last day for authors to respond to reviewers.
- November 28 - December 10: Reviewer and area chair discussion phase.

Thank you for your hard work,

Your AC

---

### Author Response · Authors · 2024-12-03
**Sincere Gratitude from Authors**

Dear Reviewers,


We sincerely thank all reviewers for their invaluable time and effort during the review process. We are encouraged by Reviewer fauf’s faith in reproducibility, Reviewer o479’s recognition of the paper’s value, Reviewer jxtp’s acknowledgment of its “strong theoretical grounding,” and Reviewer zuF2’s emphasis on its novelty. We are grateful that our rebuttals have successfully addressed your questions and concerns, and we deeply appreciate the recognition of our contributions of the proposed OC-flow framework.


The interactive discussions during the rebuttal period have been both inspiring and productive, providing valuable insights that have been instrumental in enhancing our work. We are committed to polishing the paper further to build a more comprehensive and mathematically rigorous manuscript. Once again, we sincerely thank the reviewers for their thoughtful engagement, which has greatly contributed to improving the quality and clarity of our research.


Warm regards,
The Authors

---

### Meta-Review · Area_Chair_XedZ · 2024-12-20

**Metareview:**

Among other things, the reviewers have highlighted the nice framing of guided flow matching as an optimal control problem, the contribution to formalizing and extending the existing guided-flow matching techniques to SO(3), and the applicability of the proposed approach to various domains such as image and molecular data. Furthermore, it was highlighted that the framing of existing work as special cases under the optimal control formulation helps clarify connections between different methods. Finally, the results were found promising.

As important areas for improvement, both reviewer faUf and jxTp had questions regarding the experimental setup that required clarification, and reviewer faUf raised concerns about the results of baseline methods and how they compare against what is reported previously in the literature. Another important area for improvement suggested by multiple reviewers was to provide more details on runtime comparisons. Other points include questions around how the theoretical assumptions hold in practice (raised by two reviewers). In particular, reviewer jxTp raised questions on the use of discretization in practice while the theory provided relies on a continuous framework. Furthermore, reviewer jxTp suggested that an ablation study on optimization in SO(3) would be helpful to demonstrate the additional benefits over Euclidian optimization. Finally, reviewer jxTp also asked for a clearer exposition of the contributions of the paper, while other reviewers found this clearly illustrated. Finally, several reviewers provided suggestions for clarifications and explanations to improve the accessibility of the paper, they

The two major concerns were addressed in the rebuttal and revised version of the paper. The concerns on reproducibility and details of the experimental setup were addressed by sharing anonymized code, and including additional details in the revised paper, as well as explanations in the rebuttal itself. The second major point regarding runtime comparison and scalability was addressed by including a discussion on time and memory complexity in appendix D, as well as practical runtime and memory usage. The majority of the other concerns were addressed as well. The authors have included a discussion and theoretical analysis of the discretization gap in the revised paper. The authors have also incorporated the suggestion to include an ablation study to demonstrate the benefit of optimization in SO(3).  Finally, the authors have also taken into account several suggestions by the reviewers to improve the clarity of the paper. While reviewer jxTp still finds there to be room for improvement on the exposition of the paper, they have indicated that several other concerns have been addressed and raised their score.

In summary, most of the concerns initially raised by the reviewers were addressed through fruitful discussions between the authors and reviewers, and revisions in the submitted paper. Given multiple raised scores, there is now unanimous agreement among reviewers that this paper should be accepted. Therefore I recommend to accept this paper.

**Additional Comments On Reviewer Discussion:**

Reviewer faUf and jxTp initially had questions regarding reproducibility of the work, the presented results of baseline methods and the training details. The authors have provided access to an anonymized version of the code base, and provided more details about training settings in their rebuttal. Reviewer faUf has taken their time to look at the code base, and has increased their score to be above the acceptance threshold. Reviewer jxTp has increased their score to above the acceptance threshold given the clarifications provided by the authors. Reviewer o479 has also increased their score after the rebuttal, stating the work provides useful insights that are missing in the current literature.

---

### Decision · Program_Chairs · 2025-01-22

Accept (Poster)